# Child wasting and concurrent stunting in low- and middle-income countries

Andrew Mertens[1✉], Jade Benjamin-Chung[1,2,3], John M. Colford Jr[1], Alan E. Hubbard[1], Mark J. van der Laan[1], Jeremy Coyle[1], Oleg Sofrygin[1], Wilson Cai[1], Wendy Jilek[1], Sonali Rosete[1], Anna Nguyen[1], Nolan N. Pokpongkiat[1], Stephanie Djajadi[1], Anmol Seth[1], Esther Jung[1], Esther O. Chung[1], Ivana Malenica[1], Nima Hejazi[1], Haodong Li[1], Ryan Hafen[4], Vishak Subramoney[5], Jonas Häggström[6], Thea Norman[7], Parul Christian[8], Kenneth H. Brown[9], Benjamin F. Arnold[10,11✉] & The Ki Child Growth Consortium*

Sustainable Development Goal 2.2—to end malnutrition by 2030—includes the elimination of child wasting, defined as a weight-for-length $z$-score that is more than two standard deviations below the median of the World Health Organization standards for child growth[1]. Prevailing methods to measure wasting rely on cross-sectional surveys that cannot measure onset, recovery and persistence—key features that inform preventive interventions and estimates of disease burden. Here we analyse 21 longitudinal cohorts and show that wasting is a highly dynamic process of onset and recovery, with incidence peaking between birth and 3 months. Many more children experience an episode of wasting at some point during their first 24 months than prevalent cases at a single point in time suggest. For example, at the age of 24 months, 5.6% of children were wasted, but by the same age (24 months), 29.2% of children had experienced at least one wasting episode and 10.0% had experienced two or more episodes. Children who were wasted before the age of 6 months had a faster recovery and shorter episodes than did children who were wasted at older ages; however, early wasting increased the risk of later growth faltering, including concurrent wasting and stunting (low length-for-age $z$-score), and thus increased the risk of mortality. In diverse populations with high seasonal rainfall, the population average weight-for-length $z$-score varied substantially (more than 0.5 $z$ in some cohorts), with the lowest mean $z$-scores occurring during the rainiest months; this indicates that seasonally targeted interventions could be considered. Our results show the importance of establishing interventions to prevent wasting from birth to the age of 6 months, probably through improved maternal nutrition, to complement current programmes that focus on children aged 6–59 months.

Wasting is a form of undernutrition that results from a loss of muscle and fat tissue from acute malnutrition. It affects an estimated 45 million children under the age of 5 years (6.8%) worldwide, with more than half of these children living in South Asia[2]. Children are considered wasted if their weight-for-length $z$-score (WLZ) is more than two standard deviations below the median of international standards, and severe wasting is defined as three standard deviations below the median[3]. Wasted children have weakened immune systems[4], predisposing them to infections and more severe illness once infected[5]. Wasting in very young children increases their risk of mortality (relative risk: 2.3), and this risk increases further if they are also stunted and underweight (defined as a length-for-age $z$-score (LAZ) below −2 and a weight-for-age $z$-score (WAZ) below −2, respectively; relative risk: 12.3; ref. 6). Longitudinal studies have shown that wasting is also associated with a higher risk of stunting[7] and poor neurocognitive development at older ages[8,9]. Community-based identification and management of severely wasted children (WLZ below −3), including treatment with high-calorie, ready-to-use-therapeutic foods, has proven effective[10], but rates of relapse and mortality remain high in this fragile subgroup[11,12]. Primary prevention of wasting would be preferable to treatment, but there are few proven preventive interventions, especially in infants under 6 months. Nutritional intervention trials have had mixed success, with no evidence for reductions in wasting with breastfeeding support from birth, but with larger reductions achieved by administering small-quantity lipid-based nutrient supplements between the ages of 6 months and 24 months (14% relative reduction in wasting; 31% relative reduction in severe wasting)[10,13,14].

[1]Division of Epidemiology and Biostatistics, University of California, Berkeley, Berkeley, CA, USA. [2]Department of Epidemiology and Population Health, Stanford University, Stanford, CA, USA. [3]Chan Zuckerberg Biohub, San Francisco, CA, USA. [4]Hafen Consulting, West Richland, WA, USA. [5]DVPL Tech, Dubai, United Arab Emirates. [6]Cytel, Waltham, MA, USA. [7]Quantitative Sciences, Bill & Melinda Gates Foundation, Seattle, WA, USA. [8]Center for Human Nutrition, Department of International Health, Johns Hopkins Bloomberg School of Public Health, Baltimore, MD, USA. [9]Department of Nutrition, University of California, Davis, Davis, CA, USA. [10]Francis I. Proctor Foundation, University of California, San Francisco, CA, USA. [11]Department of Ophthalmology, University of California, San Francisco, CA, USA. *A list of authors and their affiliations appears at the end of the paper. ✉e-mail: amertens@berkeley.edu; ben.arnold@ucsf.edu

Wasting is thought to occur mainly from 6–24 months—the crucial period in which exclusive breastfeeding is no longer recommended, and adequate, appropriate and safe complementary feeding is required[15,16]. However, a more complete understanding of the epidemiology of wasting and how it varies by age is key to developing and targeting preventive interventions[15,17,18]. Unlike the cumulative process of linear growth faltering and stunting[19], wasting varies considerably over time, both within individuals and within populations[20,21]. The dynamic nature of wasting means that the number of distinct episodes a child experiences might be poorly captured in cross-sectional, survey-based estimates[22]. Furthermore, seasonal patterns of wasting onset in relation to changes in food insecurity or disease can only be measured accurately with longitudinal data[18,23,24]. A synthesis of longitudinal cohort data across diverse populations provides a unique opportunity to study the timing, dynamics and burden of wasting in infants and young children, and to fill key knowledge gaps to inform future prevention efforts.

## Pooled longitudinal analyses

Here we report a pooled analysis of 21 longitudinal cohorts from 10 low- or middle-income countries (LMICs) in South Asia, sub-Saharan Africa and Latin America, in which length and weight were measured monthly in children of 0–24 months. Our primary objectives were to produce large-scale estimates of wasting incidence and recovery and to quantify the temporal and regional variation. We also assessed the concurrence of wasting and stunting and compared estimates of wasting prevalence and incidence. We analysed data from the Bill & Melinda Gates Foundation's Knowledge Integration (Ki) database, which has aggregated studies on child growth and development. Inclusion criteria were defined to select cohorts that are representative of general populations in LMICs, with sufficiently frequent measurements to investigate the acute, dynamic nature of wasting. Cohorts were included if they: (1) were conducted in LMICs; (2) enrolled children between birth and the age of 24 months and measured their length and weight repeatedly over time; (3) did not restrict enrolment to acutely ill children; (4) had a median birth year of 1990 or later; and (5) measured anthropometry at least monthly (Extended Data Fig. 1). Twenty-one cohorts met the inclusion criteria, encompassing 11,448 children and 198,154 total anthropometry measurements from birth to the age of 24 months (Fig. 1).

We calculated WLZs and LAZs using World Health Organization (WHO) 2006 growth standards[25]. We omitted 385 biologically implausible measurements (0.2%) of WLZ (z-scores of greater than 5 or less than −5), 212 (0.1%) measurements of LAZ (z-scores of greater than 6 or less than −6) and 216 measurements (0.1%) of WAZ (z-scores of greater than 5 or less than −6), following WHO recommendations[26]. Most included cohorts did not measure children after the age of 24 months, so analyses focused on birth to 24 months. Cohorts ranged in size from 160 children in the Transmission Dynamics of Crypto (TDC) cohort to 2,931 children in the MRC Keneba cohort (Fig. 1). Unless otherwise indicated, we conducted individual-level analyses within cohorts and then pooled cohort-specific estimates using random-effects models fitted with restricted maximum likelihood estimation—a conservative approach when cohort-specific estimates are heterogeneous. Cohorts were distributed throughout South Asia, Africa and Latin America, but did not cover entire regions (Extended Data Fig. 2). Most cohorts measured children every month through to the age of 24 months, but there was some attrition as children aged and there were four cohorts with few measurements beyond the age of 15 months (Extended Data Fig. 3). Pooled estimates at older ages could be slightly biased if unmeasured cohorts or children were systematically different from those that were included; for example, if children who were lost to follow-up were more likely to be wasted, we might have underestimated wasting at older ages.

We assessed the representativeness of Ki cohorts by comparing z-score measurements to population-based samples in demographic and health surveys (DHSs). Mean z-scores in Ki cohorts were generally representative on the basis of country-level DHS estimates, with lower WLZs (overall and by age) in Guatemala, Pakistan and South Africa and higher WLZs in Brazil and Guatemala (Extended Data Fig. 4). LAZs were generally lower in Ki cohorts, so estimates of concurrent wasting and stunting may be higher than estimates from population-based samples, and rates of wasting recovery in early life may be higher compared with the general population.

## Age-specific patterns of wasting

Across all cohorts, the mean WLZ was near −0.5 at birth and then increased over the first 6 months before decreasing until 12 months (Fig. 2a). Age-specific patterns in WLZ were similar across geographical regions, but levels varied substantially by region. WLZs were markedly lower among South Asian cohorts, reflecting a much higher burden of malnutrition (Fig. 2a). Children were wasted for 16,139 (8.6%) measurements, and severely wasted for 3,391 (1.8%) measurements. Wasting prevalence was highest at birth (11.9%; 95% confidence interval (CI): 7.0, 19.5), in contrast with previous studies that showed that wasting prevalence peaked between 6 months and 24 months[7,27–29] (Fig. 2b). Across regions, the wasting prevalence decreased from birth to 3 months before increasing until the age of 12 months; however, wasting was much more prevalent in South Asian cohorts. In South Asian cohorts, in which low birthweight is common[30], the at-birth wasting prevalence was 18.9% (95% CI: 15.0, 23.7), implicating causes of poor fetal growth such as maternal malnutrition, maternal morbidities and maternal small stature as key regional drivers of wasting[31,32]. Severe wasting followed a similar pattern but was much rarer (Extended Data Fig. 5a).

Wasting onset was highest during the first three months of life, owing largely to its high occurrence at birth (Fig. 2c). Overall, 12.9% (95% CI: 7.6, 20.1) of all children experienced wasting by the age of 3 months, and this accounted for almost half (47.8%) of children who ever experienced wasting in their first two years of life. Focusing on wasting prevalence alone masked how common it was for children to experience wasting in their first 24 months. After birth, up to 6.5% (95% CI: 4.9, 8.6) of children were wasted at a specific visit, but 29.2% (95% CI: 17.5, 44.7) of children experienced at least one wasting episode by 24 months of age. Cumulative incidence was highest in South Asian cohorts (52.2% (95% CI: 43.5, 60.7); Fig. 2b,c). Early onset of wasting was consistently high across countries with different levels of health spending, poverty and under-5 mortality, with early wasting particularly high (18.3% (95% CI: 13.5, 24.4) at birth) in the five cohorts with birth measures and a national health expenditure of less than 4.5% of gross domestic product (Extended Data Fig. 6). WHO child growth standards overestimate wasting at birth among children born prematurely[33], although adjusting for gestational age among four cohorts with available data only reduced the at-birth stunting prevalence by 0.8% and increased the prevalence of being underweight by 0.7% (Extended Data Fig. 7). Even if premature birth were to account for some of the at-birth wasting, the small birth size, irrespective of cause, documented in this analysis raises concern because of its consequences for child growth faltering and mortality during the first 24 months of life[32].

## Wasting incidence and recovery

There was high variability in WLZ across longitudinal measurements, with an average within-child standard deviation of WLZ measurements of 0.76, compared to 0.64 for LAZ and 0.52 for WAZ. We thus defined unique wasting episodes by imposing a 60-day recovery period, covering two consequent monthly measurements, in which a child's WLZ measurements needed to remain

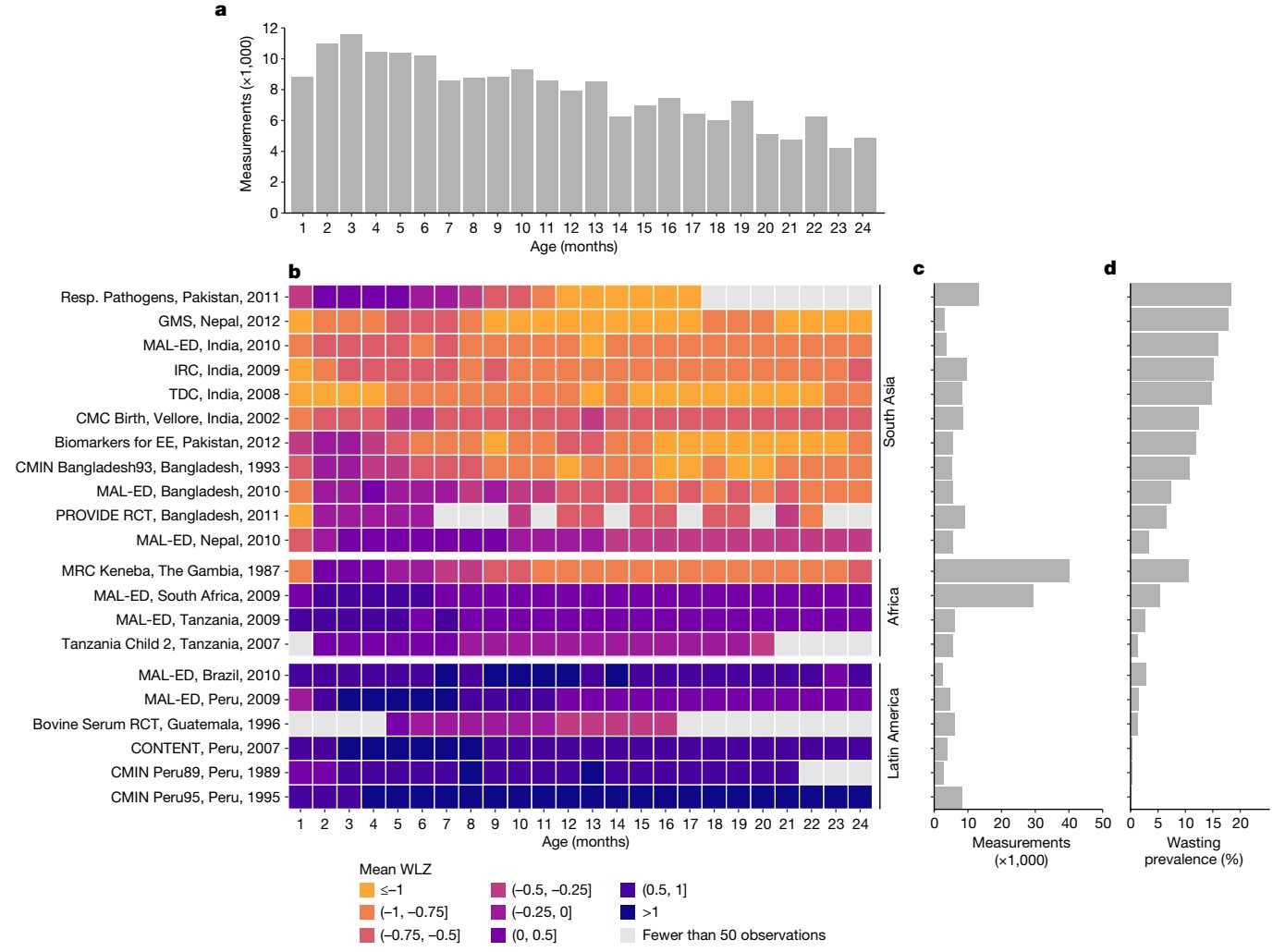

**Fig. 1 | Summaries of included cohorts. a**, Number of observations (in thousands) across cohorts by age in months. **b**, Mean WLZ by age in months for each included cohort. Cohorts are sorted by geographical region and overall mean WLZ. The country and the start date of each cohort are included.

GMS, Growth Monitoring Study; IRC, Immune Response Crypto. **c**, Number of observations included in each cohort. **d**, Overall wasting prevalence by cohort, defined as the proportion of measurements with WLZ < −2.

above −2 for the child to be considered recovered and at risk for a future episode (Fig. 3a). Children who were born wasted were considered to have an incident episode of wasting at birth. More than half of all first wasting episodes occurred before the age of 9 months (median age of first-episode onset: 264 days), with a later onset for recurrent episodes among children who experienced them (median age of second-episode onset: 449 days; median age of third-episode onset: 551 days). The mean number of wasting episodes experienced and wasting incidence at all ages was higher in South Asia, with the highest incidence in the first three months, with or without including episodes at birth (Fig. 3b). Ultimately, most children recovered from moderate or severe wasting episodes, reinforcing the status of wasting as an acute condition. Children met our definition of 'recovered' in 91.5% of episodes of moderate and 82.5% of episodes of severe wasting. Loss to follow-up might have affected recovery estimates because recovery can only occur in children who survive the episode. We did not have data on wasting treatment, so recovery might be more common in these highly monitored cohorts than in the general population, owing to treatment referrals. Across all cohorts, children recovered from 39.0% (95% CI: 34.0, 44.3) of wasting episodes within 30 days, 64.7% (95% CI: 59.3, 69.4) within 60 days and 71.1% (95% CI: 66.1, 75.6) within

90 days, with lower recovery in South Asian cohorts compared with other regions (Fig. 3c).

We examined the distribution of WLZ in the three-month period after a child was considered to be recovered from wasting, stratified by the age at which the episode occurred. We found that children recovered to a higher WLZ after at-birth wasting or wasting episodes in the 0–6-month age window, as compared with episodes that occurred at older ages (Fig. 3d). Regression to the mean (RTM) could explain some wasting recovery, especially the rapid increase in WLZ among children who were born wasted (Fig. 3e). However, there was catch-up growth beyond the calculated RTM effect[34] (Extended Data Fig. 8), and we required a 60-day recovery period to better capture true recovery. In addition, placental insufficiency causes a large proportion of intrauterine growth restriction, leading to birth wasting, and rapid WLZ gain after might reflect true recovery to a child's growth potential after this constraint is removed[35,36]. Consistent with larger increases in WLZ during recovery from wasting among younger children, a larger proportion of children recovered within 30, 60 and 90 days if the wasting episode occurred before the age of 6 months (Fig. 3d). Younger children had a more variable WLZ (s.d. 0.81 before 6 months; 0.54 from 6–24 months) and shorter wasting episodes (Extended Data Fig. 9). Wasting episodes were also longer at all ages in South Asian compared to African or Latin

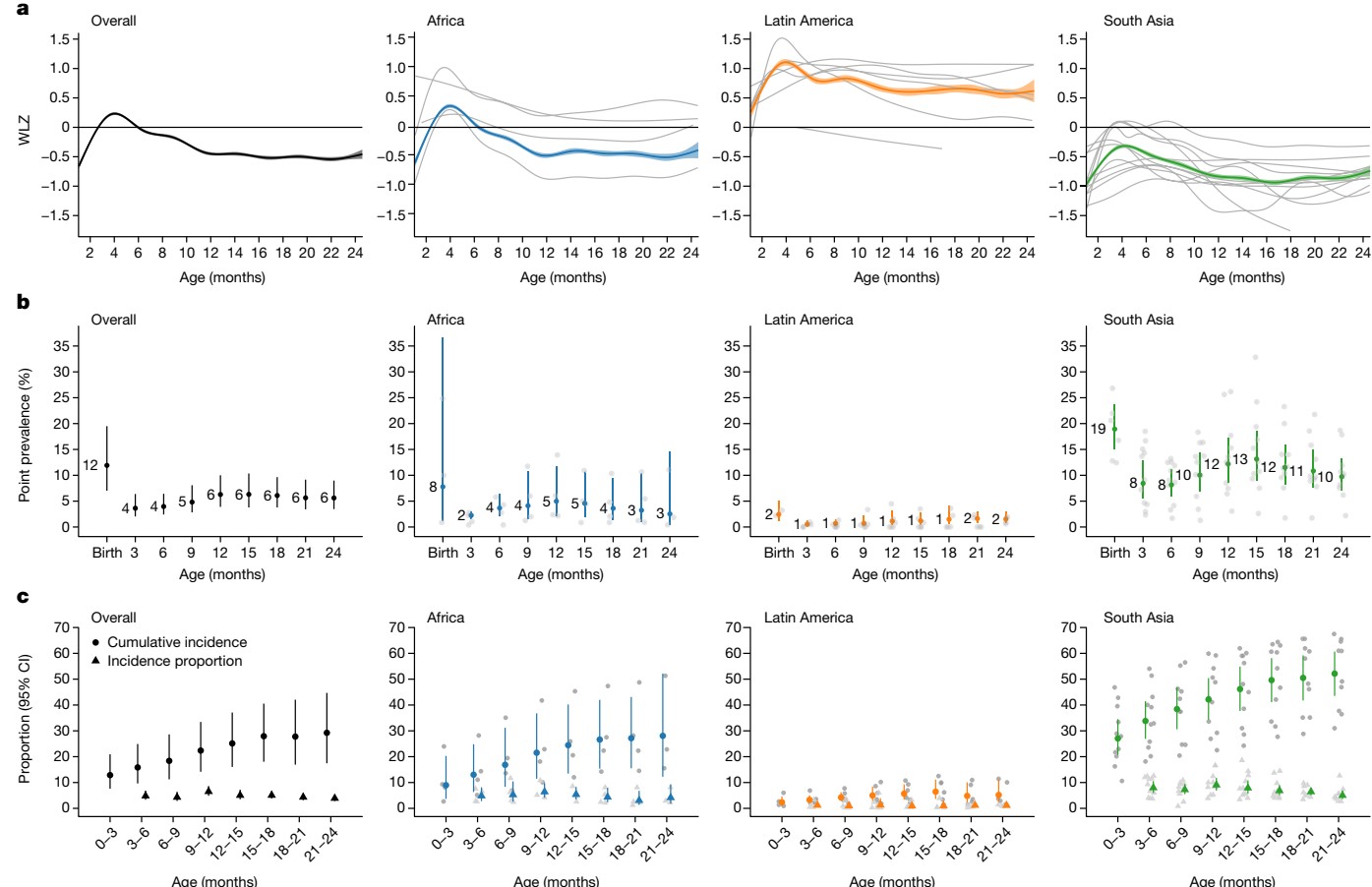

**Fig. 2 | WLZ, prevalence and incidence of wasting by age and region. a**, Mean WLZ by age in 21 longitudinal cohorts, overall (*n* = 21 studies; *n* = 4,165–10,886 observations per month) and stratified by region (Africa: *n* = 4 studies, *n* = 1,067–5,428 observations; Latin America: *n* = 6 studies, *n* = 569–1,718 observations; South Asia: *n* = 11 studies, *n* = 2,382–4,286 observations). **b**, Age-specific wasting prevalence, defined as WLZ < −2, overall (*n* = 3,985–9,906 children) and stratified by region (Africa: *n* = 1,701–5,017 children; Latin America: *n* = 290–1,397 children; South Asia: *n* = 1,994–3,751 children). The median *I²* statistic measuring heterogeneity in each meta-analysis was 97 (interquartile range (IQR) = 96–97) overall, 96 (IQR = 90–97) in Africa, 2 (IQR = 0–54) in Latin America and 92 (IQR = 88–94) in South Asia. **c**, Age-specific wasting

incidence overall (*n* = 6,199–10,377 children) and stratified by region (Africa: *n* = 2,249–5,259 children; Latin America: *n* = 763–1,437 children; South Asia: *n* = 3,076–3,966 children). Cumulative incidence measures the proportion of children who have ever experienced wasting since birth, and the new incident cases represent the proportion of children at risk who had an episode of wasting that began during the age period. The median *I²* statistic measuring heterogeneity in each meta-analysis was 99 (IQR = 92–99) overall, 98 (IQR = 92–99) in Africa, 56 (IQR = 50–78) in Latin America and 95 (IQR = 82–96) in South Asia. Error bars in **b**,**c** are 95% CI for pooled estimates. In each panel, grey curves or points show cohort-specific estimates.

American cohorts (Extended Data Fig. 9). Longer episodes among older children is consistent with higher prevalence and lower incidence compared with children under 6 months.

On average, the WLZ of children born wasted did not catch up with that of children not born wasted (Fig. 3e). Notably, children who were born wasted but who recovered had a higher cumulative incidence of wasting after the age of 6 months (38.1% cumulative incidence in children born wasted (95% CI: 28.2, 49.2) versus 24.2% cumulative incidence in children not born wasted (95% CI: 17.6, 32.4); Fig. 3f). Longitudinal analyses showed a negative, reinforcing relationship between wasting and stunting: at nearly every age, wasting increased the risk of future stunting and stunting increased the risk of future wasting, with the strength of the relationship increasing as children aged. This observation is consistent with longer durations of wasting episodes (Extended Data Fig. 10).

## Seasonality of wasting

We joined monthly rainfall totals to cohorts by location and year. For each cohort, we estimated a seasonality index based on rainfall

(details in Methods). We examined both the average WLZ over the calendar year and seasonal changes with respect to rainfall, under the hypothesis that seasonal changes in food availability and infection associated with rainfall could cause seasonal wasting[37]. The mean WLZ varied markedly by calendar date in almost all cohorts, with a consistent minimum mean WLZ coinciding with peak rainfall (Fig. 4a). When pooled across cohorts, the mean WLZ was −0.16 (95% CI: −0.19, −0.14) lower during the three-month period of peak rainfall, compared to the mean WLZ in the opposite three-month period of the year, (Fig. 4b). The mean seasonal decline in WLZ was −0.27 (95% CI: −0.31, −0.24) in cohorts with a seasonality index higher than 0.9, but some cohorts showed seasonal WLZ declines of more than −0.5 (Fig. 4b).

Children who experienced seasonal wasting during the rainy season had longer episodes and were at higher risk of future wasting episodes. Rainy-season wasting episodes lasted 7 days longer than dry season episodes (median 42 days (95% CI: 36.5, 45.5) versus 35 days (95% CI: 33.5, 38.5)). Incident wasting during the rainy season increased a child's risk of experiencing another wasting episode in the following dry season (relative risk: 2.1, 95% CI: 1.3, 3.4), and children

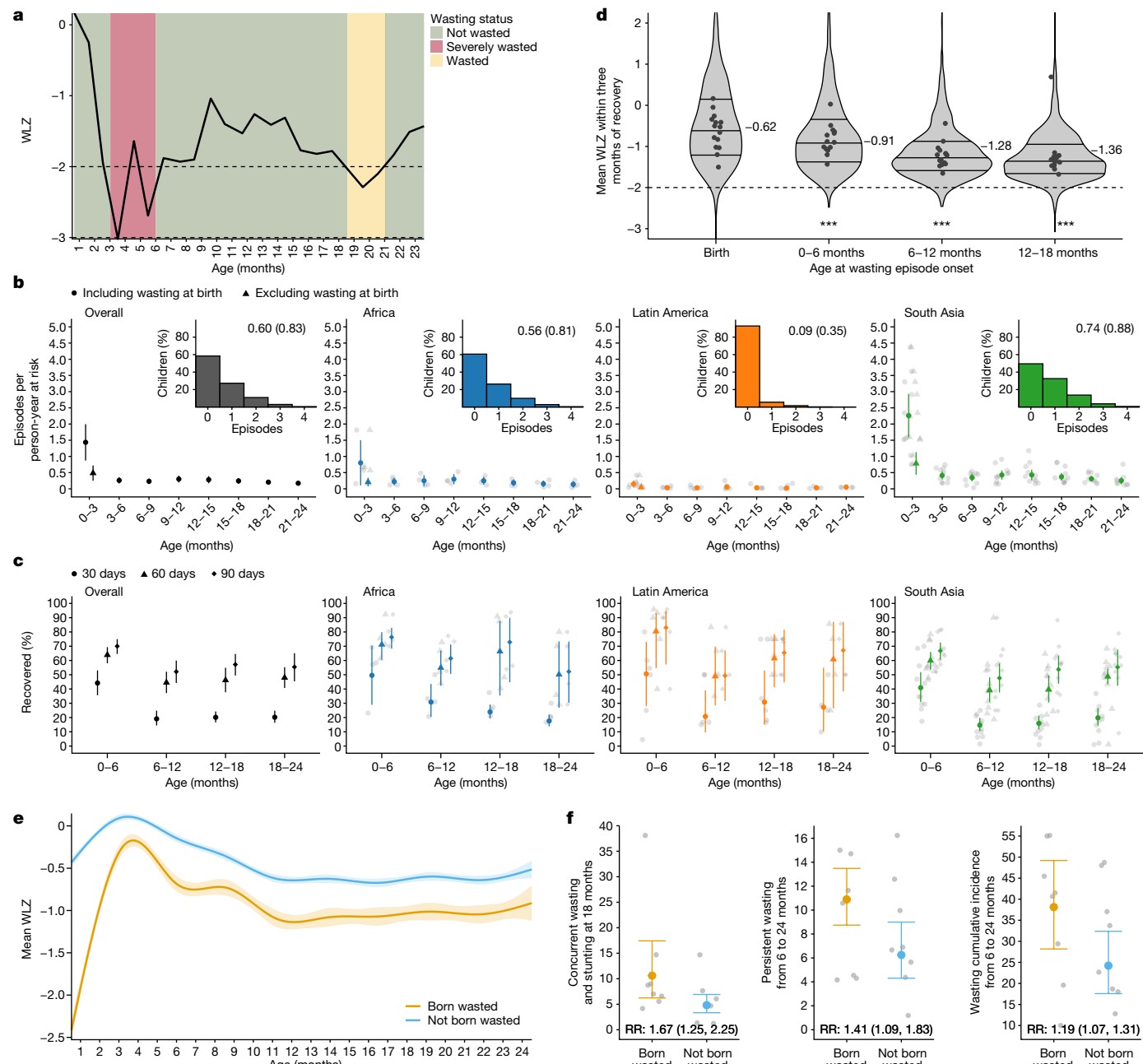

**Fig. 3 | Wasting incidence rate and recovery. a**, Example child WLZ trajectory and wasting-episode classifications. The age of wasting onset was assumed to occur halfway between a measurement of WLZ < −2 and the previous measurement of WLZ ≥ −2. Recovery from an episode of wasting or severe wasting occurred when a child had measurements of WLZ ≥ −2 for at least 60 days, with the age of recovery assumed to be halfway between the last measurement of WLZ < −2 and the first measurement of WLZ ≥ −2. The dashed horizontal line marks the cutoff used for moderate wasting, −2 WLZ. **b**, Wasting incidence rate per 1,000 days at risk, stratified by age and region (overall: *n* = 510,070–822,802 person-days per estimate, Africa: *n* = 245,046–407,940 person-days; Latin America: *n* = 65,058–142,318 person-days; South Asia: *n* = 182,694–315,057 person-days). Insets are histograms of the number of wasting episodes per child by region, with the distribution mean (s.d.) printed within the plot. The median *I*² statistic measuring heterogeneity in each meta-analysis was 97 (IQR = 96–98) overall, 92 (IQR = 90–94) in Africa, 27 (IQR = 2–68) in Latin America and 91 (IQR = 87–94) in South Asia. **c**, Percentage of children who recovered from wasting within 30, 60 and 90 days of episode onset (*n* = 21 cohorts, 5,549 wasting episodes). The median *I*² statistic measuring heterogeneity in each meta-analysis was 78 (IQR = 66–80) overall, 82 (IQR = 76–87) in Africa, 1 (IQR = 0–31) in Latin America and 77 (IQR = 45–81) in South Asia.

**d**, The distribution of children's mean WLZ in the three months after recovery from wasting, with the IQR marked. The median WLZ is annotated, and grey points mark cohort-specific medians. Children wasted before the age of 6 months experienced larger improvements in WLZ compared with children who were wasted at older ages (*P* < 0.001). The analysis uses 3,686 observations of 2,301 children who recovered from wasting episodes, with 1,264 observations at birth, 628 observations from 0–6 months, 824 observations from 6–12 months and 970 observations from 12–18 months. The dashed horizontal line marks the cutoff used for moderate wasting, −2 WLZ. **e**, Mean WLZ by age, stratified by wasting status at birth (which includes the first measure of a child within seven days of birth), shows that children born wasted (*n* = 814 children, 14,351 observations) did not catch up with children who were not born wasted (*n* = 3,355 children, 62,568 observations). **f**, Increased risk of multiple types of wasting after the age of 6 months among children born wasted (*n* = 814), with the relative risk (RR) for born wasted versus not and 95% CI printed at the bottom (*n* = 3,355). The median *I*² statistic measuring heterogeneity in each meta-analysis was 73 (IQR = 67–81) overall, 98 (IQR = 92–99) in Africa, 56 (IQR = 50–78) in Latin America and 95 (IQR = 82–96) in South Asia. In **b**,**c**,**f**, vertical lines mark 95% CI for pooled means of study-specific estimates (light points).

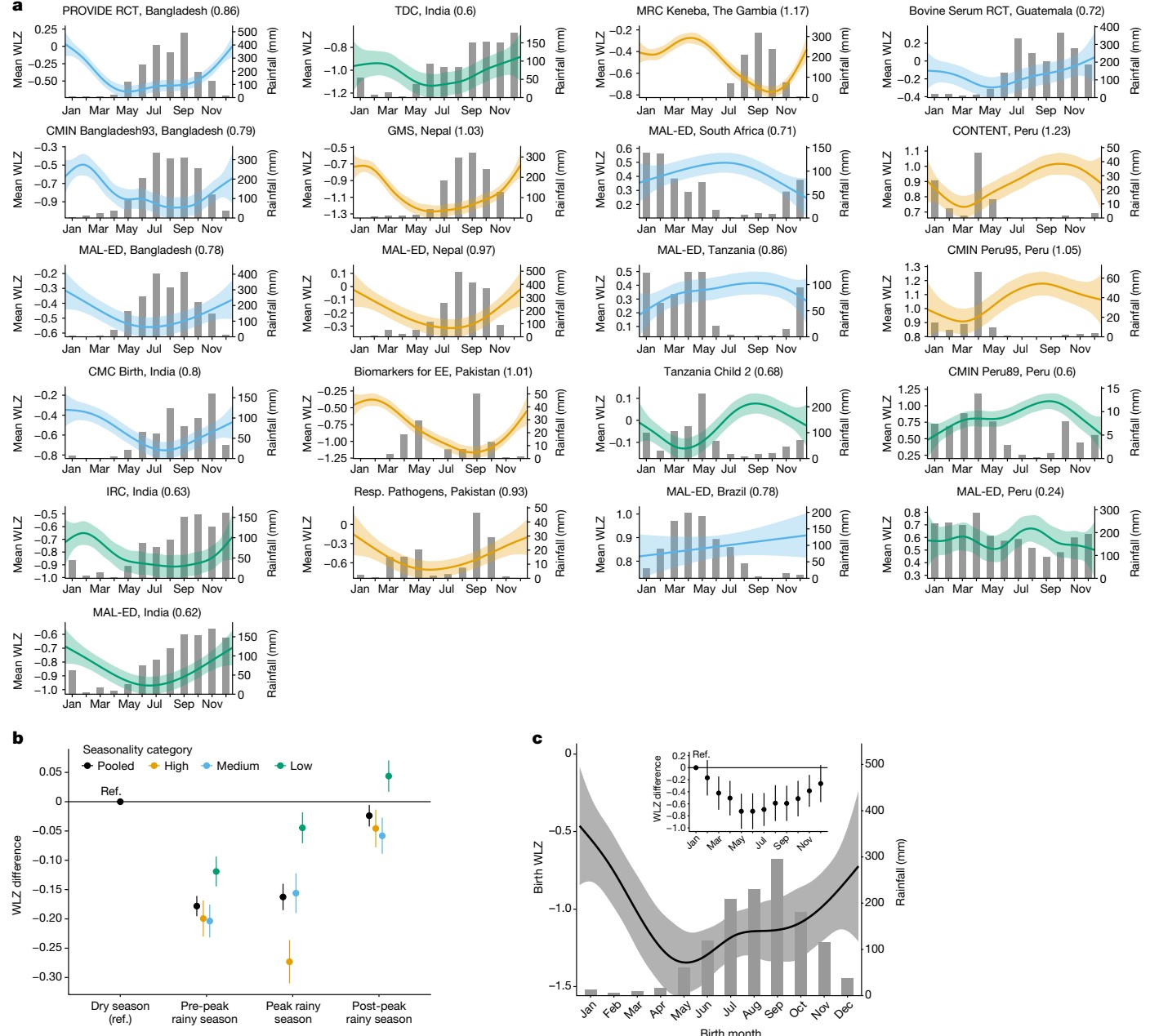

**Fig. 4 | Mean WLZ by age and season. a**, Mean WLZ by day of the year, superimposed over histograms of monthly mean rainfall over study periods, with the seasonality index included in parentheses beside the cohort name. The mean (lines) was estimated with cubic splines and the shaded regions indicate pointwise 95% CI. Panels are sorted by country and the mean WLZ is coloured by seasonality index categories: high (≥0.9), orange; medium (<0.9 and ≥0.7), blue; low (<0.7), green. Sample sizes range from 160 children (2,545 measurements) in TDC to 2,545 (40,115 measurements) in the Keneba cohort. **b**, Mean differences in child WLZ between quarters of the year defined around the adjacent three-month periods with the highest mean rainfall, pooled across cohorts in **a**, overall (*n* = 21 cohorts, 2,545–40,115 observations) and by seasonality index (high: *n* = 7 cohorts, 3,164–40,115 observations; medium:

*n* = 8 cohorts, 2,545–9,202 observations; low: *n* = 6 cohorts, 2,741–29,518 observations). Points represent mean differences compared to the reference level (Ref.) of the dry season and error bars represent 95% CI. The median *I*² statistic measuring heterogeneity in each meta-analysis was 91 (IQR = 87–94). **c**, Mean WLZ at birth by month of birth estimated with a cubic spline among 1,821 children with WLZ measured at birth in 10 South Asian cohorts. The mean (line) was estimated with cubic splines and the shaded region is its pointwise 95% CI. The inset plot summarizes mean differences in birth WLZ by month of birth (points), with January as the reference level (Ref.), and error bars represent 95% CI. The median *I*² statistic measuring heterogeneity in each meta-analysis was 18 (IQR = 0–41).

who were wasted during the rainy season during their first year of life were 1.9 times (95% CI: 1.3, 2.8) more likely to be wasted during the next rainy season.

South Asian cohorts had temporally synchronous rainfall, so we estimated the WLZ at birth by calendar month, pooled across the 11 South Asian cohorts. The mean WLZ at birth varied by up to

0.72 *z* (95% CI: 0.43, 1.02) depending on the month the child was born (range: −0.5 *z* to −1.3 *z*; Fig. 4c), which suggests that seasonally influenced maternal nutrition[38]—probably mediated through intrauterine growth restriction or premature birth[39]—was a major determinant of WLZ at birth. Birth month also influenced the effect of season on WLZ trajectories that persisted through a child's second year of life

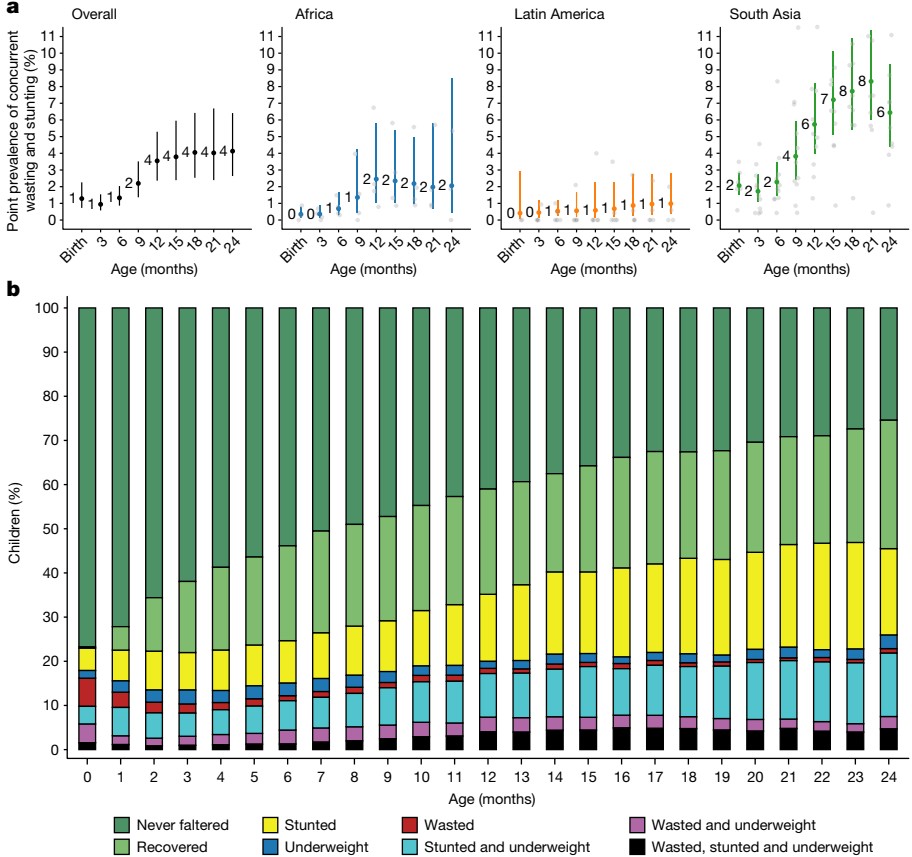

**Fig. 5 | Co-occurrence of wasting, stunting and being underweight.**
**a**, Age-specific prevalence of concurrent wasting and stunting overall
($n = 3,984–9,899$ children) and stratified by region (Africa: $n = 1,799–5,014$
children; Latin America: $n = 290–1,397$ children; South Asia: $n = 1,994–3,747$
children). Vertical lines mark 95% CI for pooled means across study-specific
estimates (grey points). The median $I^2$ statistic measuring heterogeneity in
each meta-analysis was 91 (IQR = 79–94) overall, 84 (IQR = 72–90) in Africa, 31
(IQR = 0–42) in Latin America and 86 (IQR = 73–86) in South Asia. **b**, Percentage
of children classified by different measures of growth faltering, alone or
combined. Children classified as 'never faltered' had not previously been
wasted, stunted or underweight. Children in the 'recovered' category were not
wasted, stunted or underweight but had experienced at least one of these
conditions previously. All children who were wasted and stunted were also
underweight. Proportions in each category were calculated within cohorts,
pooled using random effects and scaled so that the percentages added up to
100%. The number of children contributing to each age ranged from 3,920 to
9,077, with 11,409 children in total.

(Extended Data Fig. 11), and children born in the three-month periods
before or during maximum rainfall were more likely to be wasted
after 6 months of age than were children born in the 6 months after
maximum rainfall (cumulative incidence ratio: 1.4 (95% CI: 1.1, 1.9)).

## Persistent and concurrent growth faltering

We examined more severe forms of growth faltering, including persis-
tent wasting and concurrent wasting and stunting, because these con-
ditions are associated with a higher mortality risk[6]. We first identified
a subset of children who experienced persistent wasting during their
first 24 months. We used a pragmatic definition[40] that classified chil-
dren as persistently wasted if 50% or more of their WLZ measurements
from birth to 24 months were below −2. This allowed us to capture both
frequent short wasting episodes and less-frequent, longer episodes.
Among 10,374 children with at least 4 measurements, 3.4% (95% CI: 2.0,
5.6) were persistently wasted. Children in South Asia had the highest
proportion of measurements with wasting (Extended Data Fig. 5b) and
the highest prevalence of persistent wasting in the first 24 months of
life (7.2%; 95% CI: 5.1, 10.4; Extended Data Fig. 5c). Among all children
wasted at birth, 10.9% were persistently wasted after 6 months (95% CI:
8.7 to 13.5), whereas 6.3% of children who were not born wasted were
persistently wasted after 6 months (95% CI: 4.3, 9.0).

Finally, we examined the cumulative incidence of concurrent wasting
and stunting and the timing of their overlap. Overall, 10.6% of children
experienced concurrent wasting and stunting before 2 years of age
(Extended Data Fig. 5d), and a further 1.5% experienced concurrent
severe wasting and stunting—much higher than the estimates of point
prevalence in the present study (4.1% at 24 months, 95% CI: 2.6, 6.4;
Fig. 5a) or the 4.3% prevalence estimated in children under 5 years
old in LMICs from cross-sectional surveys[41,42]. Concurrent wasting
and stunting was most common in South Asia, with the highest preva-
lence at the age of 21 months (Fig. 5a), driven mainly by increases in
stunting prevalence as children aged[19]. Wasted and stunted children
were also underweight, as their maximum possible WAZ was below
−2.35 (ref. 43). Almost half of children who had prevalent growth fal-
tering met two or more of these three growth faltering conditions,
with 17–22% of all children experiencing multiple conditions after the
age of 12 months (Fig. 5b). Longitudinal analyses showed that early
growth faltering predisposed children to concurrent growth falter-
ing at older ages: children who were wasted by the age of 6 months
were 1.8 (95% CI: 1.6 to 2.1) times and children stunted before 6 months
were 2.9 (95% CI: 2.5 to 3.4) times more likely to experience concurrent
wasting and stunting between the ages of 18 months and 24 months.
A companion article reports an in-depth investigation of key risk
factors for persistent wasting and concurrent wasting and stunting,

along with their consequences for future severe growth faltering and mortality[32].

## Discussion

Nearly all large-scale studies of child wasting (including the DHS programme) report the point prevalence of wasting, which has enabled broad comparisons between populations but does not capture the number of episodes of wasting during the course of childhood[44,45]. By combining information across several longitudinal cohorts, this study shows that children have a much higher cumulative incidence of wasting between birth and 24 months (29.2%) than was previously known. Children in all included cohorts experienced a higher incidence compared with prevalence, owing to the episodic nature of acute malnutrition; this pattern was most stark in South Asian cohorts, in which, from birth to 24 months and pooled across cohorts, 52.2% of children experienced at least one wasting episode, 7.3% were persistently wasted and 10.1% had experienced concurrent wasting and stunting. This evidence shows that the cumulative, child-level burden of wasting is higher than cross-sectional measures of prevalence would suggest, and that assessments of wasting and stunting in isolation do not account for their joint burden, which can be substantial. This is of particular relevance in South Asia, where the largest number of stunted and wasted children live, and where, although the prevalence of stunting is decreasing, the prevalence of wasting is not[2,46].

Our results show consistent patterns in the timing of wasting onset by age and by season, with implications for therapeutic and preventive interventions. Wasting incidence was highest from birth to the age of 3 months, and among children who experienced any wasting during their first 24 months, 48% had their first episode by 3 months. Children born wasted were much more likely to experience more growth faltering at older ages, including more severe forms such as persistent wasting from 6 to 24 months and concurrent wasting and stunting at 18 months (Fig. 3f). The age-specific patterns in prevalent wasting and concurrent wasting and stunting summarized in this analysis are in accordance with previous reports[7,47–50]. To our knowledge, our finding that the wasting incidence was highest from birth to the age of 3 months has not been previously reported across diverse cohorts. We found higher rates of recovery among children who were wasted before 6 months, but early wasting episodes predisposed children to subsequent episodes at older ages, which tended to be longer and more likely to co-occur with stunting. These new findings suggest that the prenatal period through to 6 months old should be a focus for preventive wasting interventions, because preventing episodes during this early window would reduce the risk of later, more severe forms of growth faltering and lower a child's risk of all-cause mortality substantially, as we detail in a companion paper[32].

Large seasonal changes in the at-birth WLZ show a plasticity in population-level wasting whereby improvements are possible over a period of months rather than years or generations. The WLZ varied markedly by season in most cohorts, and in South Asia, the WLZ varied by up to 0.7 $z$ at birth, with consequences that persisted throughout the first 24 months (Fig. 4c and Extended Data Fig. 11). Seasonal effects might extend to older ages: children born during the rainy season in rural Gambia—coinciding with the hunger season—experienced higher rates of all-cause mortality over their entire lifespan compared with children born during other times of the year[51]. This suggests that in utero programming of the immune system could be a notable consequence of prenatal malnutrition. This study has not elucidated the mechanism that links seasonal rainfall with wasting, but the consistency of effect across diverse cultural and environmental contexts suggests shared mechanisms[52].

Consistent timing with highly seasonal rainfall helps in the planning of therapeutic programs, but seasonally targeted preventive interventions could benefit from a finer understanding of the mechanisms.

In-depth analyses of some birth cohorts that contributed to the present synthesis suggest that multiple mechanisms underlie seasonal wasting. Seasonal food insecurity coincides with peak energy demands during agricultural work periods, which, in turn, leads to undernutrition among pregnant and lactating mothers and their children[17,36,51]. Women working in agriculture often have less time to spend with their children, which can reduce the frequency of breastfeeding and can result in the early-morning preparation of weaning foods that become contaminated in hot, humid conditions[53,54]. Weather-driven seasonal increases in the transmission of infectious diseases could also have a role in seasonal wasting, but the effect would vary spatially and temporally and would not fully explain the consistent patterns observed here.

Sustainable Development Goal 2.2 calls for the elimination of malnutrition by 2030, with child wasting as its primary indicator[1]. The WHO global action plan on child wasting identified four key outcomes to achieve goal 2.2: reduce low birthweight; improve child health; improve child feeding; and improve the treatment of wasting[55]. Our results align with the WHO action plan but elevate the importance of improving at-birth child outcomes, with a focus on both maternal support during pregnancy and nutritional supplementation in food-insecure populations for women of child-bearing age, pregnant women and children under 24 months. At this time, evidence-based interventions that target this early window include balanced supplementation with protein, iron and folic acid for pregnant women and women of child-bearing age, and intermittent preventive administration of sulfadoxine–pyrimethamine to pregnant women in regions of Africa with high rates of malaria transmission[56,57]. For children aged 6–24 months, small quantities of lipid-based nutrient supplements can reduce the levels of moderate and severe wasting[10]. If preventive or therapeutic interventions focus on children younger than 6 months, then they must be integrated carefully with current recommendations for exclusive breastfeeding. In populations with highly seasonal rainfall, increased and seasonally targeted supplementation should be considered.

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

**The Ki Child Growth Consortium**

Tahmeed Ahmed[12], Asad Ali[13], France Begín[14], Pascal Obong Bessong[15], Zulfiqar A. Bhutta[16], Robert E. Black[17], Ladaporn Bodhidatta[18], William Checkley[17], Jean E. Crabtree[19], Rina Das[12], Subhasish Das[12], Christopher P. Duggan[20], Abu Syed Golam Faruque[12], Wafaie W. Fawzi[21], José Quirino da Silva Filho[22], Robert H. Gilman[17], Richard L. Guerrant[23], Rashidul Haque[12], Eric R. Houpt[23], Najeeha Talat Iqbal[13], Jacob John[24], Sushil Matthew John[25], Gagandeep Kang[26], Margaret Kosek[23], Aldo Ângelo Moreira Lima[22], Tjale Cloupas Mahopo[27], Dharma S. Manandhar[28], Karim P. Manji[29], Estomih Mduma[30], Venkata Raghava Mohan[24], Sophie E. Moore[31,32], Mzwakhe Emanuel Nyathi[33], Maribel Paredes Olortegui[34], William A. Petri[23], Prasanna Samuel Premkumar[24], Andrew M. Prentice[32], Najeeb Rahman[13], Kamran Sadiq[15], Rajiv Sarkar[24], Naomi M. Saville[35], Bhim P. Shrestha[36], Sanjaya Kumar Shrestha[37], Bakary Sonko[32], Erling Svensen[38], Sana Syed[13,23], Fayaz Umrani[13], Honorine D. Ward[39] & Pablo Penataro Yori[23]

[12]International Centre for Diarrhoeal Disease Research, Dhaka, Bangladesh. [13]Department of Pediatrics and Child Health, Aga Khan University, Karachi, Pakistan. [14]UNICEF, New York, NY, USA. [15]HIV/AIDS and Global Health Research Programme, University of Venda, Thohoyandou, South Africa. [16]Centre of Excellence in Women and Child Health, Institute for Global Health and Development, The Aga Khan University, Karachi, Pakistan. [17]Johns Hopkins University Bloomberg School of Public Health, Baltimore, MD, USA. [18]Armed Forces Research Institute of Medical Sciences, Bangkok, Thailand. [19]Leeds Institute for Medical Research, St James's University Hospital, University of Leeds, Leeds, UK. [20]Center for Nutrition, Boston Children's

Hospital, Boston, MA, USA. [21]Department of Global Health and Population, Harvard T.H. Chan School of Public Health, Boston, MA, USA. [22]Federal University of Ceará, Fortaleza, Brazil. [23]University of Virginia, Charlottesville, VA, USA. [24]Christian Medical College, Vellore, India. [25]Low Cost Effective Care Unit, Christian Medical College, Vellore, India. [26]Translational Health Science and Technology Institute, Faridabad, India. [27]Department of Nutrition, School of Health Sciences, University of Venda, Thohoyandou, South Africa. [28]Mother and Infant Research Activities, Kathmandu, Nepal. [29]Department of Pediatrics and Child Health, Muhimbili University School of Health and Allied Sciences, Dar es Salaam, Tanzania. [30]Haydom Lutheran Hospital, Haydom, Tanzania. [31]Department of Women and Children's Health, Kings College London, London, UK. [32]MRC Unit The Gambia at London School of Hygiene and Tropical Medicine, Banjul, The Gambia. [33]Department of Animal Sciences, School of Agriculture, University of Venda, Thohoyandou, South Africa. [34]AB PRISMA, Lima, Peru. [35]Institute for Global Health, University College London, London, UK. [36]Health Research and Development Forum, Kathmandu, Nepal. [37]Walter Reed AFRIMS Research Unit, Kathmandu, Nepal. [38]Haukeland University Hospital, Bergen, Norway. [39]Tufts Medical Center, Tufts University School of Medicine, Boston, MA, USA.

## Methods

### Study designs and inclusion criteria

We included all longitudinal observational studies and randomized trials available through the Ki project on April 2018 that met five inclusion criteria (Extended Data Fig. 1): (1) they were conducted in LMICs; (2) they enrolled children between birth and the age of 24 months and measured their length and weight repeatedly over time; (3) they did not restrict enrolment to acutely ill children; (4) they enrolled at least 200 children; (5) they collected anthropometry measurements at least monthly. The frequency of measurements was assessed by calculating the median days between measurements. Our pre-specified analysis protocol stipulated that if randomized trials found effects of interventions on growth, the analysis would only include the control arm; however, all interventional trials that met the inclusion criteria had null effects on growth, so all arms were included. We included all children under the age of 24 months, assuming months were 30.4167 days. We excluded extreme measurements of WLZ > 5 or WLZ < −5 and of WAZ < −6 or WAZ > 5, consistent with the recommendations of the 2006 WHO growth standards[26]. We checked the cohort-wide quality of anthropometry measurement by plotting weight and length against growth standard distribution (Supplementary Note 1) and by calculating the proportion of length measurements in which length decreased beyond the expected technical error of measurement compared to the last measurement on a child. One cohort, MAL-ED Pakistan, was excluded because measurements exhibited a multimodal WLZ distribution with scores binned at −2, −1.5 and −1 instead of continuously distributed. Some young infants with weight and height measured were missing WLZ, as WLZ cannot be calculated when the length of a child is less than 45 cm (0.8% of measurements before the age of 1 month, with a mean WAZ for those children of −2.9 z, and 10.2% of children measured at birth), so using WLZ underestimates wasting at birth[58].

### Outcome definitions

We used the following outcome measures in the analysis:

WLZs were calculated using the 2006 WHO growth standards[25], and mean WLZ was calculated within strata of interest. We used the medians of triplicate measurements of lengths and weights of children from pre-2006 cohorts to re-calculate z-scores to the 2006 standard.

Prevalent wasting was defined as the proportion of measurements within a specific stratum (for example, age) below the 2006 WHO standard −2 WLZ. Similarly, prevalent severe wasting was defined as the proportion of measurements below the 2006 WHO standard −3 WLZ. For each age, we included children with WLZ measurements within one month before and after that age in the point-prevalence estimate (for example, for point prevalence at 6 months, we include children aged 5–7 months).

Incident wasting episodes were defined as a change in WLZ from above −2 z in the prior measurement to below −2 z in the current measurement. Similarly, we defined severe wasting episodes using a cut-off of −3 z. We assumed a 60-day washout period between episodes of wasting before a new episode of wasting could occur (Fig. 4a). Children were considered at risk for wasting at birth, so children born wasted were considered to have an incident episode of wasting at birth. Children were also assumed to be at risk of wasting at the first measurement in studies that enrolled children after birth; thus, children wasted at the first measurement in a non-birth cohort were assumed to have incident wasting occurring at the age halfway between birth and the first measurement.

Incident proportion of wasting was calculated during a defined age range (for example, 6–12 months) as the proportion of children not wasted at the start of the period who became wasted during the age period (the proportion of children who had the onset of new episodes during the period). This differs from period prevalence, which would include any children who had any measurement of WLZ < −2 during the period (including children who began the period wasted). Period prevalence would thus also include, in the numerator and denominator, children who were wasted at the start of the period, whereas incidence proportion excludes them.

Recovery from wasting was defined as a change in WLZ from below to above −2 z among children who are currently wasted or severely wasted. We required a child to maintain a WLZ above −2 for 60 days to be considered 'recovered'. Children were only considered 'at risk' for recovery if their prior measurement was below −2 z. We measured the proportion of children who recovered from moderate wasting (WLZ < −2) within 30, 60 and 90 days of the onset of the episode. We assumed that recovery from wasting was spontaneous because we did not have consistent information on referral guidelines across cohorts in the analysis, but some children with moderate wasting might have been referred to clinical facilities for outpatient treatment. Wasting recovery might thus be higher in these highly monitored cohorts than in the general population owing to treatment referrals.

Wasting duration was estimated by counting the days between the onset of wasting and recovery within an individual child. We assumed that the episode started or ended at the midpoint between measurements. For example, if a child was not wasted at age 40 days, wasted at age 70 days and not wasted at age 100 days, the duration of the wasting episode was: (70 − 40)/2 + (100 − 70)/2 = 30 days. We calculated the median duration of wasting episodes among cohorts in which children recovered during the study period from more than 50% of observed wasting episodes, as the duration of episodes that extended beyond the study period is unknown. Therefore, the median duration could only be calculated when less than half of episodes were censored.

The incidence rate of wasting was calculated during defined age ranges as the number of incident episodes of wasting per 1,000 child-days at risk during the age range. Children were considered 'at risk' for incident wasting episodes if they were not currently wasted and were classified as recovered from any previous wasting episode (beyond the 60-day washout period). Therefore, wasted children, or children within the washout period, did not contribute to the person-time at risk for wasting used to calculate wasting incidence. To calculate the person-time at risk of wasting, we assumed that the onset of a wasting episode occurred when the child's age was at the midpoint between the measurement of WLZ ≥ −2 and the measurement of WLZ < −2, and conversely the time of recovery occurred when the child's age was at the midpoint between the last measurement of WLZ < −2 and the first measurement of WLZ ≥ −2 within the 60-day washout period (Fig. 4a).

Persistent wasting was defined as a greater than 50% longitudinal prevalence of below −2 WLZ (wasting), and analogously a greater than 50% longitudinal prevalence of below −3 WLZ for persistent severe wasting[40].

Concurrent prevalence of wasting and stunting was defined as the proportion of measurements at a specific age when a child was both wasted and stunted at the same measurement. For each age, we included children with WLZ and LAZ measurements within one month before and after that age in the point-prevalence estimate (for example, for point prevalence at 6 months, we include children aged 5–7 months).

Prevalent underweight was defined as the proportion of measurements at a specific age below the 2006 WHO standard −2 WAZ. For each age, we included children with WAZ measurements within one month before and after that age in the point-prevalence estimate (for example, for point prevalence at 6 months, we include children aged 5–7 months).

**Subgroups of interest.** We stratified the above outcomes of interest within the following subgroups: child age (grouped into one-, three- or six-month intervals, depending on the outcome); region of the world (Asia, sub-Saharan Africa and Latin America); month of the year; and the combinations of the above categories. We also pooled wasting-incidence proportion by categories of three country-level characteristics: the percentage of gross domestic product devoted

to healthcare goods and spending (obtained from the United Nations Development Programme; https://hdr.undp.org/); the percentage of the country living on less than US$1.90 per day; and under-5 mortality rates (both obtained from the World Bank; https://databank.worldbank.org/source/world-development-indicators). Within each country, in years without available data, we linearly interpolated values from the nearest years with available data and extrapolated values within five years of available data using linear regression models based on all available years of data.

## Statistical analysis
All analyses were conducted in R v.4.0.5.

**Fixed- and random-effects models.** We conducted a two-stage individual participant meta-analysis, estimating wasting outcomes within specific cohorts and pooling estimates within each age stratum using random-effects models. For example, we estimated each age-specific mean using a two-step process. We first estimated the mean in each cohort, and then pooled age-specific means across cohorts allowing for a cohort-level random effect. We estimated overall pooled effects, and pooled estimates specific to South Asian, African or Latin American cohorts, depending on the analysis. We repeated the pooling of all statistics presented in the figures using fixed-effects models as a sensitivity analysis (Supplementary Note 2). The pooling methods are described in greater detail in a companion paper[19]. All pooling except for episode duration was completed using the rma() function from the metaphor package in the R language (v.3.0-2)[59]. Median durations were pooled across cohorts using the unweighted median of medians method[60].

**Fitted spline curves.** Fitted smoothers used in the figures were fitted using cubic splines and generalized cross-validation[61]. We estimated approximate simultaneous 95% CIs around the cubic splines using a parametric bootstrap that resampled from the posterior the generalized additive model parameter variance–covariance matrix[62].

**Seasonality analysis.** We compared the mean WLZ over day of the year, child birthday and seasons defined by rainfall. We estimated mean WLZ by cohort over day of the year and child birthday using cubic splines[63] (Fig. 3a,c). Splines of WLZ over day of the year were plotted over monthly mean rainfall averaged over the years in which a study measured child anthropometry below the age of 24 months (Fig. 3a). We pulled monthly precipitation values from TerrraClimate, a dataset that combines readings from WorldClim data, CRU Ts4.0 and the Japanese 55-year Reanalysis Project[64]. For each study region, we averaged all readings within a 50-km radius from the study coordinates. If GPS locations were not in the data for a cohort, we used the approximate location of the cohort on the basis of the published descriptions of the cohort. Monthly measurements were matched to study data on the basis of the calendar month and year in which measurements were taken.

The season of peak rainfall was defined as the three-month period with the highest mean rainfall. Mean differences in WLZ between three-month quarters was estimated using linear regression models. We compared the consecutive three months of the maximum average rainfall over the study period, as well as the three months prior and three months after the maximum-rainfall period, to a reference level of the three months opposite the calendar year of the maximum-rainfall period. We used all WLZ measurements of children under two years of age (for example, if June–August was the period of maximum rainfall, the reference level is child mean WLZ during January–March).

Estimates were unadjusted for other covariates because we assumed that seasonal effects on WLZ were exogenous and could not be confounded. Mean differences in WLZ were pooled across cohorts using random-effects models, with cohorts grouped by the Walsh and Lawler seasonality index[65]. Cohorts from years with a seasonal index greater than or equal to 0.9 were classified as occurring in locations with high seasonality; cohorts with a seasonal index less than 0.9 and greater than or equal to 0.7 were classified as occurring in locations with medium seasonality; and cohorts with a seasonal index less than 0.7 were classified as occurring in locations with low seasonality.

$$\text{Seasonal index} = \frac{1}{R} \sum_{n=1}^{n=12} \left| X_n - \frac{R}{12} \right|,$$

where $R$ = total annual precipitation and $X_n$ = monthly precipitation.

**Estimation of mean LAZ, WAZ and WLZ by age in DHSs and Ki cohorts.** We downloaded standard DHS individual recode files for each country from the DHS program website (https://dhsprogram.com/). We used the most recent standard DHS datasets for the individual women's, household and length and weight datasets from each country, and we estimated age-stratified mean LAZ, WAZ and WLZ from ages 0 to 24 months within each DHS survey, accounting for the complex survey design and sampling weights. See a companion paper for further details on the DHS data cleaning and analysis[19]. We compared DHS estimates with mean LAZ, WAZ and WLZ by age in the Ki study cohorts with penalized cubic splines with bandwidths chosen using generalized cross-validation[63]. We did not seasonally adjust DHS measurements.

**Sensitivity analyses.** We estimated incidence rates of wasting after excluding children born or enrolled wasted (Fig. 4b). The rationale for this sensitivity analysis is that incident cases at birth imply a different type of intervention (that is, prenatal) compared with postnatal onset of wasting. We estimated the overall and region-stratified prevalence of persistent wasting and of being underweight and severe wasting by age (Extended Data Fig. 5). Within cohorts that measured child gestational age at birth, we estimated the prevalence of stunting and being underweight at birth both uncorrected and corrected for gestational age at birth using the Intergrowth standards[66] (Extended Data Fig. 7). We also assessed how much change in z-scores after children became wasted could be explained by RTM and how much is catch-up growth beyond that expected by RTM (ref. 67). We calculated the cohort-specific RTM effect and plotted the mean z-scores expected from RTM 3 months later among children who were wasted at each age in months from birth to 21 months. We used the cohort means as the population mean z-scores, and compared the expected mean WLZ with the observed mean WLZ[34,68] (Extended Data Fig. 8). We also compared estimates pooled using random-effects models, which are more conservative in the presence of study heterogeneity, with estimates pooled using fixed-effects (inverse-variance-weighted) models (Supplementary Note 2). We also compared wasting prevalence defined using middle-upper-arm circumference (MUAC), an alternative measurement for classifying wasting, with wasting prevalence estimated using WLZ within the cohort that measured MUAC (Supplementary Note 3). We also re-estimated primary results dropping observations of children at birth within the MRC Keneba cohort, which used a different team to measure child anthropometry at birth from the trained anthropometrists used in follow-up measurements (Supplementary Note 4). We also examined the effect of shorter (30-day) and longer (90-day) washout periods when determining whether a child was again at risk when estimating wasting incidence and wasting recovery rates (Supplementary Note 5). Finally, we also repeated prevalence and incidence estimates for being underweight (Supplementary Note 6) and severe wasting (Supplementary Note 7) and plotted cohort-specific wasting estimates (Supplementary Note 8).

## Inclusion and ethics
This study analysed data, collected in 10 LMICs, that were assembled by the Bill & Melinda Gates Foundation Knowledge Integration (Ki) initiative. Datasets are owned by the original investigators that collected the data. Members of the Ki Child Growth Consortium were nominated by

each study's leadership team to be representative of the country and study teams that originally collected the data. Consortium members reviewed their cohort's data within the Ki database to ensure external and internal consistency of cohort-level estimates. Consortium members provided input on the statistical analysis plan, interpretation of results and manuscript writing. Per the request of consortium members, the manuscript includes cohort-level and regional results to maximize the utility of the study findings for local investigators and public health agencies. Analysis code has been published with the manuscript to promote transparency and extensions of our research by local and global investigators.

## Reporting summary

Further information on research design is available in the Nature Portfolio Reporting Summary linked to this article.

## Data availability

The data that support the findings of this analysis are a combination of data from multiple principal investigators and institutions. The data are available upon reasonable request by contacting these individual principal investigators. The following link lists the individuals and provides their contact information to help the requestor get access to the data: https://www.synapse.org/#!Synapse:syn51570682/wiki/. For data requestors, the first link can be used to request data from each principal investigator. The analysis dataset is at https://www.synapse.org/#!Synapse:syn51570682/datasets/. This dataset is access controlled and not available publicly for privacy reasons.

## Code availability

Replication scripts for this analysis are available at https://zenodo.org/record/7937811.

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

**Acknowledgements** This research was financially supported by a global development grant (OPP1165144) from the Bill & Melinda Gates Foundation to the University of California, Berkeley. J.B.-C. acknowledges funding from the National Institute of Allergy and Infectious Diseases under award K01AI141616. J.B.-C. is a Chan Zuckerberg Biohub investigator. We would also like to thank the following collaborators on the included cohorts and trials for their contributions to study planning, data collection and analysis: M. Sharif, S. Kerio, Urosa, Alveen, S. Hussain, V. Paudel, A. Costello, B. Torun, L. M. Locks, C. M. McDonald, R. Kupka, R. J. Bosch, R. Kisenge, S. Aboud, M. Wang and all other members of the study staff and field teams. We also thank all study participants and their families for their contributions.

**Author contributions** Conceptualization: A.M., J.B.-C., J.M.C., K.H.B., P.C. and B.F.A. Funding acquisition: J.M.C., A.E.H., M.J.v.d.L. and B.F.A. Data curation: A.M., J.B.-C., J.C., O.S., W. Cai, A.N., N.N.P., W.J., E.J., E.O.C., S.R., N.H., I. Malenica, H.L., R. Hafen, V.S., J.H. and T.N. Formal analyses: A.M., J.B.-C., J.C., O.S., W. Cai, A.N., N.N.P., W.J., E.J., E.O.C., S.R., S.D., N.H., I. Malenica, H.L., V.S. and B.F.A. Methodology: A.M., J.B.-C., J.M.C, J.C., O.S., N.H., I. Malenica, A.E.H., M.J.v.d.L., K.H.B., P.C. and B.F.A. Visualization: A.M., J.B.-C., A.N., N.N.P., S.R., A.S., J.C., R. Hafen, S.D., E.J., K.H.B., P.C. and B.F.A. Writing (original draft preparation): A.M., J.B.-C. and B.F.A. Writing (review and editing): A.M., J.B.-C., J.M.C., K.H.B., P.C., B.F.A. and members of the Ki Child Growth Consortium.

**Competing interests** T.N. is an employee of the Bill & Melinda Gates Foundation. K.H.B. and P.C. are former employees of the Bill & Melinda Gates Foundation. J.C., V.S., R. Hafen and J.H. work as research contractors funded by the Bill & Melinda Gates Foundation.

**Additional information**
**Correspondence and requests for materials** should be addressed to Andrew Mertens or Benjamin F. Arnold.

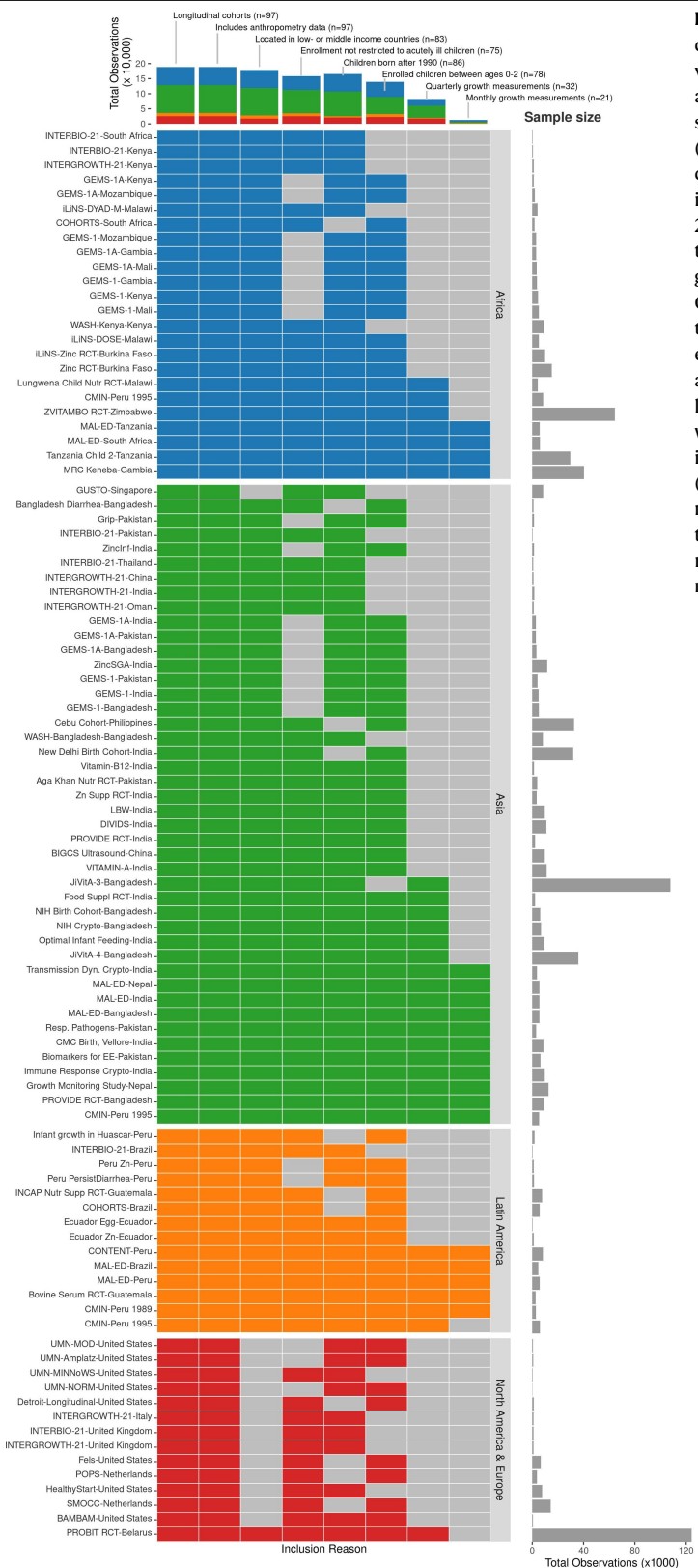

**Extended Data Fig. 1 | Ki cohort selection.** Analyses focused on longitudinal cohorts to enable the estimation of prospective incidence rates and growth velocity. On 15 July 2018, there were 97 longitudinal studies with data collected and made available by Ki. From this set, we applied five inclusion criteria to select cohorts for analysis. Our rationale for each criterion is as follows.
(1) Studies were conducted in LMICs. Our target of inference for analyses was children in LMICs, which remains a key target population for preventive interventions. (2) Studies measured length and weight between birth and age 24 months. We were principally interested in growth faltering during the first two years of life including at birth, thought to be the key window for linear growth faltering. (3) Studies did not restrict enrolment to acutely ill children. Our focus on descriptive analyses led us to target, to the extent possible, the general population. We thus excluded some studies that exclusively enrolled acutely ill children, such as children who presented to hospital with acute diarrhoea or who were severely malnourished. (4) Studies enrolled at least 200 children. Age-stratified incident episodes of stunting and wasting were sufficiently rare that we wanted to ensure each cohort would have enough information to estimate rates before contributing to pooled estimates.
(5) Studies collected anthropometry measurements at least every three months. We limited studies to those with higher temporal resolution to ensure that we adequately captured incident episodes and recovery. We further restricted analyses of wasting incidence and recovery to cohorts with monthly measurements because of high temporal variation in WHZ within individuals.

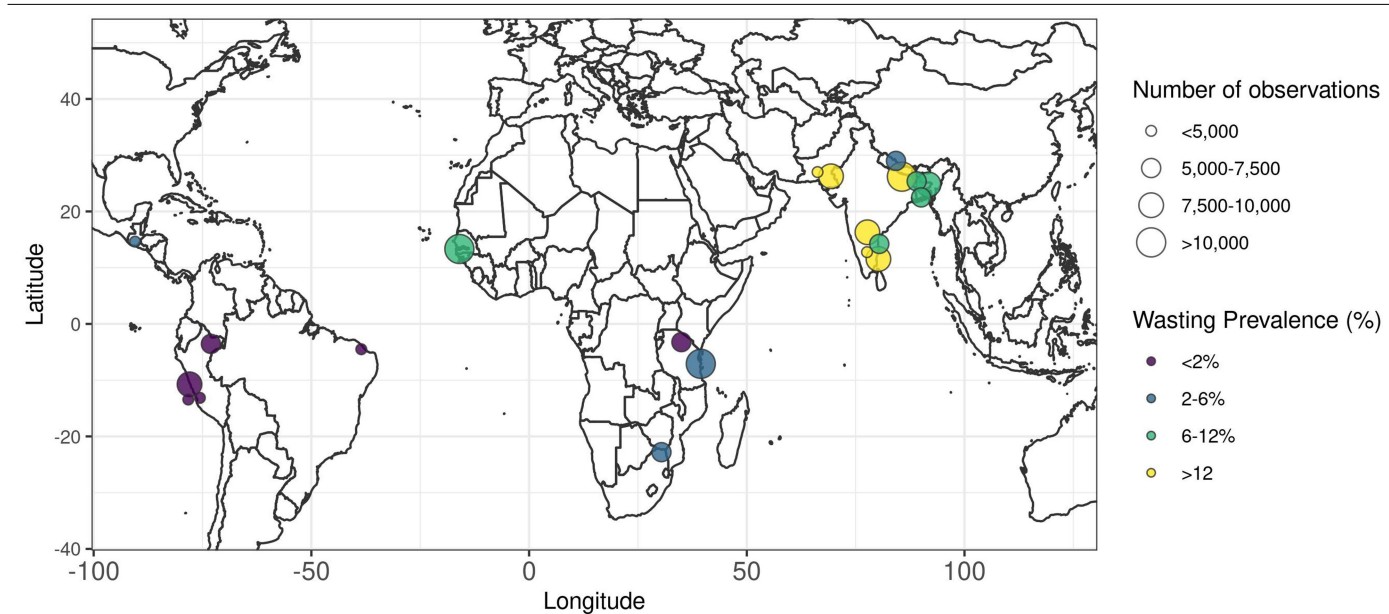

**Extended Data Fig. 2 | Geographical location of Ki cohorts.** Locations are approximate and jittered slightly for display.

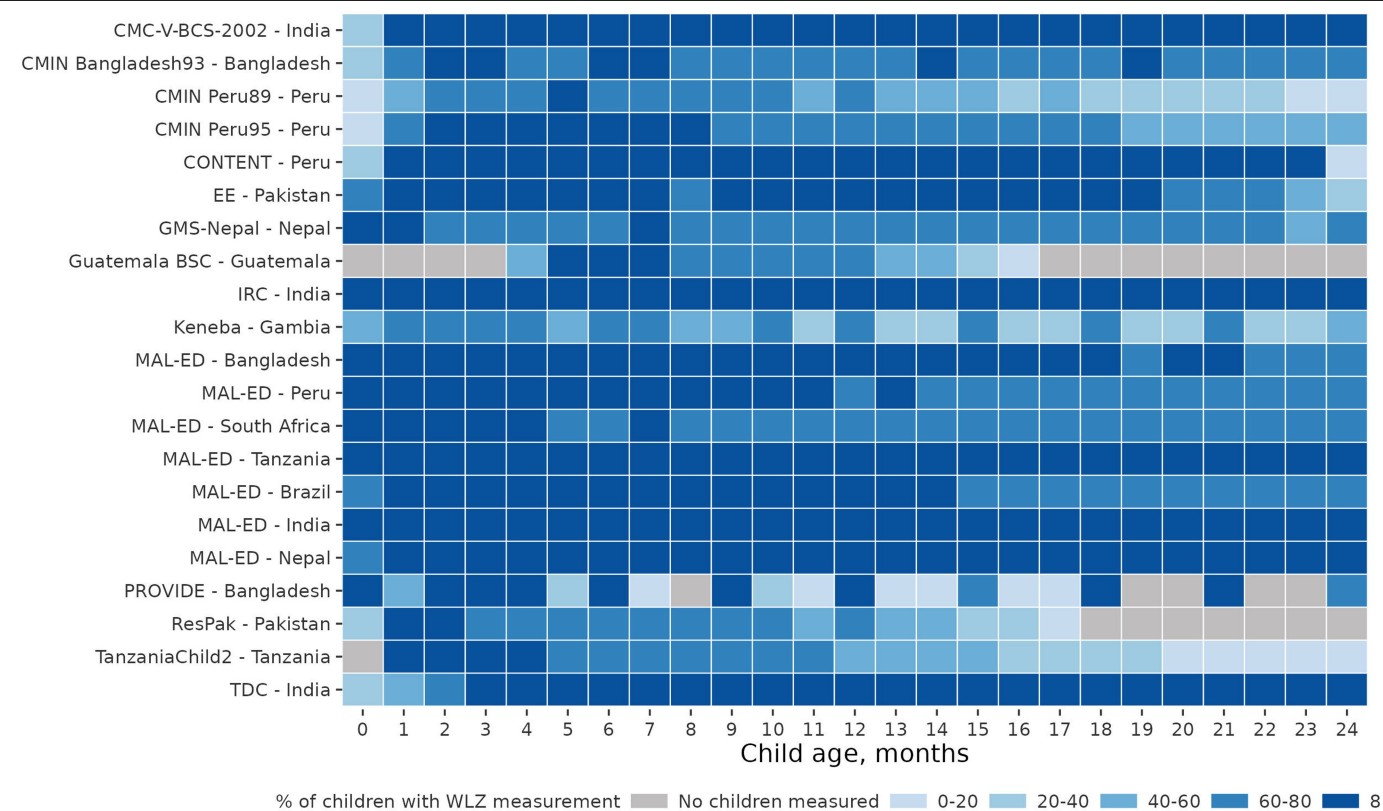

**Extended Data Fig. 3 | Percentage of enrolled children measured in each Ki cohort with monthly measurements.** Each coloured cell indicates the percentage of children with a WLZ measurement for a given cohort at a particular child age. Grey cells indicate that no children had a WLZ measurement for that age.

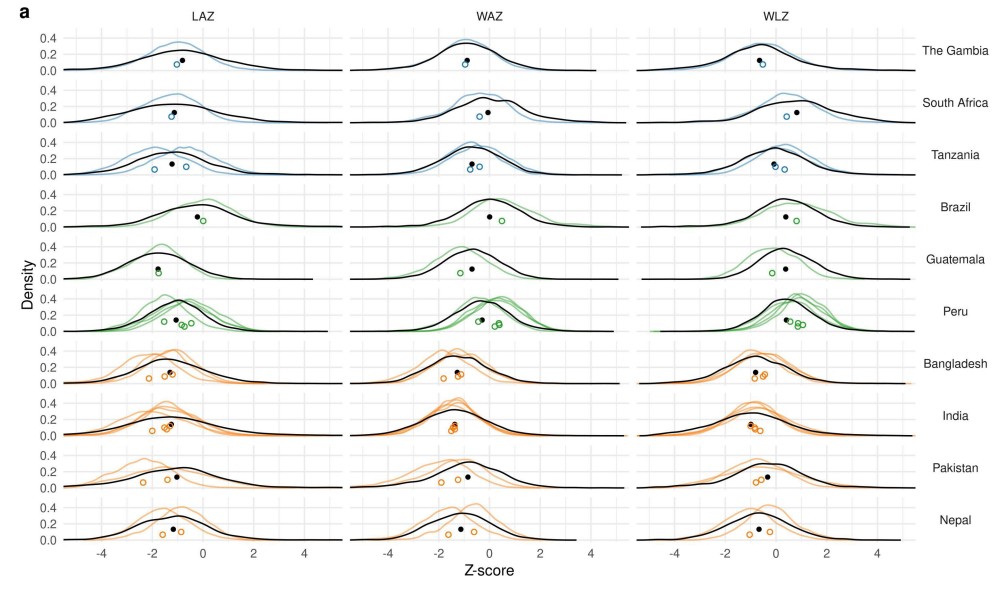

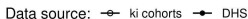

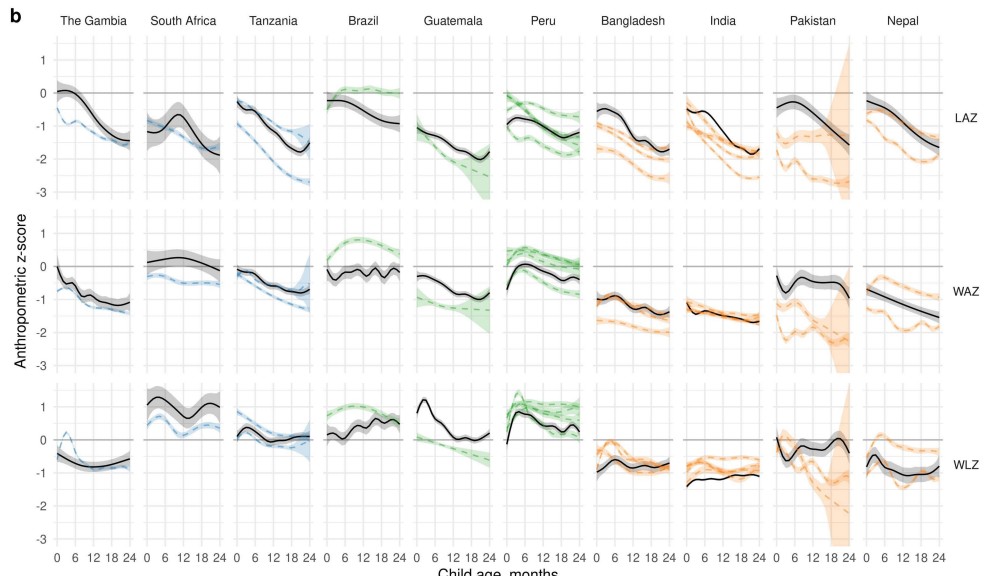

**Extended Data Fig. 4 | Comparison of cohort anthropometry to population-based samples. a**, Kernel density distributions of LAZ, WAZ and WLZ from measurements among children under 24 months old in 21 Ki longitudinal cohorts (coloured line) and among children measured in the most recent population-based DHS for each country (black). Sub-Saharan African countries are coloured blue, Latin American countries are coloured green and south Asian countries are coloured orange. Median z-scores are denoted with points under the density curves, with open circles for Ki cohorts and solid points for DHSs. **b**, Mean LAZ, WAZ and WLZ by age and country among 21 Ki longitudinal cohorts (dashed lines) and in DHSs (solid). Means estimated with cubic splines and shaded regions show approximate, simultaneous 95% CI. Each panel includes n = 117,664 children with LAZ, 117,783 children with WAZ, and 117,619 children with WHZ measured from DHS data and n = 11,442 children with LAZ, 11,443 children with WAZ, 11,407 children with WLZ measured from Ki cohorts.

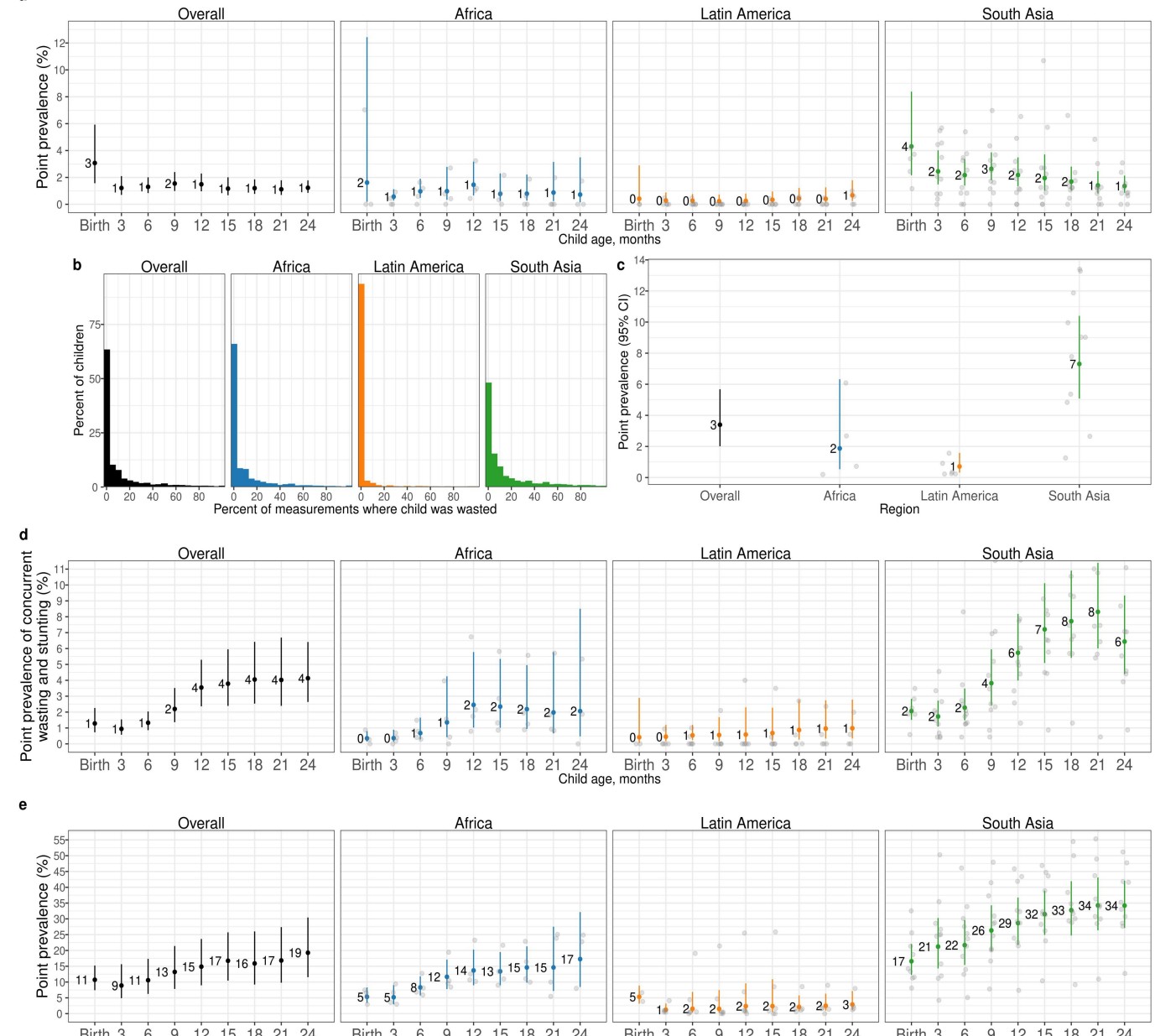

**Extended Data Fig. 5 | Prevalence of persistent wasting, severe wasting and being underweight, wasting measurement frequency, and concurrent wasting and stunting by region. a**, Prevalence of severe wasting (WLZ < −3) by age and region. **b**, Histograms of the proportions of visits at which children have wasted measurements, overall and stratified by region (20 bins at 5% bin-width). **c**, Proportion of children persistently wasted (≥50% of measurements from birth to 24 months of age, overall and stratified by region. **d**, Incidence proportion of concurrent wasting and stunting by age and region. Across all ages before 24 months, 10.6% of children experienced at least one concurrent wasted and stunted measurement. **e**, Prevalence of being underweight (WAZ < −2) by age and region.

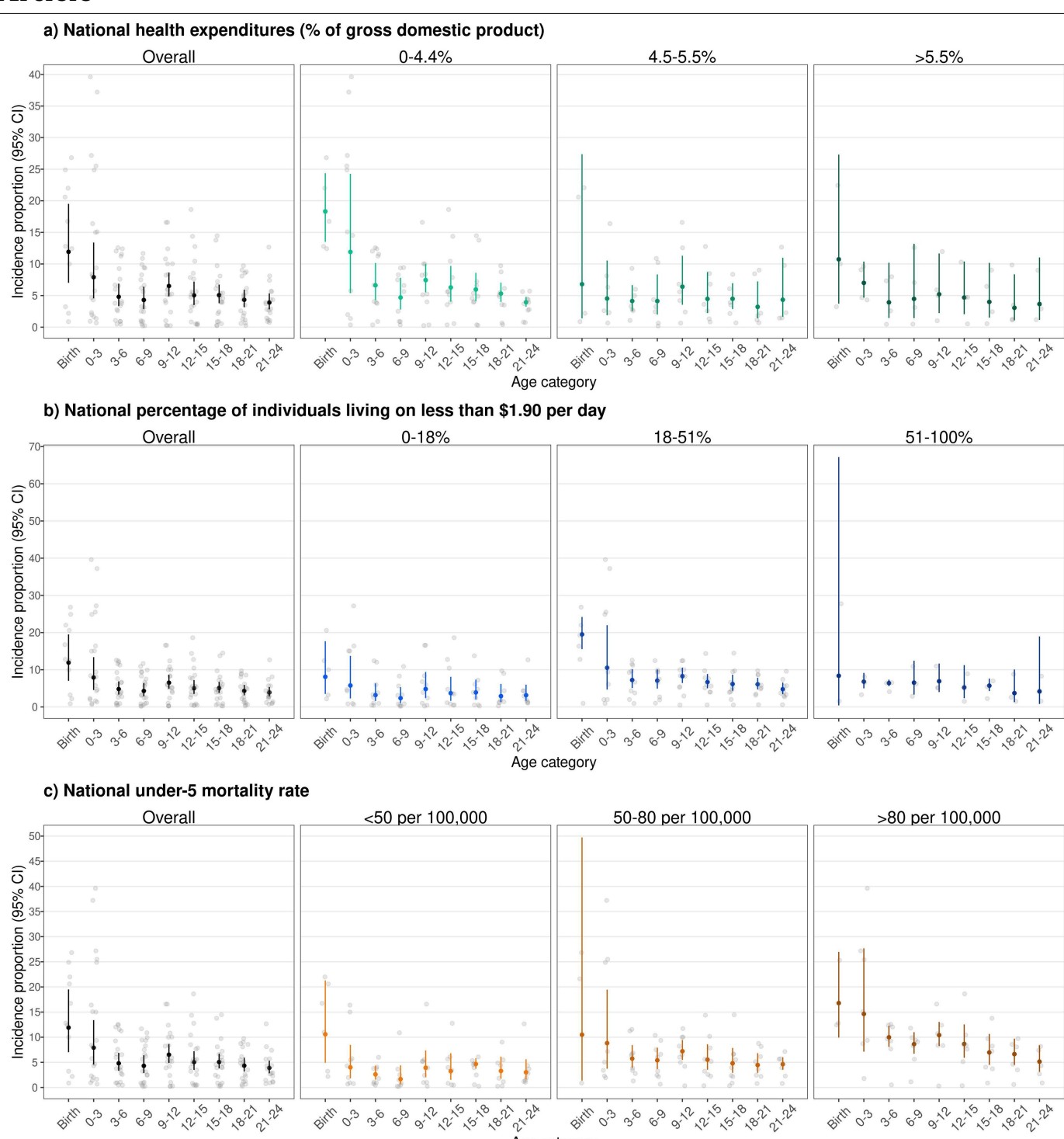

**Extended Data Fig. 6 | Incidence of wasting by age and country characteristics.** Proportion of children at different ages in months who experienced the onset of wasting episodes (**a**) by national health expenditures as a percentage of gross domestic product (0–4.4%: $n$ = 5 studies, $n$ = 3,305 children; 4.5–5.5%: $n$ = 5 studies, $n$ = 2,697 children; >5.5%: $n$ = 3 studies, $n$ = 3,325 children); (**b**) by national percentage of individuals living on less than $1.90 US per day (0–18%: $n$ = 5 studies, $n$ = 2,818 children; 18–51%: $n$ = 6 studies, $n$ = 3,584 children; 51–100%: $n$ = 2 studies, $n$ = 3,381 children); (**c**) and by national under-5 mortality rate (<50 per 100,000: $n$ = 6 studies, $n$ = 2,743 children; 50–80 per 100,000: $n$ = 3 studies, $n$ = 3,952 children; >80 per 100,000: $n$ = 3 studies, $n$ = 3,611 children). The 'Birth' age category includes measurements in the first 7 days of life and the '0–3' age category includes ages from 8 days up to 3 months. Vertical bars indicate 95% CI from random-effects meta-analysis models with restricted maximum likelihood estimation, and grey points indicate cohort-specific estimates.

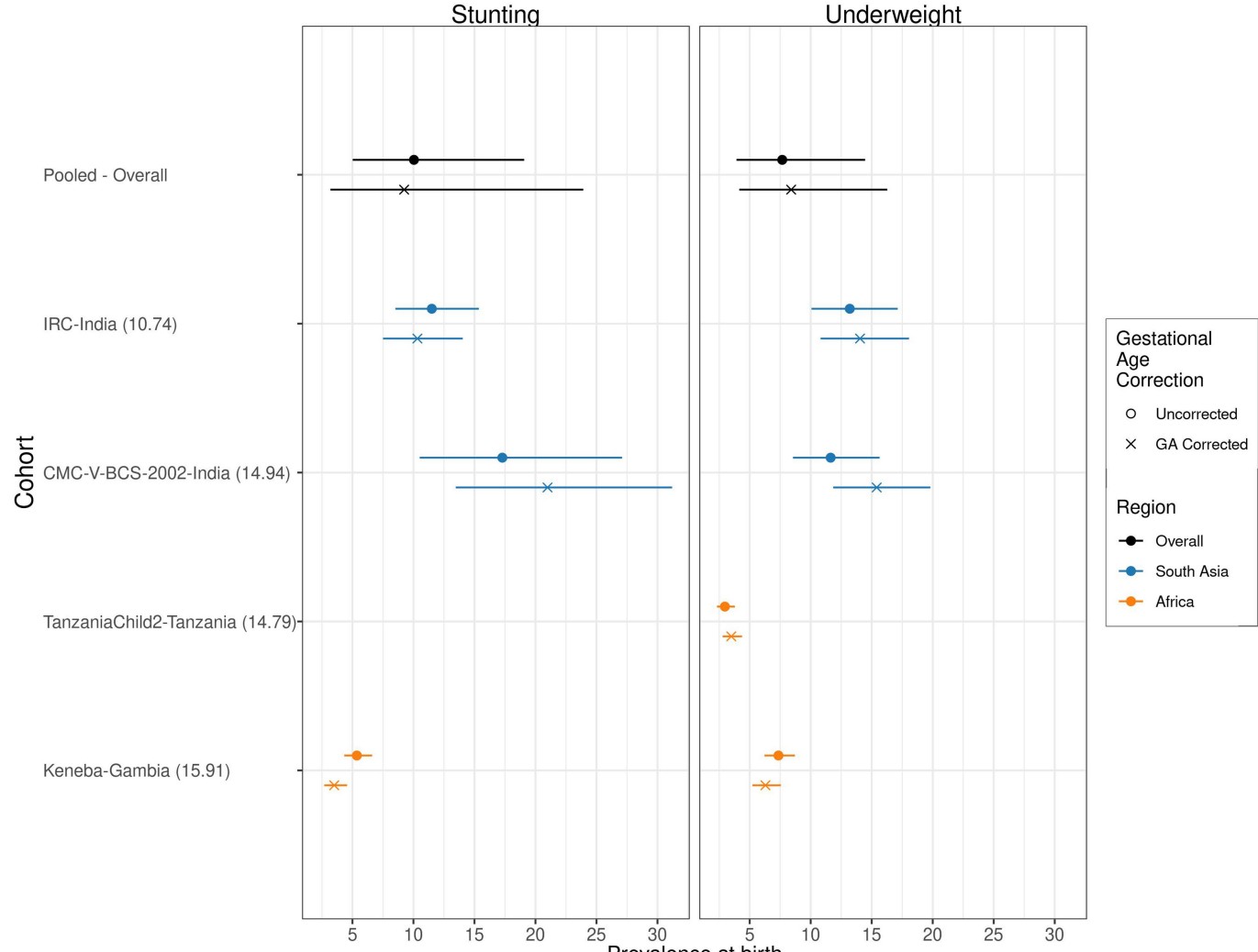

**Extended Data Fig. 7 | Comparison of the prevalence of being underweight and stunting at birth, with and without correction for gestational age.** This figure includes the results from correcting at-birth *z*-scores in the Ki cohorts that measured gestational age (GA). The corrections are using the Intergrowth standards and are implemented using the R growthstandards package (https://ki-tools.github.io/growthstandards/). Overall, the prevalence at birth decreased slightly after correcting for gestational age, but the cohort-specific results are inconsistent. Observations with GA outside of the Intergrowth standards range (<168 or >300 days) were dropped for both the corrected and uncorrected data. Prevalence increased after GA correction in some cohorts owing to high rates of late-term births based on reported GA. There were no length measurements at birth in the Tanzania Child 2 cohort, so they do not have stunting estimates. There were 4,449 measurements used in the underweight analysis and 1,931 measurements used in the stunting analysis. Gestational age was estimates based on mother's recall of the last menstrual period in the IRC, CMC-V-BCS-2002 and Tanzania Child cohorts, and was based on the Dubowitz method (newborn exam) in the MRC Keneba cohort.

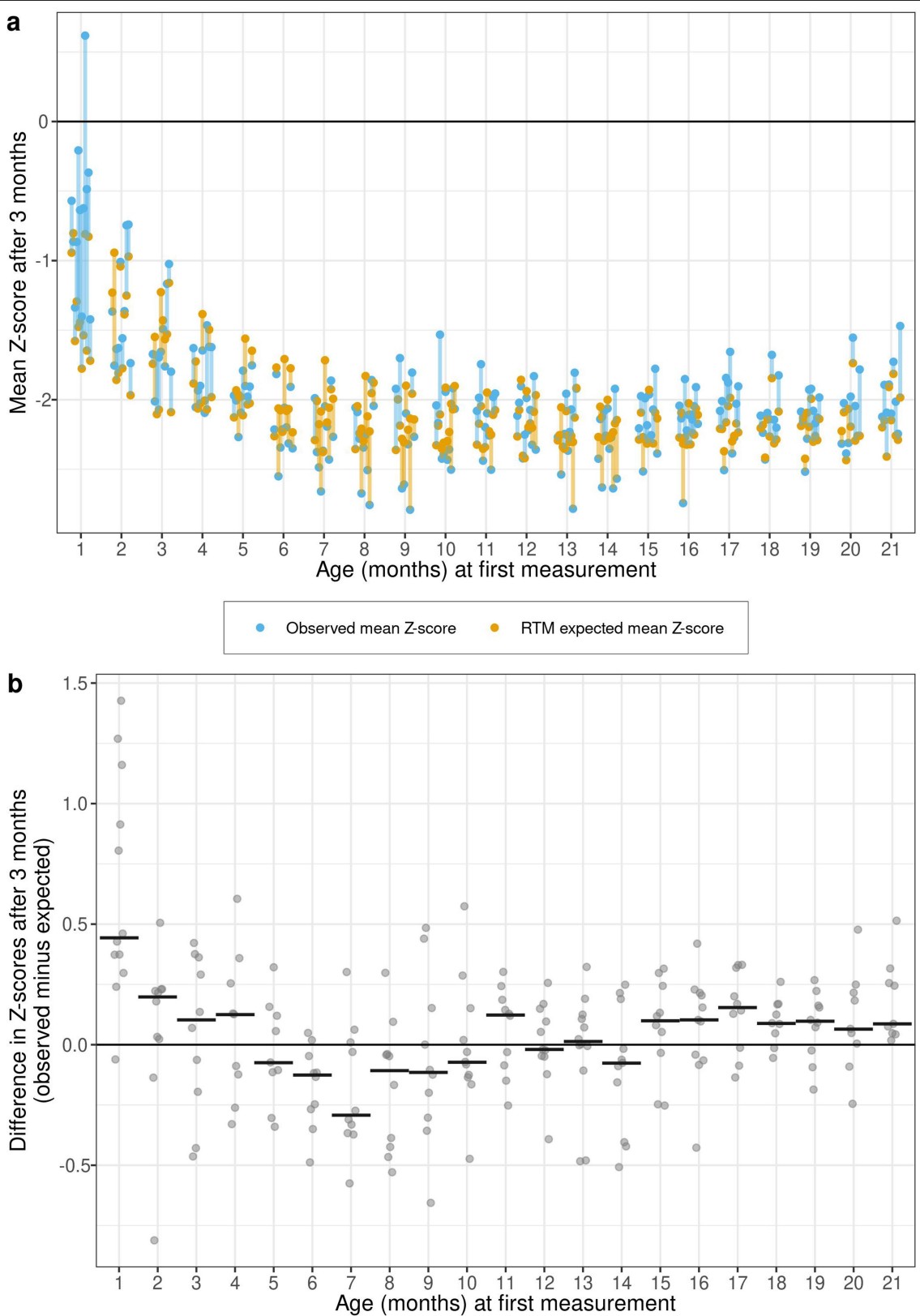

**Extended Data Fig. 8 | RTM effects in wasted children. a**, Expected mean *z*-scores based on the RTM effect (orange) and observed mean *z*-scores (blue) three months after wasted children are measured. The lines connecting cohort-specific observed and expected WLZ at each age are coloured orange if the expected estimate under RTM was higher than the observed mean (indicating lower than expected change in WLZ under RTM alone), and blue if the observed mean was higher than the expected estimate under RTM (indicating higher than expected change in WLZ under RTM alone). For examples, most cohorts experienced larger increases in WLZ than expected in the three-month period beginning in their first month of life (blue lines) and most cohorts experienced smaller increases in WLZ than expected in the three-month periods beginning at ages 6–9 months (orange lines). **b**, Difference between observed means and expected means under a pure RTM effect by cohort, with the median differences by age indicated with horizontal lines. Details on estimation of the RTM effects are in the Methods.

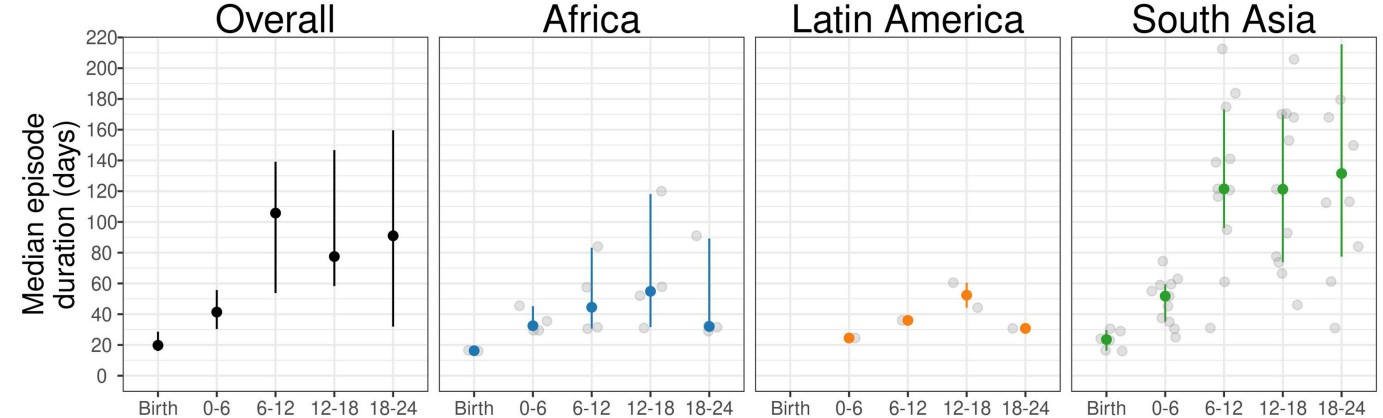

Age in months

**Extended Data Fig. 9 | Duration of wasting episodes by child age and region.**
Duration of wasting episodes that began in each age category, overall ($n$ = 787–
1,940 episodes per age category) and stratified by region (Africa: 377–916
episodes, Latin America: $n$ = 11–25 episodes, South Asia: $n$ = 410–1,146 episodes).
The 'Birth' age category includes measurements in the first 7 days of life and the
'0–3' age category includes ages from 8 days up to 3 months. Estimates were
pooled across cohorts using the median of medians method[60]. Vertical bars
indicate 95% CI around the pooled estimates, and grey points indicate cohort-
specific estimates. Episodes are assumed to start halfway between non-
wasted and wasted measurements, and end halfway between the last wasted
measurement and first recovered estimate. Birth episodes start at birth, so
episodes at birth are generally shorter that post-birth episodes with the same
number of wasted measurements.

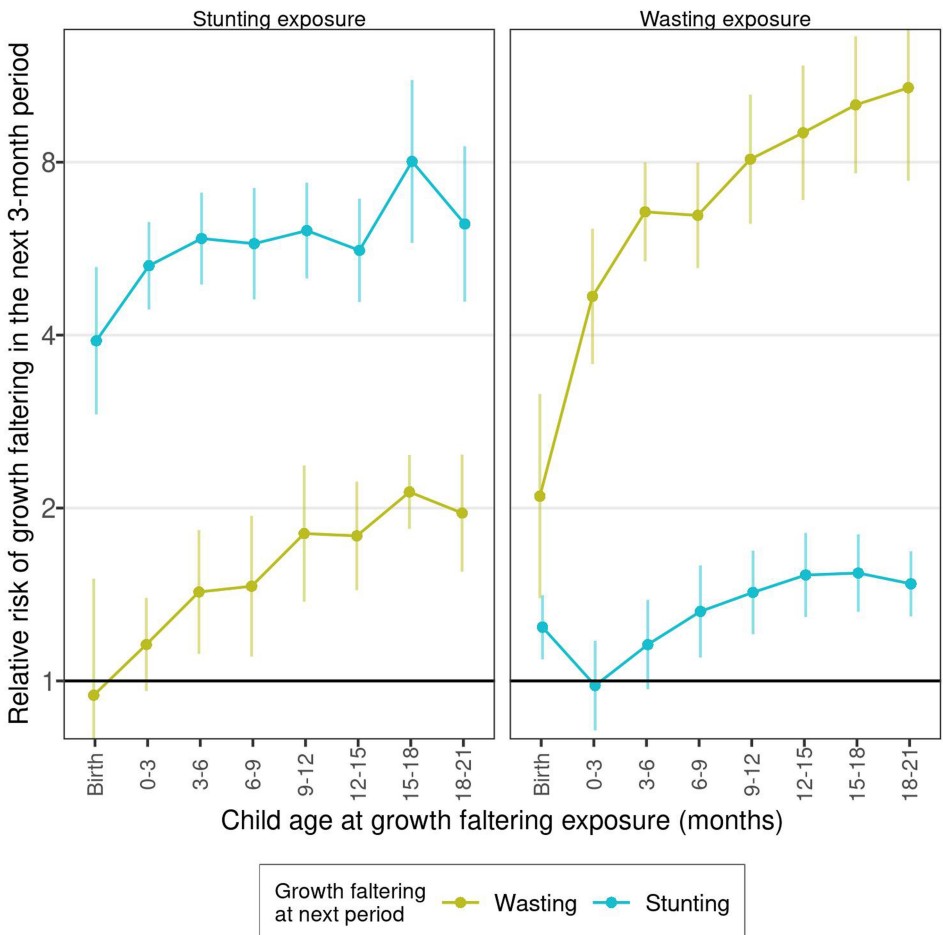

**Extended Data Fig. 10 | Risk of growth faltering in the subsequent three months by whether a child was wasted or stunted at each age.** Relative risk of wasting and stunting in the next three-month period among children who were wasted (right) or stunted (left) at each age. For example, children who were wasted at birth had an approximately twofold increased risk of wasting during the subsequent 0–3 month period, and children who were stunted at birth had an approximately fourfold increased risk of being stunted during the subsequent 0–3 month period. Points indicate relative risks and vertical lines mark 95% CI.

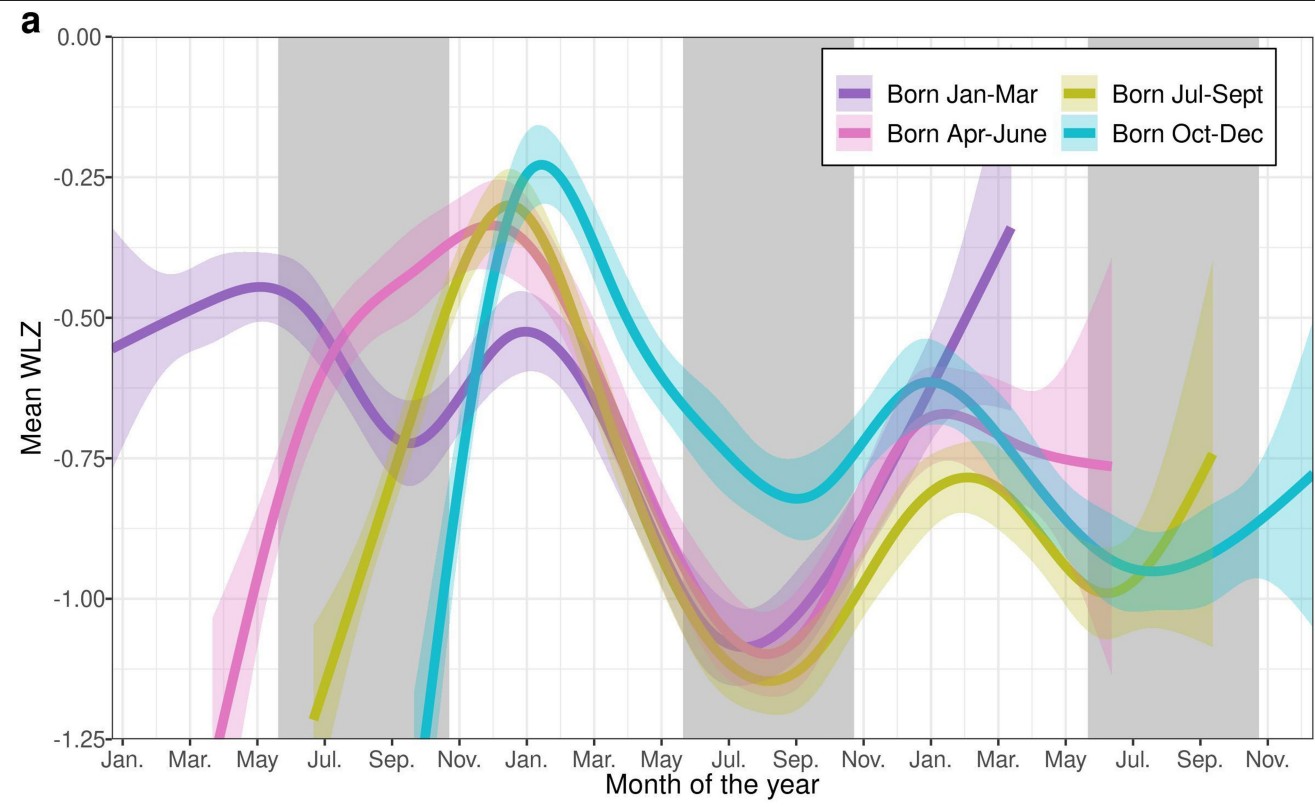

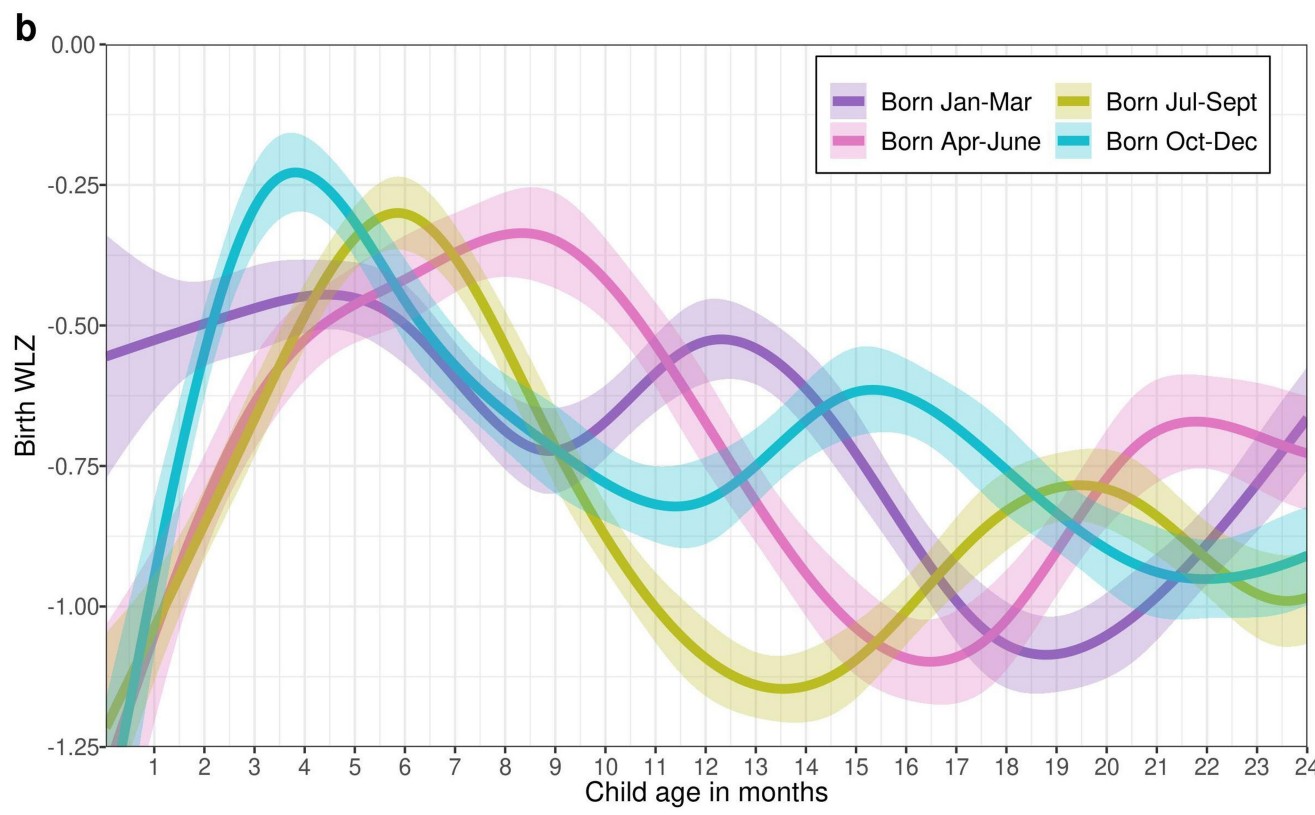

**Extended Data Fig. 11 | Month of birth affects seasonal patterns in WLZ.**
**a**, Mean WLZ by calendar month among South Asian cohorts, with children
stratified by birth month. The *x* axis begins at 1 January of the year of the first
January birthday in each cohort. Grey backgrounds indicate the approximate
timing of seasonal monsoons in South Asia (June–September) Lines represent
mean WLZ estimated using semiparametric cubic splines, and shaded regions
around the means indicate simultaneous 95% CI. Eleven cohorts, 4,040
children and 78,573 measurements were used to estimate the splines. South

Asian children born in July–September had the lowest mean WLZ overall and
children born in April–September had larger seasonal declines in WLZ during
their second year of life than children born October–March. **b**, Mean WLZ from
birth to age 24 months among children from South Asian cohorts stratified by
birth month. Lines represent mean WLZ estimated using semiparametric cubic
splines, and shaded regions around the means indicate simultaneous 95% CI.
Estimates were fit using the same data as in **a**.

**Extended Data Table 1 | Summary of Ki cohorts**

| Region, Study ID | Country | Study Years | Design | Children Enrolled* | Anthropometry measurement ages (months) | Total WLZ measurements* | Primary References |
|---|---|---|---|---|---|---|---|
| **South Asia** | | | | | | | |
| Biomarkers for EE | Pakistan | 2013-2015 | Prospective cohort | 380 | Birth, 1, 2, ..., 18 | 8428 | Iqbal et al 2018 Nature Scientific Reports[69] |
| Resp. Pathogens | Pakistan | 2011-2014 | Prospective cohort | 284 | Birth, 1, 2, ..., 17 | 3164 | Ali et al 2016 Journal of Medical Virology[70] |
| Growth Monitoring Study | Nepal | 2012 - Ongoing | Prospective cohort | 698 | Birth, 1, 2, ..., 24 | 13340 | Not yet published |
| MAL-ED | Nepal | 2010 - 2014 | Prospective cohort | 240 | Birth, 1, 2, ..., 24 | 5695 | Shrestha et al 2014 Clin Infect Dis[71] |
| CMIN93 | Bangladesh | 1993 - 1996 | Prospective Cohort | 280 | Birth, 3, 6, ..., 24 | 5372 | Pathela et al. 2006 Acta Paediatrica[72] |
| Transmission Dynamics of Crypto (TDC) | India | 2008-2011 | Quasi-experimental | 160 | Birth, 1, 2, ..., 24 | 3591 | Sarkar et al. 2013 BMC Public Health[73] |
| CMC Birth Cohort, Vellore | India | 2002 - 2006 | Prospective cohort | 373 | Birth, 0.5, 1, 1.5, ..., 24 | 8697 | Gladstone et al. 2011 NEJM[74] |
| MAL-ED | India | 2010 - 2012 | Prospective cohort | 251 | Birth, 1, 2, ..., 24 | 5697 | John et al 2014 Clin Infect Dis[75] |
| Immune Response Crypto Study | India | 2008 - 2011 | Prospective cohort | 410 | Birth, 1, 2, ..., 24 | 9729 | Kattula et al. 2014 BMJ Open[76] |
| MAL-ED | Bangladesh | 2010 - 2014 | Prospective cohort | 265 | Birth, 1, 2, ..., 24 | 5592 | Ahmed et al 2014 Clin Infect Dis[77] |
| PROVIDE RCT | Bangladesh | 2011 - 2014 | Individual RCT | 700 | Birth, 6, 10, 12, 14. 17, 18, 24, 39, 40, 52, 53 (weeks) | 9202 | Kirkpatrick et al 2015 Am J Trop Med Hyg[78] |
| **Africa** | | | | | | | |
| MAL-ED | Tanzania | 2009 - 2014 | Prospective cohort | 262 | Birth, 1, 2, ..., 24 | 5698 | Mduma et al 2014 Clin Infect Dis[79] |
| Tanzania Child 2 | Tanzania | 2007 - 2011 | Individual RCT | 2400 | 1, 2, ..., 20 | 29518 | Locks et al Am J Clin Nutr 2016[80] |
| MAL-ED | South Africa | 2009 - 2014 | Prospective cohort | 314 | Birth, 1, 2, ..., 24 | 6151 | Bessong et al 2014 Clin Infect Dis[81] |
| MRC Keneba | Gambia | 1987 - 1997 | Cohort | 2931 | Birth, 1, 2, ..., 24 | 40117 | Schoenbuchner et al. 2019, AJCN[7] |
| **Latin America** | | | | | | | |
| CMIN Peru95 | Peru | 1995 - 1998 | Prospective Cohort | 224 | Birth, 1, 2, ..., 24 | 3978 | Checkley et al. 2003 Am J Epidemiol.[82] |
| CMIN Peru89 | Peru | 1989 - 1991 | Prospective Cohort | 210 | Birth, 1, 2, ..., 24 | 2741 | Checkley et al. 1998 Am J Epidemiol.[83] |
| MAL-ED | Peru | 2009 - 2014 | Prospective cohort | 303 | Birth, 1, 2, ..., 24 | 6127 | Yori et al 2014 Clin Infect Dis[84] |
| CONTENT | Peru | 2007 - 2011 | Prospective cohort | 215 | Birth, 1, 2, ..., 24 | 8339 | Jaganath et al 2014 Helicobacter[85] |
| Bovine Serum RCT | Guatemala | 1997 - 1998 | Individual RCT | 315 | Baseline, 1, 2, ...,8 | 2545 | Begin et al. 2008, EJCN[86] |
| MAL-ED | Brazil | 2010 - 2014 | Prospective cohort | 233 | Birth, 1, 2, ..., 24 | 4838 | Lima et al 2014 Clin Infect Dis[87] |

*'Children enrolled' refers to children with any anthropometry measurements under 2 years of age. 'Total WLZ measurements' refers to the number of measurements of weight-for-length Z-score measurements in children under 2 years of age. The following references are included in this table:[7,69-87].

# Extended Data Table 2 | Summary of baseline characteristics in Ki cohorts

| | | CMC-V-BCS-2002, India (N=373) | CMIN Bangladesh93 (N=280) | CMIN Peru89 (N=210) | CMIN Peru95 (N=224) | CONTENT, Peru (N=215) | EE, Pakistan (N=380) | GMS-Nepal (N=698) | Guatemala BSC (N=315) | IRC, India (N=410) | Keneba, The Gambia (N=2954) | MAL-ED, Bangladesh (N=265) | MAL-ED, Bangladesh (N=265) | MAL-ED, Brazil (N=233) | MAL-ED, India (N=251) | MAL-ED, Nepal (N=240) | MAL-ED, Peru (N=303) | MAL-ED, South Africa (N=314) | MAL-ED, Tanzania (N=262) | PROVIDE, Bangladesh (N=700) | ResPak, Pakistan (N=284) | Tanzania Child2 (N=240) | TDC, India (N=160) |
|---|---|---|---|---|---|---|---|---|---|---|---|---|---|---|---|---|---|---|---|---|---|---|---|
| **Sex** | Female | 187 (50.1%) | 122 (43.6%) | 100 (47.6%) | 96 (42.9%) | 109 (50.7%) | 185 (48.7%) | 328 (47.0%) | 162 (51.4%) | 185 (45.1%) | 1426 (48.3%) | 136 (51.3%) | 136 (51.3%) | 113 (48.5%) | 138 (55.0%) | 110 (45.8%) | 143 (47.2%) | 159 (50.6%) | 133 (50.8%) | 332 (47.4%) | 136 (47.9%) | 1184 (49.3%) | 75 (46.9%) |
| | Male | 186 (49.9%) | 158 (56.4%) | 110 (52.4%) | 128 (57.1%) | 106 (49.3%) | 195 (51.3%) | 370 (53.0%) | 153 (48.6%) | 225 (54.9%) | 1528 (51.7%) | 129 (48.7%) | 129 (48.7%) | 120 (51.5%) | 113 (45.0%) | 130 (54.2%) | 160 (52.8%) | 155 (49.4%) | 129 (49.2%) | 368 (52.6%) | 148 (52.1%) | 1216 (50.7%) | 85 (53.1%) |
| **Birthweight: Mean (SD)** | | 2910 (434) | 2560 (967) | 3550 (492) | 3690 (367) | 3070 (323) | 2640 (506) | 2660 (423) | | 2890 (448) | 2970 (421) | 2800 (412) | 2800 (412) | 3340 (483) | 2890 (442) | 2980 (390) | 3130 (430) | 3130 (464) | 3180 (452) | 2780 (371) | 2930 (525) | 3230 (474) | 2910 (450) |
| **Maternal age: Mean (SD)** | | 24.1 (4.11) | | | | | 30.0 (3.99) | 24.0 (5.09) | 25.2 (6.21) | 23.7 (3.68) | 27.4 (7.16) | 24.8 (4.99) | 24.8 (4.99) | 24.8 (5.53) | 23.9 (4.10) | 26.4 (3.76) | 24.2 (6.06) | 26.4 (6.86) | 28.4 (6.67) | 24.7 (4.65) | | 26.4 (5.04) | |
| **Maternal weight: Mean (SD)** | | | | | | | | | | | | 49.6 (8.54) | 49.6 (8.54) | 61.9 (11.8) | 50.4 (9.37) | 56.3 (8.27) | 55.5 (8.95) | 67.5 (14.9) | 55.7 (9.11) | 49.3 (9.41) | | 62.2 (11.7) | |
| **Maternal education (years): Mean (SD)** | | 5.43 (4.10) | | | | | 0.873 (2.51) | 2.48 (4.01) | 4.07 (2.95) | 5.06 (4.61) | | 5.63 (2.58) | 5.63 (2.58) | 9.18 (2.81) | 7.88 (3.17) | 8.77 (3.44) | 7.81 (2.79) | 10.3 (1.94) | 6.17 (1.79) | 4.33 (3.62) | | 7.90 (2.27) | |
| **Number of rooms** | 4+ | 14 (3.75%) | | | | 78 (36.3%) | | 323 (46.3%) | | 17 (4.17%) | | 12 (4.96%) | 12 (4.96%) | 127 (60.5%) | 25 (10.6%) | 131 (55.5%) | 139 (51.1%) | 196 (76.3%) | 108 (43.2%) | 23 (3.29%) | | | 5 (3.13%) |
| | 1 | 202 (54.2%) | | | | 44 (20.5%) | | 49 (7.02%) | | 185 (45.3%) | | 152 (62.8%) | 152 (62.8%) | 4 (1.90%) | 84 (35.7%) | 52 (22.0%) | 19 (6.99%) | 14 (5.45%) | 13 (5.20%) | 507 (72.4%) | | | 91 (56.9%) |
| | 2 | 106 (28.4%) | | | | 54 (25.1%) | | 145 (20.8%) | | 170 (41.7%) | | 50 (20.7%) | 50 (20.7%) | 20 (9.52%) | 78 (33.2%) | 31 (13.1%) | 52 (19.1%) | 22 (8.56%) | 63 (25.2%) | 108 (15.4%) | | | 49 (30.6%) |
| | 3 | 51 (13.7%) | | | | 39 (18.1%) | | 181 (25.9%) | | 36 (8.82%) | | 28 (11.6%) | 28 (11.6%) | 59 (28.1%) | 48 (20.4%) | 22 (9.32%) | 62 (22.8%) | 25 (9.73%) | 66 (26.4%) | 62 (8.86%) | | | 15 (9.38%) |
| **Number of children <5yrs** | 1 | | | | | | | | | 89 (21.7%) | | | | | | | | | | 512 (73.1%) | | 1640 (68.6%) | |
| | 2+ | | | | | | | | | 321 (78.3%) | | | | | | | | | | 188 (26.9%) | | 749 (31.4%) | |
| **Improved sanitation** | 1 | | | | | | 201 (93.5%) | | | | | 204 (84.3%) | 204 (84.3%) | 206 (98.1%) | 108 (46.4%) | 235 (99.6%) | 65 (24.7%) | 4 (1.60%) | | 58 (96.7%) | | | |
| | 0 | | | | | | 14 (6.51%) | | | | | 38 (15.7%) | 38 (15.7%) | 4 (1.90%) | 125 (53.6%) | 1 (0.424%) | 198 (75.3%) | 246 (98.4%) | | 2 (3.33%) | | | |
| **Food security level** | Food Secure | | | | | | | 479 (71.1%) | | | | 161 (83.0%) | 161 (83.0%) | 3 (2.33%) | 190 (89.6%) | 94 (73.4%) | 27 (23.9%) | 132 (56.7%) | | | | | |
| | Mildly Food Insecure | | | | | | | 106 (15.7%) | | | | 4 (2.06%) | 4 (2.06%) | 11 (8.53%) | 5 (2.36%) | 15 (11.7%) | 29 (25.7%) | 19 (8.15%) | | | | | |
| | Food Insecure | | | | | | | 89 (13.2%) | | | | 29 (14.9%) | 29 (14.9%) | 115 (89.1%) | 17 (8.02%) | 19 (14.8%) | 57 (50.4%) | 82 (35.2%) | | | | | |

# Reporting Summary

## Statistics

For all statistical analyses, confirm that the following items are present in the figure legend, table legend, main text, or Methods section.

| n/a | Confirmed | |
|---|---|---|
| ☐ | ☒ | The exact sample size (*n*) for each experimental group/condition, given as a discrete number and unit of measurement |
| ☐ | ☒ | A statement on whether measurements were taken from distinct samples or whether the same sample was measured repeatedly |
| ☒ | ☐ | The statistical test(s) used AND whether they are one- or two-sided *Only common tests should be described solely by name; describe more complex techniques in the Methods section.* |
| ☐ | ☒ | A description of all covariates tested |
| ☐ | ☒ | A description of any assumptions or corrections, such as tests of normality and adjustment for multiple comparisons |
| ☐ | ☒ | A full description of the statistical parameters including central tendency (e.g. means) or other basic estimates (e.g. regression coefficient) AND variation (e.g. standard deviation) or associated estimates of uncertainty (e.g. confidence intervals) |
| ☐ | ☒ | For null hypothesis testing, the test statistic (e.g. *F*, *t*, *r*) with confidence intervals, effect sizes, degrees of freedom and *P* value noted *Give P values as exact values whenever suitable.* |
| ☒ | ☐ | For Bayesian analysis, information on the choice of priors and Markov chain Monte Carlo settings |
| ☐ | ☒ | For hierarchical and complex designs, identification of the appropriate level for tests and full reporting of outcomes |
| ☐ | ☒ | Estimates of effect sizes (e.g. Cohen's *d*, Pearson's *r*), indicating how they were calculated |

*Our web collection on statistics for biologists contains articles on many of the points above.*

## Software and code

Policy information about availability of computer code

| Data collection | This manuscript is a secondary data analysis of existing study data from 21 cohorts, so we were not involved in original data collection. |
|---|---|
| Data analysis | All analyses were conducted using R statistical software, and scripts that reproduce all analyses are available on Github here: https://github.com/child-growth/ki-longitudinal-growth |

For manuscripts utilizing custom algorithms or software that are central to the research but not yet described in published literature, software must be made available to editors and reviewers. We strongly encourage code deposition in a community repository (e.g. GitHub). See the Nature Portfolio guidelines for submitting code & software for further information.

## Data

Policy information about availability of data

All manuscripts must include a data availability statement. This statement should provide the following information, where applicable:
- Accession codes, unique identifiers, or web links for publicly available datasets
- A description of any restrictions on data availability
- For clinical datasets or third party data, please ensure that the statement adheres to our policy

Data are available upon agreement from the data contributors of individual studies, whose contact information is available from the Bill and Melinda Gates Foundation Knowledge Integration project (email [TO BE ADDED] upon reasonable request.

# Research involving human participants, their data, or biological material

Policy information about studies with [human participants or human data](). See also policy information about [sex, gender (identity/presentation), and sexual orientation]() and [race, ethnicity and racism]().

| | |
|---|---|
| Reporting on sex and gender | We use the term sex throughout; data on gender was not collected in the original studies used. We include sex as a risk factor and examine sex as an effect modifier of mortality risk (supplementary material) |
| Reporting on race, ethnicity, or other socially relevant groupings | We did not have information on the race or ethnic groups of study participants. |
| Population characteristics | See below. |
| Recruitment | N/A: Secondary analysis of 21 completed studies. |
| Ethics oversight | N/A |

Note that full information on the approval of the study protocol must also be provided in the manuscript.

# Field-specific reporting

Please select the one below that is the best fit for your research. If you are not sure, read the appropriate sections before making your selection.

☐ Life sciences    ☒ Behavioural & social sciences    ☐ Ecological, evolutionary & environmental sciences

For a reference copy of the document with all sections, see [nature.com/documents/nr-reporting-summary-flat.pdf]()

# Behavioural & social sciences study design

All studies must disclose on these points even when the disclosure is negative.

| | |
|---|---|
| Study description | This study performed a quantitative analysis of de-identified secondary, longitudinal data on child growth. |
| Research sample | The data analyzed in this study were amassed as part of the Knowledge Integration (ki) initiative of the Bill & Melinda Gates Foundation, which aggregated observations on millions of participants from a global collection of studies on child birth, growth and development. We selected longitudinal cohorts from the database that met five inclusion criteria: 1) conducted in LMICs; 2) enrolled children between birth and age 24 months and measured their length and weight repeatedly over time; 3) did not restrict enrollment to acutely ill children; 4) enrolled at least 200 children; and 5) collected anthropometry measurements at least every month. 18 cohorts met inclusion criteria, including 10,854 children and 187,215 total anthropometry measurements. |
| Sampling strategy | Not applicable. |
| Data collection | Not applicable. |
| Timing | Included datasets were collected between 1987 and 2014. |
| Data exclusions | We dropped 385 biologically implausible measurements (0.2%) of WLZ (> 5 or < −5 Z-score), 212 (0.1%) of LAZ (> 6 or < −6 Z-score), and 216 (0.1%) of WAZ (>5 or < −6), following WHO recommendations. |
| Non-participation | Not applicable. |
| Randomization | Participants were not randomly assigned. |

# Reporting for specific materials, systems and methods

We require information from authors about some types of materials, experimental systems and methods used in many studies. Here, indicate whether each material, system or method listed is relevant to your study. If you are not sure if a list item applies to your research, read the appropriate section before selecting a response.

## Materials & experimental systems

| n/a | Involved in the study |
|-----|----------------------|
| ☒ | ☐ Antibodies |
| ☒ | ☐ Eukaryotic cell lines |
| ☒ | ☐ Palaeontology and archaeology |
| ☒ | ☐ Animals and other organisms |
| ☒ | ☐ Clinical data |
| ☒ | ☐ Dual use research of concern |
| ☒ | ☐ Plants |

## Methods

| n/a | Involved in the study |
|-----|----------------------|
| ☒ | ☐ ChIP-seq |
| ☒ | ☐ Flow cytometry |
| ☒ | ☐ MRI-based neuroimaging |

