## [Peer Review File · Nature]

Manuscript Title: Child wasting and concurrent stunting in low- and middle-income countries

Reviewer Comments & Author Rebuttals

Reviewer Reports on the Initial Version:

Referees' comments:

Referee #1 (Remarks to the Author):

question has been of interest in child nutrition and health due to both short term risks associated with wasting and the potential for its cumulative risk when repeated episodes happen, especially close to one another.

My comments on the stunting paper in relation to how to pool and interpret cohorts that are from different periods and places also applies here noting that the overall duration of cohorts is shorter here (although still spanning 2.5 to 3 decades).

Similarly, I couldn't figure out what metric/parameter was meta-analyzed in this paper. The paper states, "Unless otherwise indicated, we calculated estimates within cohorts and then pooled estimates across cohorts using random effects models ...". In a typical meta-analysis, a single parameter, for example HR from a proportional hazard model, is pooled. Here, an entire age+season pattern of incidence/prevalence is analyzed. What is pooled?

The paper contrasts its findings with those of cross-sectional surveys stating that "suggesting that the wasting burden is likely far higher than cross-sectional surveys suggest". I think this statement and similar ones should be made clear and specific. For any episodic condition, we can measure point prevalence or life time incidence. Both are relevant: the former is what the system manages; the latter is what the individual experiences. The two measures are not really comparable nor should they be contrasted; they are complementary in a way that depends on inter-individual correlation. The "burden" to the population and individual and to the system can differ, and can only be informed with repeated cross sectional data (to capture the secular trends) together with individual follow up data that measure the inter-individual variations. Therefore, best to avoid unclear terminology such as "burden" and state the differences between point prevalence, life time incidence and the role of how episodes are distributed among individuals.

Seasonality is interesting but probably could use some more in depth analysis/discussion. Among others, in Figure 3a, cohorts from the same country seem to have different seasonal patterns (e.g. Tanzania and Peru). Also, work by some of the consortium members has indicated that wasting follows diarrhea episodes, with other work indicating (I believe in Peru) a relationship between rain and diarrhea. I believe there are also some studies using DHS data and satellite images – to detect for example measures of greenness – to related seasonal indicators and short-term nutritional status.

Referee #2 (Remarks to the Author):

Key results: This is an important and insightful piece of work on the occurrence and reoccurrence of wasting during the early years of life. The paper provides a clear epidemiologic presentation of important age ranges for the incidence of wasting and its reoccurrence. The work sheds light on the inadequacy of current methods to examine wasting patterns during, not only the early years of

life but also beyond the first 2 years of life. The manuscript does not have flaws that should prohibit its publication based on what I can detect from the materials provided. This paper has obviously been a gargantuan task from the original studies to pooling the data, analyzing it and the data's interpretation - for this, my sincere credit to the authors.

Originality and significance: This an important contribution to this body of literature. I would say more broadly that this paper does not focus on early childhood as it states but instead on infants/toddler. (Line 164: 'Life-course of early childhood'). The reasons for the importance of these semantics is and why it may be important to make this differentiation is because of the very fluctuations in nutritional status the authors describe that could and have been documented to continue to fluctuate post the 2 years of age period. For example: Gausman, J.; Kim, R.; Subramanian, S.V. Stunting trajectories from post-infancy to adolescence in Ethiopia, India, Peru, and Vietnam. *Matern. Child Nutr.* 2019, 15, e12835

Results & methods: The results presented will broadly be of immediate interest to many researchers working in the area of child growth and practitioners who focus their efforts on undernutrition prevention and treatment. This paper is especially relevant during a time when there is much debate about the best way to intervene when a child is wasted and/or stunted, discussions and funding put towards the treatment of moderate acute malnutrition programs and siloed programming for stunted and wasted children. Much of these efforts take place with primarily cross-sectional data informing them and thus this work is appropriately timed. Because of the multi-sectorality of nutrition as whole, other disciplines may well find use for the information contained in this paper.

The methods undertaken by the authors to answer their research questions of interest are appropriate. Reporting is transparent but more gained, from a readers point of view, from understanding the following:

There were some issues I had with the reporting of the interpretation of the mean WLZ (Figure 2a) in the outlined text in lines 203-208. It would perhaps to place more emphasis on the regional difference in WLZ and that mean WLZ in South Asia is well below zero reflecting, even if the same pattern, very different nutritional status early in infancy. This is also reflected in prevalence estimates reflected in Figure 2c.

Line 236-238: While this overall hypothesis is understandable, the authors do not provide ground truthing in each of these countries/ from study PIs if this is the best way to assess seasonal patterns. Agricultural lean periods in each of these regions could vary and not entirely be based on rainfall, for example, long period of arid temperature.

Lines 254-255 state the high variability in wasting status of children from month to month but do not quantify it. Providing that range of variability here would help readers understand the justification of the imposed 60 day recovery period.

Line 307-309 - This is a methodological issue and while stated in the limitations, explaining how this effects the studies results and interpretations to the reader is important. It is not quite enough to just point this out.

Lines 636-648 - there is no mention of controlling for time (year study was conducted) in the random effect models, was this done?

The authors do not discuss issues of regression to the mean, an issue most longitudinal studies of anthropometry may experience. Was RTM assessed across longitudinal measurements especially among those extrema/ wasted children who 'recovered' during their next visit?

Finally, there is no description of the populations being studied as far as I can tell anywhere. The

average age of mothers at the time children were enrolled, food security status, sex distribution and so forth by study are not provided. Thus the conclusions while broadly on point leave the reader lacking the context of some statements such as those made in lines 329-330 about target populations of interest.

Appropriate use of statistics and treatment of uncertainties: The authors switch back and forth between 95% CI and 95% credible intervals which may not be understood by readers at first glance. Perhaps a footnote as to why credible intervals vs confidence intervals would be of value? Finally, given the longitudinal nature of cohort included in this study and its analyses, the authors do not provide as much insights to the variability between children within regions. Again here, interpretations and conclusions could potentially be more rich if this is done.

Suggested improvements:

Line 117: What does unrealistic mean? Please expand.

Lines 258-260: Incidence of what? Wasting? Or recovery from wasting?

Lines 294-296: Was the converse also applicable? Were children born stunted at higher risk of wasting later within the first 2 years?

Lines 305-307- To what extent did the authors consider timing (which month for example the measurement was taken) of anthro measurement when comparing DHS distribution and study cohorts included in this study?

Lines 307: What is the resultant issue of the limitation of children being measured at different times? This is not explained by the authors in the discussion and is important to expand upon in a paper so heavily focused on anthropometry.

Lines 314-316 - Did the cohort studies not have mortality data to assess this bias?

The section on the overlap between wasting and stunting does not seem adequately flushed out. Not many insights are provided either in the results or discussion about these findings and their implications

Figure 3a: There appears to be some variation in seasonal patterns that are not necessarily congruent with the explained results. For example, it appears that in the MAL-ED Peru sample and the ResPark sample from Pakistan, the mean WHZ is not necessarily lowest during periods of highest rainfall. Encourage further description and details of these findings.

The dramatic shifts in WLZ by season but larger fluctuations in WLZ by birth month in South Asia noted by the authors ought to be flushed out in the manuscript.

Extended Data Table 1009 1| Summary of ki cohorts: There are formatting issues with this table.

Referee #3 (Remarks to the Author):

1. Summary

- a. What makes wasting a "leading" indicator?
- b. Overall very good summary – the intro should be crafted to support the analysis performed in this paper. Currently it comes across as only tangentially connected.
- c. What happened to stunting? Its in the title, intro, and toward the end of the paper, but missing from summary.

2. Introduction, especially the first paragraph, is quite poorly written. It is repetitive, meandering, and poorly cited. The concepts are fine, but the presentation needs a lot of work, including the choice and placement of citations.
3. I asked this for the other papers as well, but why stop at 24 months? Don't these data sources continue beyond that? Seeing as there appears to be accumulation of faltering with age, stopping at age 2 seems like an odd choice.
4. Line 172-4 ◊ this hypothesis is not really a hypothesis. It is essentially stating that "we hypothesize we will learn something new." Please either drop this sentence or reframe as to what the actual hypotheses were. Also, the title of the paper and the intro mention stunting. Is that not part of the hypotheses or analysis?
5. Line 176 and 184 – similar to the other papers, how did the dataset go from millions to <11k? And why so many fewer than the "causes and consequences" paper that presumably was done from the same dataset(s)?
6. Line 197-201 – either need to define WAZ and LAZ or reframe this part about Asia/Africa cohorts to be consistent with the description for LatAm. Also, the finding that LAZ was steady in first 6 months in DHS is suspect.
7. On the topic of comparisons to DHS (extended data fig 3), the description and performance of these comparisons could be improved:
 - a. need to be more specific with which DHS they were matched. Same year only? +/- 5 years? Seasonality adjusted in DHS?
 - b. Were they matched to the specific location where the KI studies were performed within each country? Or only the summary country-level results from each DHS?
 - c. Need to match study-by-study and present that information, not pooled across entire continents.
8. It is not helpful – and potentially counterproductive – to make regional estimates at a time, and for a topic, when most health information will need to be as local as possible. The settings across regions are very diverse and, even within regions, they are very different. It would be more instructive and helpful to present data and results for each study individually and then show the pooled results across the studies for an overall. This is already done for Figure 1 and Figure 3 and should be repeated for figures 2, 4, and 5.
9. Line 204 – mean WLZ near zero at birth? This seems unlikely unless there were few low birthweight or preterm births in the cohorts. Do you have that information? Also, figure 2 does not support this statement. Mean WLZ at birth was -0.8 in South Asia, -0.5 in Africa, and +0.5 in LatAm.
10. Line 209 – need to define "severely wasted" in the main text somewhere.
11. Line 211 – random effects pooled prevalence of wasting at birth is not meaningful across such diverse settings, especially in a section about "regional differences."
12. Figure 2 –
 - a. the text description of the figures is inaccurate. The methods state this is a spline with 95% UI, but the figure text states these are percentiles. Which is it?
 - b. All of these figures that show overall, Africa, LatAm, and S Asia are all stated to be random effects meta-analyses. If that is the case, they should be presented as such with all the data and the pooled effect. Right now only the results are shown which, when pooled across such diverse places, cannot be interpreted readily.
13. Line 224-226 – this sentence is misleading. The two strongest factors in the cited study influencing birthweight were newborn LAZ and duration of pregnancy. In turn, the three biggest factors were duration of pregnancy, placental weight, and maternal height. Placental weight can be influenced by acute illness and nutrition, but in other large studies (such as this one - <https://pubmed.ncbi.nlm.nih.gov/24354883/>) has also been shown to be strongly correlated with chronic health conditions such as hypertension, pregnancy complications such as preeclampsia, and fetal abnormalities. The conclusion of Ashorn and colleagues' analysis is that constitutional factors play an outsized role. That conclusion is not reflected in the text here. The text should be revised accordingly and, ideally, re-contextualized. Maternal height is extensively explored in one of the companion papers to this one – can any of that insight be brought to bear here?

14. The sections on "Wasting incidence and recovery" and "persistent and concurrent growth failure" are, in my opinion, the only novel insights of this paper. The paper could be substantially strengthened by focusing much more, perhaps almost exclusively, on those two aspects of the analysis, with much of the rest relegated to supplementary material.

15. The authors' discussion of seasonally-targeted interventions is compelling as well although the seasonality finding is not novel. I would like to see how these seasonality patterns compare to those in the matched DHS studies.

16. Figure 5 and "persistent and concurrent growth failure" – multiple questions/ comments:

a. the world nutrition targets 2025 only list WLZ (wasting) and LAZ (stunting) as targets. The authors also largely "ignore" WAZ (underweight) until this figure. My question is, why include here at all?

b. Is there any evidence that wasting + underweight is any worse than wasting alone? Or is it just a manifestation of more severe wasting? Put another way, there is ample evidence that both WLZ and LAZ are independently modifiable risk factors for poor outcomes in children (death and developmental) – is there any evidence that WAZ is either an independent risk factor or is independently modifiable? Or is it entirely mediated through the health effects of wasting and stunting? (McDonald et al did not evaluate severity of stunting, wasting, and underweight so citing their findings here would not be particularly helpful.) This may not be entirely answerable within this paper, but this is a vitally important question that is thematically missing from the companion papers and needs to be strengthened.

c. I would think it potentially more instructive – and novel – to a) evaluate overlap between moderate, severe stunting and moderate, severe wasting and see if that "gets rid of" underweight entirely and b) investigate how this overlap of concurrent growth failure categories compare to the overlaps in matched DHS studies?

17. Line 257-260 and Line 587-590 – what is the validity of this assumption? How many of those who had birthweight measured and who did NOT have wasting at birth also had incident wasting in the first 60 days? Or is that assessment of high incidence of wasting in early life an artifact of this assumption?

18. Line 336 – this sentence, "Wasting treatment programs have traditionally focused on children from 6-59 months." Is probably one of the single most important statements in this entire paper. What is the history of that focus? Pragmatic? Evidence based? If evidence based, does the current analysis suggest therefore that the phenotype of CGF has changed? Or is the finding instead that the historical, programmatic guidance has been chronically misguided? Either way, this theme needs to be brought forward and emphasized.

19. Line 591 – how is this different than "period prevalence"?

20. Line 593 – what is the conceptual basis/ principle for this definition of "recovery from wasting"?

21. References

a. Line 1 ◊ #1 (growth standards definitions) is not relevant here (suggest dropping) and #2 (black et al 2013) is quite out of date (suggest updating).

b. Line 131 ◊ Neither #1 nor #6 support the statement in this sentence.

c. Suggest carefully checking all references to ensure they are appropriately matched to the statements they are intended to support. Is it possible that something got out of sync?

Author Rebuttals to Initial Comments:

Referees' comments:

Referee #1 (Remarks to the Author):

- (1) question has been of interest in child nutrition and health due to both short term risks associated with wasting and the potential for its cumulative risk when repeated episodes happen, especially close to one another.**

My comments on the stunting paper in relation to how to pool and interpret cohorts that are from different periods and places also applies here noting that the overall duration of cohorts is shorter here (although still spanning 2.5 to 3 decades).

Response: We agree that observations from historical cohorts could reflect different conditions than more recent ones. In response to this suggestion, we revised the cohort eligibility criteria for all three papers to include studies that had a median year of birth in 1990 or later, though this change did not affect the set of monthly-measured studies used in this manuscript. Additionally, 3 smaller studies were added after we included cohorts with $N < 200$ children in response to comments on the linear growth failure manuscript (Referee #3). This change increased the number of monthly measured cohorts used to study child wasting in this paper from 18 ($n=10,854$ children, 187,215 measurements) to 21 ($n=11,448$ children and 198,154 measurements). The revised Figure 1 includes a list of included cohorts, ED Figure 1 summarizes those included and excluded, and ED Figure 7 includes a summary of follow-up by age for each cohort.

- (2) Similarly, I couldn't figure out what metric/parameter was meta-analyzed in this paper. The paper states, "Unless otherwise indicated, we calculated estimates within cohorts and then pooled estimates across cohorts using random effects models ...". In a typical meta-analysis, a single parameter, for example HR from a proportional hazard model, is pooled. Here, an entire age+season pattern of incidence/prevalence is analyzed. What is pooled?**

Response: We have clarified in the methods that each analysis pools a single parameter (such as mean WLZ) within specific age-strata. We estimated each age-specific mean using a separate estimation and pooling step. We first estimated the mean in each cohort, and then pooled age-specific means across cohorts allowing for a cohort-level random effect. This approach enabled us to include the most information possible for each age-specific mean, while allowing for slightly different measurement schedules across the cohorts. Each cohort's data only contributed to WLZ or prevalence estimates at ages for which it contributed data.

We have updated the methods to clarify this: Lines 846-855

“We conducted a 2-stage individual participant meta-analysis, estimating wasting outcomes within specific cohorts and pooling estimates within each age strata using random effects models. For example, we estimated each age-specific mean using a two-step process. We first estimated the mean in each cohort, and then pooled age-specific means across cohorts allowing for a cohort-level random effect. We estimated overall pooled effects, and pooled estimates specific to South Asian, African, or Latin American cohorts, depending on the analysis. We repeated the pooling of all statistics presented in the figures using fixed effects models as a sensitivity analysis (<https://child-growth.github.io/wasting/fixed-effects.html>). The pooling methods are described in greater detail in Benjamin-Chung (2020).²⁹ All pooling was completed using the `rma()` function from the “metafor” package in the R language (version 3.4.0).⁶⁵”

(3) The paper contrasts its findings with those of cross-sectional surveys stating that “suggesting that the wasting burden is likely far higher than cross-sectional surveys suggest”. I think this statement and similar ones should be made clear and specific. For any episodic condition, we can measure point prevalence or life time incidence. Both are relevant: the former is what the system manages; the latter is what the individual experiences. The two measures are not really comparable nor should they be contrasted; they are complementary in a way that depends on inter-individual correlation. The “burden” to the population and individual and to the system can differ, and can only be informed with repeated cross sectional data (to capture the secular trends) together with individual follow up data that measure the inter-individual variations. Therefore, best to avoid unclear terminology such as “burden” and state the differences between point prevalence, life time incidence and the role of how episodes are distributed among individuals.

Response: This is a valuable distinction that we have clarified in the revised manuscript, noting that the novel contributions of the present analysis focus on elucidating the child-level burden based on incidence and repeated episodes. We have been more specific throughout the manuscript, replacing “burden” with “incidence”. Due to ages of children measured in the included cohorts, we cannot look at life-time incidence, but we examine cumulative incidence from birth to age 24 months. The revised discussion makes this more nuanced distinction on lines 292-305:

“Nearly all large-scale studies of child wasting including DHS report point prevalence of wasting, which has enabled broad comparisons between populations but failed to capture the number of episodes of wasting during the course of childhood. By combining information across several longitudinal cohorts, this study demonstrated that children experience far higher cumulative incidence of wasting between birth and 24 months (29.2%) than previously known. Children in all included cohorts experienced

higher incidence compared with prevalence due to the episodic nature of acute malnutrition, but the pattern was most stark in South Asian cohorts where, from birth to 24 months and pooled across cohorts, 29.2% of children experienced at least one wasting episode, 3.4% were persistently wasted, and 10.6% had experienced concurrent wasting and stunting. This evidence shows that the cumulative, child-level burden of wasting is higher than cross-sectional prevalence measures would suggest, and assessments of wasting and stunting in isolation fail to account for their joint burden, which can be substantial. This is of particular relevance in South Asia where the largest number of stunted and wasted children live, and while stunting prevalence is decreasing, wasting prevalence is not.”

(4) Seasonality is interesting but probably could use some more in depth analysis/discussion. Among others, in Figure 3a, cohorts from the same country seem to have different seasonal patterns (e.g. Tanzania and Peru). Also, work by some of the consortium members has indicated that wasting follows diarrhea episodes, with other work indicating (I believe in Peru) a relationship between rain and diarrhea. I believe there are also some studies using DHS data and satellite images – to detect for example measures of greenness – to related seasonal indicators and short-term nutritional status.

Response: Based on this comment and Reviewer 2 comments 5 and 20, we have updated the seasonality analysis to use spatially-explicit rainfall from the cities/towns where cohorts were measured rather than national averages. When matched to finer resolution data the consistency between seasonal rainfall patterns and WLZ was improved. The overall findings were strengthened by this improved resolution of data in the analysis, summarized in Figure 3a-c. In the revised analysis, we show that some cohorts from the same country had different seasonal patterns in WLZ and also had different rainfall patterns. For example, within Peru CONTENT and CMIN 1995 cohorts had a high seasonality index based on rainfall and exhibited large variation in WLZ by month, while the CMIN 1989 and MAL-ED Peru cohorts had low seasonality indices, and exhibited much less variation in WLZ by month (Figure 3a).

The drivers of the seasonal pattern of wasting may vary be location or calendar year. Diarrhea, other infectious diseases, and seasonal food insecurity will all vary by location and by year, so the seasonality analysis still have limitations when using calendar months and rainfall to define seasons, because we are not directly measuring the cause of the wasting, rather a uniform proxy. Even with these limitations, though, there is a clear pattern of higher weight-for-length Z-scores during the dry season across the cohorts.

We have also updated the discussion to include the link between wasting and seasonal diarrhea, though we are limited by word count to discuss the nuances in drivers of seasonal wasting.

- Lines 311-313: *“Nevertheless, the high degree of consistency of the rainfall-wasting association provides strong support for the development of seasonally targeted interventions to prevent wasting in food-insecure populations, akin to seasonal malaria chemoprevention programs in the Sahel.”*

Referee #2 (Remarks to the Author):

(1) Key results: This is an important and insightful piece of work on the occurrence and reoccurrence of wasting during the early years of life. The paper provides a clear epidemiologic presentation of important age ranges for the incidence of wasting and its reoccurrence. The work sheds light on the inadequacy of current methods to examine wasting patterns during, not only the early years of life but also beyond the first 2 years of life. The manuscript does not have flaws that should prohibit its publication based on what I can detect from the materials provided. This paper has obviously been a gargantuan task from the original studies to pooling the data, analyzing it and the data's interpretation - for this, my sincere credit to the authors.

Response: We are grateful for the reviewer's time and constructive input that has significantly improved the manuscript.

(2) Originality and significance: This an important contribution to this body of literature. I would say more broadly that this paper does not focus on early childhood as it states but instead on infants/ toddler. (Line 164: 'Life-course of early childhood'). The reasons for the importance of these semantics is and why it may be important to make this differentiation is because of the very fluctuations in nutritional status the authors describe that could and have been documented to continue to fluctuate post the 2 years of age period. For example: Gausman, J.; Kim, R.; Subramanian, S.V. Stunting trajectories from post-infancy to adolescence in Ethiopia, India, Peru, and Vietnam. *Matern. Child Nutr.* 2019, 15, e12835

Response: In the revision we have been more careful about the nomenclature for the study's age range. We focused mainly on referring to children using specific age ranges rather than more ambiguous labels, which the reviewer notes can have different meanings to different readers.

We have updated Line 141-144 to say: *“A synthesis of longitudinal cohort data across diverse populations provides a unique opportunity to study the timing, dynamics, and burden of wasting in infants and young children — key knowledge gaps to inform future prevention efforts.”*

(3) Results & methods: The results presented will broadly be of immediate interest to many researchers working in the area of child growth and practitioners who focus their efforts on undernutrition prevention and treatment. This paper is especially relevant during a time when there is much debate about the best way to intervene when a child is wasted and/or stunted, discussions and funding put towards the treatment of moderate acute malnutrition programs and siloed programming for stunted and wasted children. Much of these efforts take place with primarily cross-sectional data informing them and thus this work is appropriately timed. Because of the multi-sectorality of nutrition as whole, other disciplines may well find use for the information contained in this paper.

Response: Thank you for your supportive views in this regard.

(4) The methods undertaken by the authors to answer their research questions of interest are appropriate. Reporting is transparent but more gained, from a readers point of view, from understanding the following:

There were some issues I had with the reporting of the interpretation of the mean WLZ (Figure 2a) in the outlined text in lines 203-208. It would perhaps to place more emphasis on the regional difference in WLZ and that mean WLZ in South Asia is well below zero reflecting, even if the same pattern, very different nutritional status early in infancy. This is also reflected in prevalence estimates reflected in Figure 2c.

Response: We have updated this passage to better highlight the regional differences in WLZ trajectories.

Lines 184-187: “Across all cohorts, mean WLZ was near -0.5 at birth and then increased over the first 6 months before decreasing until 12 months (Fig 2a). Age-specific patterns in WLZ were similar across geographic regions, but levels varied substantially by region. WLZ was markedly lower among South Asian cohorts, reflecting a much higher burden of malnutrition (Fig 2a).”

(5) Line 236-238: While this overall hypothesis is understandable, the authors do not provide ground truthing in each of these countries/ from study PIs if this is the best way to assess seasonal patterns. Agricultural lean periods in each of these regions could vary and not entirely be based on rainfall, for example, long period of arid temperature.

Response: In response to this comment and Referee 1 comment 4, we updated the seasonality analysis to use spatially-explicit rainfall from the cities/towns where cohorts were measured rather than national averages. Enhancement of the analysis to include more granular rainfall data served to strengthen the results of the analysis throughout — resulting in even more consistency between seasonal rainfall patterns and WLZ.

The high degree of consistency in the relationship between rainfall and WLZ across countries and over time is important because it means that rainfall provides a pragmatic, universal, and widely-measured indicator of population-level risk for seasonal malnutrition. We acknowledge that the causal mechanism between seasonal rainfall and child WLZ is likely mediated through several, inter-dependent pathways such as infectious disease transmission and crop yields/prices, which will be highly context specific. Moreover, as the referee notes, the relationship could be modified by other environmental drivers such as temperature/aridity. After discussing the analysis amongst the authorship team, we decided that expanding this work beyond an analysis of seasonal rainfall would constitute an entire article in itself and was thus beyond the scope of this work. We believe that the results constitute an important finding even without elucidating the full causal mechanism since rainfall data are widely available and could provide a useful guide to seasonal intervention timing, as it does for seasonal malaria chemoprevention (SMC) in Africa's Sahel region. We hope that the present synthesis motivates additional research into mechanisms — but it is quite possible that the mechanism varies depending on specific environmental, cultural, and socioeconomic context. We have updated the text to provide more nuance following the referee's suggestion:

Lines 309-313: *“The present study has not elucidated the mechanism that links seasonal rainfall with wasting, indeed mechanisms could differ by geographic and cultural context. Nevertheless, the high degree of consistency of the rainfall-wasting association provides strong support for the development of seasonally targeted interventions to prevent wasting in food-insecure populations, akin to seasonal malaria chemoprevention programs in the Sahel.”*

(6) Lines 254-255 state the high variability in wasting status of children from month to month but do not quantify it. Providing that range of variability here would help readers understand the justification of the imposed 60 day recovery period.

Response: We have updated the sentence to quantify variability of WLZ compared to LAZ and WAZ: *“There was high variability in WLZ across longitudinal measurements, with an average within-child standard deviation of WLZ measurements of 0.76, compared to 0.64 for LAZ and 0.52 for WAZ.”*

(7) Line 307-309 - This is a methodological issue and while stated in the limitations, explaining how this effects the studies results and interpretations to the reader is important. It is not quite enough to just point this out.

Response: In the revision we provide a summary of child measurement for each cohort included (new Extended Data Figure 3). The figure shows that nearly all included cohorts contributed data at nearly every age to these analyses. We acknowledge that 4 cohorts had few/no measures beyond 15 months. We therefore would not expect slight

differences in measurement completeness across cohorts to affect the main results or interpretations.

We have updated the sentence in the limitation on lines 169-174 to clarify the implications for the inference and interpretation:

“Most cohorts measured children every month through age 24 months, but there was some attrition as children aged and there were four cohorts with few measurements beyond 15 months (Extended Data Fig 3). Pooled estimates at older ages could be slightly biased if cohorts or children who were not measured were systematically different from those that were included — for example, if children who were lost were more likely to be wasted, we might have under-estimated wasting at older ages.”

(8) Lines 636-648 - there is no mention of controlling for time (year study was conducted) in the random effect models, was this done?

Response: We did not control for the calendar year of anthropometry measurement, but we have visualized child growth over calendar year (as studies measured child growth from 1987 to 2017) in the online supplementary materials (<https://child-growth.github.io/wasting/anthro.html>). There does not appear to be secular trends in Z-score measurements. We have also added cohort-specific estimates where possible throughout the figures to better highlight cohort-specific differences in estimates. We also have conducted

(9) The authors do not discuss issues of regression to the mean, an issue most longitudinal studies of anthropometry may experience. Was RTM assessed across longitudinal measurements especially among those extrema/ wasted children who 'recovered' during their next visit?

Response: We agree that with any correlated serial measurement, regression to the mean (RTM) is a factor, and our main strategy to address this was through design, following the best practice recommendation for measurement design to deal with RTM (<https://pubmed.ncbi.nlm.nih.gov/15333621/>). We required two measurements above the threshold to consider a child recovered, reducing the chance that we mistake statistical artifacts from RTM as true catch-up growth that reflect the biological process of recovering from wasting.

In response to this suggestion, we have presented a new analysis that summarizes the regression to the mean effect and shows the expected and observed mean Z-scores after three months among children who became wasted (new Extended data figure 8). Estimating RTM effects on a change in WLZ is tractable, but estimating the effect of

RTM on wasting recovery incidence would require a full simulation-based approach because, to our knowledge, there is no closed-form solution to estimate RTM for that particular measure (<https://pubmed.ncbi.nlm.nih.gov/8551887/>). We felt was beyond the scope of the present paper but would be a useful avenue for future research. We have added a sentence to the limitation section to discuss the potential impact of RTM:

Lines 250-254: *“Regression to the mean (RTM) could explain some wasting recovery, especially the rapid WLZ gain among children born wasted (Figure 4e), but there was catch-up growth beyond the calculated RTM effect (Extended Data Fig 8).⁴⁶ Additionally, we required a 60-day recovery period to better capture true recovery, and mean WLZ of recovered children did not regress to cohort means (Figure 4e), especially among older children (Figure 4c).”*

In the methods section, we discuss the methods for calculating the RTM effects shown in Extended data figure 8.

“We also assessed how much change in Z-scores after children became wasted could be explained by regression to the mean (RTM) and how much is catch-up growth beyond that expected by RTM.¹ We calculated the cohort-specific RTM effect and plotted the mean Z-scores expected from RTM 3 months later among children who were wasted at each age in months from birth to 21 months. We used the cohort means as the population mean Z-scores, and compared the expected mean WLZ with the observed mean WLZ (Extended Data Fig. 8).^{1,2”}

Extended data figure 8 is below, and shows catch-up growth beyond that expected from RTM alone in most cohorts, especially for children born wasted or who became wasted in the first month of life.

Extended Data Figure 8 | Regression to the mean effects in wasted children

(a) Expected mean Z-scores based on the regression to the mean effect (orange) and observed mean Z-scores (blue) 3 months after wasted children are measured. The lines connecting cohort-specific observed and expected WLZ at each age are colored orange if the expected estimate under RTM was higher than the observed mean (indicating lower than expected change in WLZ under RTM alone), and blue if the observed mean was higher than the expected estimate under RTM (indicating higher than expected change in WLZ under RTM alone). For examples, most cohorts experienced larger increases in WLZ than expected in the three-month period beginning in their first month of life (blue lines) and most cohorts experienced smaller increases in WLZ than expected in the three month periods beginning at ages 6-9 months (orange lines).

(b) Difference between observed means and expected means under a pure RTM effect by cohort, with the median differences by age indicated with horizontal lines.

Details on estimation of the RTM effects are in the methods.

- (10) Finally, there is no description of the populations being studied as far as I can tell anywhere. The average age of mothers at the time children were enrolled, food security status, sex distribution and so forth by study are not provided. Thus the conclusions while broadly on point leave the reader lacking the context of some statements such as those made in lines 329-330 about target populations of interest.**

Response: We have added Extended Data Table 2 to tabulate the following baseline characteristics by cohort (where measured): child sex, maternal age, education, and weight, child birthweight, number of rooms and number of children <5yrs in the household, improved sanitation presence, and household food security level.

- (11) Appropriate use of statistics and treatment of uncertainties: The authors switch back and forth between 95% CI and 95% credible intervals which may not be understood by readers at first glance. Perhaps a footnote as to why credible intervals vs confidence intervals would be of value?**

Response: We have updated the language throughout the manuscript to only refer to confidence intervals. The Z-score spline curves in figures 2a, 3a, 3c, and 4e have simultaneous credible intervals estimated around the curves to have better coverage than pointwise confidence intervals, but because we did not include informative priors they are interpretable as 95% simultaneous confidence intervals (Marra and Wood 2012, <https://onlinelibrary.wiley.com/doi/abs/10.1111/j.1467-9469.2011.00760.x>).

- (12) Finally, given the longitudinal nature of cohort included in this study and its analyses, the authors do not provide as much insights to the variability between children within regions. Again here, interpretations and conclusions could potentially be more rich if this is done.**

Response: We have added additional discussion and visualization of the differences in wasting between children within regions. In Figure 4b, we have added inset histograms plotting the number of wasting episodes by region, showing children in South Asia have the highest mean number of episodes (0.74) and a larger proportion of children with 2 or more episodes of wasting across the first two years of life. This aligns with the higher incidence rate of wasting shown in the main figures, and is referenced on lines 236-238:

“The mean number of wasting episodes experienced and wasting incidence at all ages was higher in South Asia, with highest incidence in the first 3 months with- or without- including episodes at birth (Fig 4b).”

We have updated the discussion on lines 292-305:

“Nearly all large-scale studies of child wasting including DHS report point prevalence of wasting, which has enabled broad comparisons between populations but failed to capture the number of episodes of wasting during the course of childhood. By combining information across several longitudinal cohorts, this study demonstrated that children experience far higher cumulative incidence of wasting between birth and 24 months (29.2%) than previously known. Children in all included cohorts experienced higher incidence compared with prevalence due to the episodic nature of acute malnutrition, but the pattern was most stark in South Asian cohorts where, from birth to 24 months and pooled across cohorts, 29.2% of children experienced at least one wasting episode, 3.4% were persistently wasted, and 10.6% had experienced concurrent wasting and stunting. This evidence shows that the cumulative, child-level burden of wasting is higher than cross-sectional prevalence measures would suggest, and assessments of wasting and stunting in isolation fail to account for their joint burden, which can be substantial. This is of particular relevance in South Asia where the largest number of stunted and wasted children live, and while stunting prevalence is decreasing, wasting prevalence is not.”

Suggested improvements:

(13) Line 117: What does unrealistic mean? Please expand.

Response: we have update “unrealistic” to “biologically implausible” to reflect the language in the WHO 2006 child growth standards “Recommendations for data collection, analysis and reporting on anthropometric indicators in children under 5 years old” (<https://www.who.int/nutrition/publications/anthropometry-data-quality-report/en/>).

Biologically implausible values are Z-score values that are not generally compatible with life. From section 3.1.5 “Implausible z-score values”:

“Implausible values are z-score values that fall outside a specified range. The currently recommended flagging system to detect implausible z-score values was defined in 2006 on the release of the WHO Child Growth Standards, replacing the NCHS/WHO reference for child growth6 (see discussion on flagging in section 3.2 on Data analysis). System cutoffs were defined on the basis of what is biologically implausible, in other words incompatible with life. These flagging cut-offs have been challenged based on

observations of living children whose z-scores are beyond currently defined implausible values (16), although true z-score values beyond the implausible value cut-offs recommended by the WHO rarely occur in any population. Nonetheless, this is a topic for future research.”

(14) Lines 258-260: Incidence of what? Wasting? Or recovery from wasting?

Response: Wasting incidence. We have updated the sentence to clarify:
“The mean number of wasting episodes experienced and wasting incidence at all ages was higher in South Asia, with highest incidence in the first 3 months with- or without- including episodes at birth (Fig 4b).”

(15) Lines 294-296: Was the converse also applicable? Were children born stunted at higher risk of wasting later within the first 2 years?

Response: We did not specifically examine the association between birth wasting/stunting and the risk of later stunting or wasting in this manuscript, but the companion manuscript (“Risk factors and impacts of child growth faltering in low- and middle-income countries”) examined different measures of early growth failure and later growth failure. We found that all different combinations of wasting, stunting, and underweight were significantly associated with an increased risk of persistent wasting between 6 and 24 months and concurrent stunting and wasting at 18 months.

Here, among the monthly-measured cohorts analyzed in this manuscript, children who experienced wasting in the first 6 months had an increased risk of concurrent wasting and stunting at 24 months of age (RR: 1.9, [95% CI: 1.5, 2.4]). We updated the analysis to also examine the association between stunting in the first 6 months and concurrent wasting and stunting at 24 months of age, and found an even stronger association (RR: 2.9, [95% CI: 2.5, 3.3]).

(16) Lines 305-307- To what extent did the authors consider timing (which month for example the measurement was taken) of anthro measurement when comparing DHS distribution and study cohorts included in this study?

Response: We did not adjust anthropometry measurements for the time of year the measurements were taken.

There is a risk of dry-season bias in the DHS data, with prior research finding less implementation in wet seasons than expected (“Do international surveys and censuses exhibit ‘Dry Season’ bias?” <https://onlinelibrary.wiley.com/doi/abs/10.1002/psp.681>), so it is possible that the DHS surveys overestimate Z-scores if they vary considerably by season, as we have shown is likely for WLZ in populations with highly seasonal rainfall

and likely food insecurity. It was beyond the scope of this analysis to investigate seasonality in the DHS data.

We compared DHS estimates to the ki cohorts to assess whether included cohorts were reasonably representative of similar measurements in population-based surveys, or if they were enrolled from populations that were smaller/less healthy than the general population, affecting the generalizability of the results. We found that the ki-cohort mean LAZ was generally higher in DHS surveys from Bangladesh, India, Guatemala, Nepal, and Pakistan, but were roughly equal in Peru, South Africa, or Tanzania (Extended data figure 4). WLZ scores were generally higher in DHS datasets from Guatemala, Pakistan, and South Africa, mostly equal in Bangladesh, Nepal, and Tanzania, and lower in India and Peru.

If there was a strong dry-season bias in DHS data but not the longitudinal ki data, we'd expect the gap between WLZ in the DHS data and in the ki data to be larger than the gap between LAZ in the DHS and ki data, and therefore believe there isn't a strong seasonal bias in the DHS data we included in the analysis. Additionally, other studies analyzing wasting in DHS data across all LMIC countries found that seasonal adjustment had relatively little effect on the raw data. (<https://www.nature.com/articles/s41586-019-1878-8>)

(17) Lines 307: What is the resultant issue of the limitation of children being measured at different times? This is not explained by the authors in the discussion and is important to expand upon in a paper so heavily focused on anthropometry.

Response: Please also see our earlier response to Referee #1 comment 7, which includes additional details. We have updated the sentence in the limitation on lines 169-174 to read:

“Most cohorts measured children every month through age 24 months, but there was some attrition as children aged and there were four cohorts with few measurements beyond 15 months (Extended Data Fig 3). Pooled estimates at older ages could be slightly biased if cohorts or children who were not measured were systematically different from those that were included — for example, if children who were lost were more likely to be wasted, we might have under-estimated wasting at older ages.”

(18) Lines 314-316 - Did the cohort studies not have mortality data to assess this bias?

Response: Of 21 cohorts included in the detailed wasting analyses, 11 had mortality data. Of the 11 cohorts that had mortality data, all had fewer than 5 deaths except the PROVIDE study (10 deaths) and the MRC Keneba study (100 deaths) and not all

recorded deaths had an exact age-of-death. Given the limited information available regarding mortality in the included cohorts, survivorship bias is hard to accurately assess. However, the very low death rate among cohorts with mortality measurements (described in more detail in the companion “Causes and consequences...” article), suggests survivorship bias was not a major source of bias in the analyses. We have therefore removed the sentence in the revision.

(19) The section on the overlap between wasting and stunting does not seem adequately flushed out. Not many insights are provided either in the results or discussion about these findings and their implications.

Response: We have revised the section on concurrent wasting and stunting to better flesh out key results. We have further added a new sentence to the Discussion to comment on implications.

Revised text, Lines 274 - 289

“Next, we examined the cumulative incidence of concurrent wasting and stunting and the timing of their overlap. Overall, 10.6% of children experienced concurrent wasting and stunting before 2 years of age (Extended data figure 5d), and a further 1.5% experienced concurrent severe wasting and stunting — far higher than point prevalence estimates in the present study (4.1% at 24 months, 95% CI: 2.6, 6.4; Fig 5a) or the 4.3% prevalence estimated in children under 5 in LMIC from cross-sectional surveys. Concurrent wasting and stunting was most common in South Asia, with highest prevalence at age 21 months (Fig 5a), driven primarily by increases in stunting prevalence as children aged. Children wasted and stunting are also underweight, as their maximum possible weight-for-age z-score is below -2.35. Almost half of children who had prevalent growth faltering met two or more of these three growth faltering conditions, with 17-22% of all children experiencing multiple conditions after age 12 months (Fig 5b). Longitudinal analyses showed early growth faltering predisposed children to experience concurrent growth faltering at older ages: children who were wasted by age 6 months were 1.8 (95% CI: 1.6, 2.1) times and children stunted before 6 months were 2.9 (95% CI: 2.5, 3.4) times more likely to experience concurrent wasting and stunting between ages 18-24 months. A companion article reports an in depth investigation of key determinants of persistent wasting and concurrent wasting and stunting, along with their consequences for severe growth faltering and mortality.”

Revised Discussion, Lines 292 - 305

“Nearly all large-scale studies of child wasting including DHS report point prevalence of wasting, which has enabled broad comparisons between populations but failed to capture the number of episodes of wasting during the course of childhood. By combining information across several longitudinal cohorts, this study demonstrated that children experience far higher cumulative incidence of wasting between birth and 24 months (29.2%) than previously known. Children in all included cohorts experienced higher incidence compared with prevalence due to the episodic nature of acute malnutrition, but the pattern was most stark in South Asian cohorts where, from birth to 24 months and pooled across cohorts, 29.2% of children experienced at least one wasting episode, 3.4% were persistently wasted, and 10.6% had experienced concurrent wasting and stunting. This evidence shows that the cumulative, child-level burden of wasting is higher than cross-sectional prevalence measures would suggest, and assessments of wasting and stunting in isolation fail to account for their joint burden, which can be substantial. This is of particular relevance in South Asia where the largest number of stunted and wasted children live, and while stunting prevalence is decreasing, wasting prevalence is not.”

(20) Figure 3a: There appears to be some variation in seasonal patterns that are not necessarily congruent with the explained results. For example, it appears that in the MAL-ED Peru sample and the ResPark sample from Pakistan, the mean WHZ is not necessarily lowest during periods of highest rainfall. Encourage further description and details of these findings.

Response: Based on this comment, Referee 1 comment 4, and Referee comment 5, we have updated our approach to the seasonality analysis to use spatially-explicit rainfall from the cities/towns where cohorts were measured rather than national averages. See the response to Referee 1 comment 4 for more details on how the methodological changes affected the results and on variation between cohorts.

(21) The dramatic shifts in WLZ by season but larger fluctuations in WLZ by birth month in South Asia noted by the authors ought to be flushed out in the manuscript.

Response: Within tight word constraints, we have done our best to expand on these findings. In the Seasonality of Wasting section, we have devoted a paragraph to highlight the WLZ by birth month in South Asia results:

Lines 222 - 228

“South Asian cohorts had temporally synchronous rainfall, so we estimated WLZ at birth by calendar month, pooled across the 11 South Asian cohorts. Mean WLZ at birth varied by up to 0.72 Z (95% CI: 0.43, 1.02) depending on the month the child was born (range: -0.5 Z to -1.3 Z; Fig 3c), which suggests that seasonally-influenced maternal

nutrition, likely mediated through intrauterine growth restriction or preterm birth, was a major determinant of child WLZ at birth. Birth month also influenced the effect of season on WLZ trajectories that persisted through a child's second year of life (Extended Data Fig 7)"

We further expanded on these results and their implications in the Discussion:

Lines 306 - 313

"Our results show consistent patterns in the timing of wasting onset by season and by age, with important implications for preventive interventions. WLZ varied dramatically by season in most cohorts, and in South Asia WLZ varied by up to 0.7 z-scores at-birth with consequences that persisted throughout the first 24 months (Fig 3c, Extended Data Figure 7). The present study has not elucidated the mechanism that links seasonal rainfall with wasting, indeed mechanisms could differ by geographic and cultural context. Nevertheless, the high degree of consistency of the rainfall-wasting association provides strong support for the development of seasonally targeted interventions to prevent wasting in food-insecure populations, akin to seasonal malaria chemoprevention programs in the Sahel."

(22) Extended Data Table 1009 1| Summary of ki cohorts: There are formatting issues with this table.

Response: We have fixed the formatting issues.

Referee #3 (Remarks to the Author):

1. Summary

a. What makes wasting a "leading" indicator?

Response: We have revised the sentence to clarify that wasting is used as a key indicator to measure the success of achieving Sustainable Development Goal 2.2.2:

Lines 100-102

"Sustainable Development Goal 2.2.2, to end malnutrition by 2030, measures progress through elimination of child wasting, defined as weight-for-length more than 2 standard deviations below international standards."

b. Overall very good summary – the intro should be crafted to support the analysis performed in this paper. Currently it comes across as only tangentially connected.

Response: We have revised the summary introduction so that it more directly connects to the paper. The revised introduction sentences are:

Lines 119-133

“Wasting is a form of undernutrition that results from a loss of muscle and fat tissue from acute malnutrition, affecting an estimated 45.4 million children under 5 years (6.7%) worldwide with over half living in South Asia. Children are considered wasted if their weight-for-length z-score (WLZ) falls 2 standard deviations below the median of international standards. Wasted children have weakened immune systems, predisposing them to infections and more severe illness once infected. Wasting in very young children increases their risk of mortality (relative risk: 2.3), with mortality risk further increasing if they are also stunted and underweight, defined as length-for-age (LAZ) below -2 and weight-for-age (WAZ) below -2 (relative risk: 12.3). Longitudinal studies have shown that wasting is also associated with higher risk of stunting⁶ and poor neurocognitive development at older ages. Treatment of severely wasted children (WLZ below -3) with high calorie, ready-to-use-therapeutic foods has proven effective, but rates of relapse and mortality remain high in this fragile subgroup. Primary prevention of wasting would be preferable to treatment, but there are scant proven preventive interventions. Key among them, nutritional interventions have led to only small reductions in wasting, with breastfeeding support from birth not reducing wasting and small-quantity lipid-based nutrient supplementation between ages 6-24 months leading to a 14% relative reduction in wasting.”

c. What happened to stunting? Its in the title, intro, and toward the end of the paper, but missing from summary.

Response: We have more prominently discussed concurrent wasting and stunting in the revised summary by adding this sentence:

Lines 109-110

“Children wasted before 6 months were more likely to experience concurrent wasting and stunting (low height-for-age) later, increasing their risk of mortality.”

2. Introduction, especially the first paragraph, is quite poorly written. It is repetitive, meandering, and poorly cited. The concepts are fine, but the presentation needs a lot of work, including the choice and placement of citations.

Response: We have rewritten the two introductory paragraphs in their entirety to improve their presentation and clarity.

3. I asked this for the other papers as well, but why stop at 24 months? Don't these data sources continue beyond that? Seeing as there appears to be acculamation of faltering with age, stopping at age 2 seems like an odd choice.

Response: The ki data collation effort was principally focused on child growth during the first 1000 days of life. Among 21 cohorts that met our inclusion criteria for this study, only two cohorts from India and a small subset of children from the MAL-ED studies followed children beyond age 24 months with monthly resolution. Below is the histogram

of child ages from each of the monthly-measured cohorts, with the vertical line denoting age 24 months. Observations to the left of the vertical line were included in the analysis.

We added this clarification to the manuscript:

Lines 163 - 164

“Most included cohorts did not measure children past 24 months, so analyses focused on birth to 24 months.”

4. Line 172-4 this hypothesis is not really a hypothesis. It is essentially stating that “we hypothesize we will learn something new.” Please either drop this sentence or reframe as to what the actual hypotheses were. Also, the title of the paper and the intro mention stunting. Is that not part of they hypotheses or analysis?

Response: This point is well taken. We have dropped the use of “hypothesis” in the revised sentence. We have also emphasized stunting. The revised sentences are:

Lines 149 - 152

“Our primary objectives were to produce the first large-scale estimates of wasting incidence and recovery and to quantify the temporal and regional variation. We also assessed the concurrence of wasting and stunting and compared estimates of wasting prevalence and incidence.”

5. Line 176 and 184 – similar to the other papers, how did the dataset go from millions to <11k? And why so many fewer than the “causes and consequences” paper that presumably was done from the same dataset(s)?

Response:

The original sentence referring to millions of children included cross-sectional datasets and data from high-income and historic cohorts. We revised this sentence to clarify

Lines 152 - 153

“We analyzed data from the Bill & Melinda Gates Foundation’s Knowledge Integration (ki) database, which has aggregated studies on child growth and development.”

There are fewer children and observations in this wasting manuscript because we subset the analyses to cohorts that measured children every month (median gap between followup measurements 31 days or less), which tended to have smaller sample sizes overall than cohorts with less frequent measurements. We restricted wasting analyses to more frequently measured cohorts because of the more acute, dynamic nature of WLZ and wasting and because of the number of analyses focused on patterns of wasting incidence and recovery.

We noted this in the inclusion criteria, relevant text in bold:

Lines 153 - 158

“Inclusion criteria were defined to select cohorts representative of general populations in LMICs with sufficiently frequent measurements to investigate the acute, dynamic nature of wasting: 1) conducted in low- or middle-income countries; 2) enrolled children between birth and age 24 months and measured their length and weight repeatedly over time; 3) did not restrict enrollment to acutely ill children; 4) had median birth year of 1990 or later 5) measured anthropometry at least monthly (Extended Data Fig 1).”

The “causes and consequences” paper included studies with at least quarterly (~3 month) anthropometry measurement frequency, which included many large trials. Extended data figure 1 shows which studies were excluded from this manuscript for not being monthly-measured but are included in the stunting and “Risk factors and impacts of child growth faltering in low- and middle-income countries” manuscripts for meeting all other inclusion criteria.

6. Line 197-201 – either need to define WAZ and LAZ or reframe this part about Asia/Africa cohorts to be consistent with the description for LatAm. Also, the finding that LAZ was steady in first 6 months in DHS is suspect.

Response: In response to Referee 4 comment 7c, we separated the DHS analyses by country, and updated the section of the manuscript to reflect this change. Country-stratified figures are presented in Extended Data Figure 4. We believe the overall steady LAZ in the first 6 months of life in the included DHS countries is accurate. In the country-specific estimates, LAZ remained relatively steady for the first 6 months in The Gambia, Peru, Bangladesh, and India, and increased in South Africa and Pakistan. Another study that examined LAZ in DHS data also found very little faltering in the first 6 months (<https://pubmed.ncbi.nlm.nih.gov/29132614/>). We have revised the passage to

discuss country-specific differences and dropped the description of steady LAZ in the first 6 months because that pattern is less consistent when examining LAZ by country instead of region.

Lines 175 - 181:

“We assessed ki cohort representativeness by comparing Z-score measurements to population-based samples in Demographic and Health Surveys (DHS). Mean z-scores in ki cohorts were generally representative based on country-level DHS estimates, with lower WLZ (overall and by age) in Guatemala, Pakistan, and South Africa and higher WLZ in Brazil and Guatemala (Extended Data Fig 4). LAZ was generally lower in ki cohorts, so estimates of concurrent wasting and stunting may be higher than estimates from population-based samples, and rates of wasting recovery in early life may be higher compared with the general population.”

7. On the topic of comparisons to DHS (extended data fig 3), the description and performance of these comparisons could be improved:

7a. need to be more specific with which DHS they were matched. Same year only? +/- 5 years? Seasonality adjusted in DHS?

Response: We used the most recent DHS estimates for each country, so the DHS surveys are not matched by year to the ki cohorts (which vary in the year collected in each country) and they are generally more recent. We did not seasonally adjust the DHS data, as another study analyzing wasting in DHS data across all LMIC countries found that seasonal adjustment had relatively little effect on the raw data.”

<https://www.nature.com/articles/s41586-019-1878-8>

We updated the methods section on the DHS analysis to clarify the methods

Lines 894-901:

“We downloaded standard DHS individual recode files for each country from the DHS program website (<https://dhsprogram.com/>). We used the most recent standard DHS datasets for the individual women’s, household, and length and weight datasets from each country, and we estimated age-stratified mean LAZ, WAZ, and WLZ from ages 0 to 24 months within each DHS survey, accounting for the complex survey design and sampling weights. See Benjamin-Chung et. al (2020) for additional details on the DHS data cleaning and analysis.⁴ We compared DHS estimates with mean LAZ, WAZ, and

WLZ by age in the ki study cohorts with penalized cubic-splines with bandwidths chosen using generalized cross-validation.⁸ We did not seasonally adjust DHS measurements. ”

7b. Were they matched to the specific location where the KI studies were performed within each country? Or only the summary country-level results from each DHS?

Response: We only used the summary country-levels results from each DHS survey because we were interested in whether the patterns of Z-scores in *ki* cohorts differed from nationally-representative estimates. Also, in many cases the DHS sampling areas might not overlap with *ki* cohort locations.

7c. Need to match study-by-study and present that information, not pooled across entire continents.

Response: We have updated the Extended Data Figure 4 to present country-specific DHS estimates compared to *ki* cohort-specific estimates (updated figures above in the response to comment #6).

8. It is not helpful – and potentially counterproductive – to make regional estimates at a time, and for a topic, when most health information will need to be as local as possible. The settings across regions are very diverse and, even within regions, they are very different. It would be more instructive and helpful to present data and results for each study individually and then show the pooled results across the studies for an overall. This is already done for Figure 1 and Figure 3 and should be repeated for figures 2, 4, and 5.

Response: We agree that cohort-specific estimates are valuable. We updated figures 2, 4b-d, 4f, 5a, and Extended Data Figure 5 to include cohort-specific point estimates in light grey behind the pooled estimates whenever possible based on figure design. This enables readers to visualize between-study heterogeneity for all pooled estimates, including those summarized by geographic region. For some visualizations, such as Figure 5b, we were unable to include cohort-specific estimates due to the type of plot. In addition, we included cohort-specific versions of each primary figure in the online supplementary material (<https://child-growth.github.io/wasting/cohort.html>).

We retained regional estimates because many nutritional programs, policy makers, and funders make decisions on a regional basis. Regional-specific estimates were consistently requested throughout the development of the analysis plan across the study consortium, including members throughout Africa, Latin America, and South Asia.

9. Line 204 – mean WLZ near zero at birth? This seems unlikely unless there were few low birthweight or preterm births in the cohorts. Do you have that information? Also, figure 2 does not support this statement. Mean WLZ at birth was -0.8 in South Asia, -0.5 in Africa, and +0.5 in LatAm.

Response: We have revised this sentence to be more precise:

“Across all cohorts, mean WLZ was near –0.5 at birth and then increased over the first 6 months before decreasing until 12 months (Fig 2a). Age-specific patterns in WLZ were similar across geographic regions, but levels varied substantially by region. WLZ was markedly lower among South Asian cohorts, reflecting a much higher burden of malnutrition (Fig 2a).”

We have data on infant birthweight and preterm births from some, but not all, cohorts, and analyze the effect of correcting for gestational age in Extended Data Figure 6. However, we only have gestational age data from 4 cohorts, where preterm birth prevalence ranged from 11-16%, and early term births ranged from 23-36%. None of these four cohorts are from Latin America, though, where at-birth WLZ was highest (pooled mean of 0.14). Twenty of twenty-one cohorts had birth weight data, and 13% of children were low birth weight.

10. Line 209 – need to define “severely wasted” in the main text somewhere.

Response: We have defined severe wasting in the revised introduction:

Lines 128 - 129

“Treatment of severely wasted children (WLZ below –3) with high calorie, ready-to-use-therapeutic foods has proven effective, but rates of relapse and mortality remain high in this fragile subgroup.”

11. Line 211 – random effects pooled prevalence of wasting at birth is not meaningful across such diverse settings, especially in a section about “regional differences.”

Response: We have removed the specific overall pooled wasting prevalence at birth from the sentence. However, we believe there is value in reporting pooled estimates by age to examine summary age-specific patterns across cohorts overall and within geographic regions. We have added cohort-specific point estimates to Fig 2b to show between-study heterogeneity that underlie the region-specific means.

12. Figure 2 –

12a. the text description of the figures is inaccurate. The methods state this is a spline with 95% UI, but the figure text states these are percentiles. Which is it?

Response:

We have updated the methods, including updating the figure method description, to clarify. Fig 2a has also been updated to add cohort-specific estimates as well to address other comments.

Updated Fig 2a caption:

“(a) Mean weight-for-length Z-score (WLZ) by age in 21 longitudinal cohorts, overall (N=21 studies; N=4,165-10,886 observations per month) and stratified by region (Africa: N=4 studies, N=1,067-5,428 observations; Latin America: N=6 studies, N= 569-1718 observations, South Asia 11 studies, N=2,382-4,286 observations). ”

Updated methods:

Lines 858 - 861

“Fitted smoothers used in manuscript figures were fit using cubic splines and generalized cross-validation.⁶ We estimated approximate 95% simultaneous confidence intervals around the cubic splines using a parametric bootstrap that resampled from the posterior the generalized additive model parameter variance-covariance matrix.⁷”

12b. All of these figures that show overall, Africa, LatAm, and S Asia are all stated to be random effects meta-analyses. If that is the case, they should be presented as such with all the data and the pooled effect. Right now only the results are shown which, when pooled across such diverse places, cannot be interpreted readily.

Response: We have updated the figures throughout, when possible, to include cohort-specific estimates in a lighter shade behind the pooled estimates. See the response to Reviewer 3, comment 8 for more details on the plot updates.

13. Line 224-226 – this sentence is misleading. The two strongest factors in the cited study influencing birthweight were newborn LAZ and duration of pregnancy. In turn, the three biggest factors were duration of pregnancy, placental weight, and maternal height. Placental weight can be influenced by acute illness and nutrition, but in other large studies (such as this one - <https://pubmed.ncbi.nlm.nih.gov/24354883/>) has also been shown to be strongly correlated with chronic health conditions such as hypertension, pregnancy complications such as preeclampsia, and fetal abnormalities. The conclusion of Ashorn and colleagues’ analysis is that constitutional factors play an outside role. That conclusion is not reflected in the text here. The text should be revised accordingly and, ideally, re-contextualized. Maternal height is extensively explored in one of the companion papers to this one – can any of that insight be brought to bear here?

Response: These are great suggestions and we revised the sentence to avoid any confusion. In the companion paper, we include more discussion about prenatal versus postnatal risk factors, but due to space constraints we have revised the sentence to highlight the spectrum of prenatal risk factors that may drive high South Asian infant wasting (mediated through preterm/low birth weight):

Lines 191 - 195

“In South Asian cohorts, where low birthweight is common, 37 at-birth wasting prevalence was 18.9% (95% CI: 15.0, 23.7), implicating causes of poor fetal growth like maternal malnutrition, maternal morbidities, and maternal small stature as key regional drivers of wasting. Severe wasting followed a similar pattern but was much rarer (Extended Data Fig 5).”

14. The sections on “Wasting incidence and recovery” and “persistent and concurrent growth failure” are, in my opinion, the only novel insights of this paper. The paper could be substantially strengthened by focusing much more, perhaps almost exclusively, on those two aspects of the analysis, with much of the rest relegated to supplementary material.

Response: We agree that the analysis of wasting incidence, wasting persistence, and its relationship with stunting are novel insights made possible through analyses of the multi-cohort, longitudinal data. In the revision we reduced space devoted to “Age specific patterns of wasting” by an entire paragraph and focused only on new insights afforded by the longitudinal analyses. We have not shortened the remaining section, “Seasonality of wasting”, because although a small number of previous studies in individual cohorts have documented seasonality in wasting, the strength and consistency of the relationship between seasonal rainfall patterns and WLZ across 16 cohorts in this analysis provides a strong evidence base from which to consider seasonally-targeted interventions. We have updated the discussion on lines 292-305 to discuss the novelty of our findings:

“Nearly all large-scale studies of child wasting including DHS report point prevalence of wasting, which has enabled broad comparisons between populations but failed to capture the number of episodes of wasting during the course of childhood. By combining information across several longitudinal cohorts, this study demonstrated that children experience far higher cumulative incidence of wasting between birth and 24 months (29.2%) than previously known. Children in all included cohorts experienced higher incidence compared with prevalence due to the episodic nature of acute malnutrition, but the pattern was most stark in South Asian cohorts where, from birth to 24 months and pooled across cohorts, 29.2% of children experienced at least one wasting episode, 3.4% were persistently wasted, and 10.6% had experienced concurrent wasting and stunting. This evidence shows that the cumulative, child-level burden of wasting is higher than cross-sectional prevalence measures would suggest, and assessments of wasting and stunting in isolation fail to account for their joint

burden, which can be substantial. This is of particular relevance in South Asia where the largest number of stunted and wasted children live, and while stunting prevalence is decreasing, wasting prevalence is not”

15. The authors’ discussion of seasonally-targeted interventions is compelling as well although the seasonality finding is not novel. I would like to see how these seasonality patterns compare to those in the matched DHS studies.

Response: We have not analyzed seasonality in the matched DHS studies because we felt such analyses would be beyond the scope of this paper. Introduction of DHS data to this analysis was to assess the representativeness of the *ki* cohort anthropometry measurements vis-a-vis population-based samples. DHS surveys vary considerably in their timing vis-a-vis calendar date and in their availability of geographic coordinates for sampling clusters. The analysis complexity arising from any type of seasonal analyses of cross-sectional DHS surveys would warrant a separate publication in its own right. In response to Referee 1 comment 4, we enhanced the seasonality analysis in the revision so that it relies on more granular, spatio-temporally explicit rainfall data, which further strengthened the seasonality analysis.

16. Figure 5 and “persistent and concurrent growth failure” – multiple questions/comments:

16a. the world nutrition targets 2025 only list WLZ (wasting) and LAZ (stunting) as targets. The authors also largely “ignore” WAZ (underweight) until this figure. My question is, why include here at all?

Response: We included WAZ (underweight) in Figure 5b for completeness across categories. We felt that a sole focus on WLZ (wasting) and LAZ (stunting) in Fig 5b would potentially mask the full spectrum of growth failure by age displayed in the figure. Additionally, the companion paper “Causes and Consequences...” includes some assessment of WAZ (underweight) with respect to its association with subsequent mortality — both alone and in conjunction with other growth failure measures. Our rationale for including WAZ in the mortality analyses is that it is under active consideration by global malnutrition programs to screen infants at high risk of mortality because weight is generally simpler than measuring child length.

16b. Is there any evidence that wasting + underweight is any worse than wasting alone? Or is it just a manifestation of more severe wasting? Put another way, there is ample evidence that both WLZ and LAZ are independently modifiable risk factors for poor outcomes in children (death and developmental) – is there any evidence that WAZ is either an independent risk factor or is independently modifiable? Or is it entirely mediated through the health effects of wasting and stunting? (McDonald et al did not evaluate severity of stunting, wasting, and underweight so citing their findings here would not be particularly helpful.) This

may not be entirely answerable within this paper, but this is a vitally important question that is thematically missing from the companion papers and needs to be strengthened.

Response:

We present evidence in the revised companion paper “Causes and Consequences...” to address this comment as part of a sensitivity analysis. While we cannot look at whether WAZ is independently associated with disease outcomes with the data we have, as WAZ is a composite measure that doesn’t distinguish between wasting and stunting. A child with low weight for her age could be either short with more fat mass or tall and thin. But we did assess associations with mortality in a subset of studies that measured it. In time-to-event, Cox proportional hazard models, underweight status alone (so children who were underweight, but not stunted or wasted) was associated with mortality (HR: 2.63; 95% CI: 1.29, 3.59). Any underweight (with or without other measures of growth failure) had a stronger association with mortality (HR: 3.41; 95% CI: 3.31, 3.50). After controlling for wasting status in the model, the association was attenuated but remained significant (HR: 1.74; 95% CI: 1.67, 1.81). After controlling for stunting status in the model, the association was less attenuated (HR: 2.83; 95% CI: 2.73, 2.94). This implies that being underweight can an independent risk factor for mortality, though underweight children may have suboptimal LAZ and WLZ, that is not < -2 but still carries a risk. We discuss this nuance more in the “Causes and consequences...” paper. In addition to the independent associations with mortality even after controlling for wasting and stunting status, underweight status and WAZ are useful to study because of the potential programmatic use. We found in our companion paper that underweight children were at higher risk of death than wasted or stunted children, and WAZ is easier to measure, especially in infants, because it only requires weight, potentially making it a more useful screening tool for therapeutic feeding programs.

16c. I would think it potentially more instructive – and novel – to a) evaluate overlap between moderate, severe stunting and moderate, severe wasting and see if that “gets rid of” underweight entirely and b) investigate how this overlap of concurrent growth failure categories compare to the overlaps in matched DHS studies?

Response: Figure 5b summarizes mutually exclusive combinations of different measures of child growth failure by age and shows that few children (2.1% overall) are underweight alone (without wasting or stunting). Because few children are only underweight, and because WAZ is a combination of LAZ and WLZ (as weight is the sum of lean tissue, fat tissue, water and length of skeleton, and LAZ capture the latter, and then weight independent of length (WLZ) captures the rest), we focus less on underweight in this manuscript. But believe it is important to include the overlap between wasting, stunting, and underweight in this manuscript because of the increased risk of death associated with multiple anthropometric deficits (examined in the

companion manuscript “Risk factors and impacts of child growth faltering in low- and middle-income countries”).

Below, we tabulated the overlap between moderate and severe wasting, stunting, and underweight in the tables for the ki cohorts and matched DHS countries. In the ki cohorts, 2.1% of all measures were underweight but not wasted or stunted. The similar percentage was nearly identical in the DHS surveys: 1.9%.

Because of the correlations between WAZ, WLZ, and LAZ, based on the WHO growth reference charts, all children who are both wasted and stunted will also be underweight, and based on our data, all severely underweight children ($WAZ < -3$) will also be at least moderately stunted or wasted.

There is existing research using a large set of cross-sectional data to extensively evaluate the overlap between wasting, stunting, and underweight (<https://archpublichealth.biomedcentral.com/articles/10.1186/s13690-018-0277-1>), so we believe the most novel contribution of this research is the longitudinal nature of the cohorts that allow us to study the joint relationship between wasting and concurrent stunting (Figs 4e, 4f, 5a, 5b). We have updated the “Persistent and concurrent growth failure” section on lines 274-289 to discuss the overlap and longitudinal joint relationship between measures of child growth failure:

“Next, we examined the cumulative incidence of concurrent wasting and stunting and the timing of their overlap. Overall, 10.6% of children experienced concurrent wasting and stunting before 2 years of age (Extended data figure 5d), and a further 1.5% experienced concurrent severe wasting and stunting — far higher than point prevalence estimates in the present study (4.1% at 24 months, 95% CI: 2.6, 6.4; Fig 5a) or the 4.3% prevalence estimated in children under 5 in LMIC from cross-sectional surveys. Concurrent wasting and stunting was most common in South Asia, with highest prevalence at age 21 months (Fig 5a), driven primarily by increases in stunting prevalence as children aged. Children wasted and stunting are also underweight, as their maximum possible weight-for-age z-score is below -2.35. Almost half of children who had prevalent growth faltering met two or more of these three growth faltering conditions, with 17-22% of all children experiencing multiple conditions after age 12 months (Fig 5b). Longitudinal analyses showed early growth faltering predisposed children to experience concurrent growth faltering at older ages: children who were wasted by age 6 months were 1.8 (95% CI: 1.6, 2.1) times and children stunted before 6 months were 2.9 (95% CI: 2.5, 3.4) times more likely to experience concurrent wasting and stunting between ages 18-24 months. A companion article reports an in depth investigation of key determinants of persistent wasting and concurrent wasting and stunting, along with their consequences for severe growth faltering and mortality”

Tables of percent of children in mutually exclusive combinations of child growth failure in KI cohorts and DHS data.

Percentage of all measurements of children under 2 with different non-overlapping combinations of child growth failure. For example, children were moderately underweight (WAZ < -2 and > -3) and severely stunted (LAZ < -3) but not wasted in 2.91% of all measurements.

KI studies:

Wasting	Stunting	Underweight		
		Not underweight	Mod. underweight	Sev. underweight
Not wasted	Not stunted	66.95	2.13	0.00
	Mod. stunted	10.98	4.96	0.22
	Sev. stunted	1.96	2.91	1.36
Mod. wasted	Not stunted	1.69	2.35	0.12
	Mod. stunted	0.00	0.79	0.79
	Sev. stunted	0.00	0.01	0.96
Sev. wasted	Not stunted	0.29	0.45	0.40
	Mod. stunted	0.00	0.00	0.36
	Sev. stunted	0.00	0.00	0.32

DHS data (children under 2 from the same set of countries as the KI cohorts):

Wasting	Stunting	Underweight		
		Not underweight	Mod. underweight	Sev. underweight
Not wasted	Not stunted	50.76	1.93	0

	Mod. stunted	10.4	4.86	0.23
	Sev. stunted	4.31	4.55	1.96
Mod. wasted	Not stunted	5.16	4.04	0.18
	Mod. stunted	0	0.92	1.13
	Sev. stunted	0	0.01	1.45
Sev. wasted	Not stunted	2.3	2.39	1.71
	Mod. stunted	0	0	0.95
	Sev. stunted	0	0	0.75

17. Line 257-260 and Line 587-590 – what is the validity of this assumption? How many of those who had birthweight measured and who did NOT have wasting at birth also had incident wasting in the first 60 days? Or is that assessment of high incidence of wasting in early life an artifact of this assumption?

Response: We agree with the Reviewer that this assumption is a possible limitation, and Figure 4b shows that the wasting incidence rate in the first three months of life is much lower after excluding episodes of wasting that started at birth, though the incidence rate is still higher than at other ages. After excluding at-birth episodes of wasting, the incidence rate of wasting in the 0-3 month age period was 1.33 (95% CI: 0.68, 1.97), and the cumulative incidence was 12.9% (95% CI: 7.6 20.9). With all cohorts included, as in Fig 4c, the incidence rate for the 0-3 month age period was 3.93 (95% CI: 2.40, 5.47) and the cumulative incidence was 11.9% (95% CI: 7.0 19.5).. We therefore conclude that the analysis was not sensitive to this assumption.

Regarding the reviewer's specific question about incident wasting in the first 60 days, among children measured at birth and not wasted (n =1,357), 12% experienced wasting in the next 60 days, (95% CI: 7.0%, 20.0%), with a mean WLZ of -0.8.

There may be differences in the causes of pre-natal growth failure leading to wasting at birth and the causes of early postnatal wasting that are important to differentiate, but we believe that both at-birth and near-birth wasting episodes are critical to study and prevent because of the higher risk of later growth failure and mortality both experience.

18. Line 336 – this sentence, “Wasting treatment programs have traditionally focused on children from 6-59 months.” Is probably one of the single most important statements in this entire paper. What is the history of that focus? Pragmatic? Evidence based? If evidence based, does the current analysis suggest therefore that the phenotype of CGF has changed? Or is the finding instead that the historical, programmatic guidance has been chronically misguided? Either way, this theme needs to be brought forward and emphasized.

Response:

The WHO recommends exclusive breastfeeding in the first 6 months of life, which is deemed adequate to meet the growth and development requirements of all infants around the world. It is also assumed that if exclusive breast feeding is appropriately followed growth faltering is uncommon during this period. Anthropometry is also not frequently measured during this young age including at birth. There is limited representation of this age group even in large scale DHS data, which doesn't include birthweight data. Suboptimal breast feeding practices are well documented (<https://pubmed.ncbi.nlm.nih.gov/26869575/>) and thus, the programmatic focus and efforts have been directed to promotion of exclusive breastfeeding (rather than its consequence, which may well be growth faltering). In parallel a related WHO recommendation is the introduction of complementary feeding at 6 months of age, which has also been found to be suboptimal, and linked to diarrheal and other infection risk due to introduction of unsafe complementary foods in the diets of infants and related growth faltering, requiring intervention. Finally, the severe acute malnutrition treatment protocols call for special therapeutic foods, which are not meant for exclusively breast-fed children, and therefore can only be used after 6 mo of age. Thus, historically and programmatically 6-24 and 6-59 month olds have been the focus of wasting treatment programs. We have

19. Line 591 – how is this different than “period prevalence”?

Response: We have updated the Methods on lines 789-795 to better explaining incidence proportion and the difference from period prevalence:

“Incidence proportion of wasting was calculated during a defined age range (e.g. 6-12 months) as the proportion of children not wasted at the start of the period who became wasted during the age period (the proportion of children who had the onset of new episodes during the period). This differs from period prevalence, which would include any children who had any measurement of WLZ < -2 during the period (including children who began the period wasted). Period prevalence would thus additionally include in the numerator and denominator children who were wasted at the start of the period, whereas incidence proportion excludes them.”

20. Line 593 – what is the conceptual basis/ principle for this definition of “recovery from wasting”?

Response: Since there is no standard definition for wasting recovery, we used a pragmatic definition of 60 days to reduce the chance that measurement error or a temporary fluctuation in WLZ would lead a child to be classified as recovered. In Fig 4d, we show the different recovery rates under different follow-up periods from wasting onset (30d, 60d, 90d). The results show that most episodes that resolve do so within 60 days, with only a small percentage additionally recovering by 90 days — a pattern that was consistent across ages and geographic regions. In the online supplement we completed a sensitivity analysis of the effect of changing the recovery definitions on wasting incidence rates (rather than the recovery proportions shown in 4d; https://child-growth.github.io/wasting/ir_sensitivity.html). The sensitivity analysis (figure copied below) shows that the choice of recovery period does not affect our interpretation of the age-specific patterns in wasting incidence rates.

21. References

21a. Line 1 #1 (growth standards definitions) is not relevant here (suggest dropping) and #2 (black et al 2013) is quite out of date (suggest updating).

Response: Per Referee 3 comment #2, we have rewritten the two introductory paragraphs in their entirety to improve their presentation and clarity. This sentence is no longer in the manuscript.

21b. Line 131 Neither #1 nor #6 support the statement in this sentence.

Response: Per Referee 3 comment #2, we have rewritten the two introductory paragraphs in their entirety to improve their presentation and clarity. This sentence is no longer in the manuscript.

21c. Suggest carefully checking all references to ensure they are appropriately matched to the statements they are intended to support. Is it possible that something got out of sync?

Response: We have carefully checked all references and have confirmed that in the revised manuscript all references support statements where they are cited.

Reviewer Reports on the First Revision:

Referees' comments:

Referee #1 (Remarks to the Author):

As I commented in the stunting paper, pooling data is useful to get higher precision and granularity, the cohorts pooled are of good quality. Wasting, which is an episodic condition, compared to stunting, requires even more attention to how to pool and how to interpret pooling result so removing early cohorts, in my view, oversimplifies the issue of heterogeneity and especially of generalizability.

In terms of whether region should be the sole basis for stratifying studies, my comments in the stunting paper also apply here. One could argue that for wasting, other strata like time are even more relevant – after all, if the data on rise in obesity in young children that WHO and UNICEF regularly release are true, there should also be a temporal dimension to wasting, even if in interplay with shorter term phenomena like seasons.

I still find the statement below epidemiologically problematic and, without being put in the context of inter- and intra-child dynamics of wasting, misleading:

“By age 24 months 29.2% of children had experienced at least one wasting episode, more than 5-fold higher than point prevalence (5.6%), demonstrating that wasting incidence is far higher than cross-sectional surveys suggest.”

I trust the calculations are mechanically correct. But the results are from two different types of data, and more importantly on two metrics that are designed to measure two different phenomena: life-time incidence vs. point prevalence. Take an equally correct statement: the number of people who ever have a headache (or common cold) is far higher than the number who have a headache/cold on any given day. Would this mean that pain medicine should get much more resources than it does and everyone being targeted for it? By presenting what is an obvious and inherent characteristic of episodic conditions as a “result”, and mixing point and life-time metrics, the statement risks mis-informing interventions: do we need regular health service response for wasted children (the point prevalence), perhaps with a seasonal plan, or do we need changes in nutrition supplements or is this about how land and agriculture resources should be (re)distributed? What could make the paper's results novel and actionable for targeting prevention/treatment would be to take advantage of having a large sample size and actually quantify the inter- and intra-individual temporal dynamics and frequency of wasting, including in relation to age (and for relevant subsets of studies): What is the distribution of number of episodes? what is the age distributions of first/second/etc episodes? How does being wasted at a specific age increase/decrease the risk for X months later or equivalently are these episodes independent? How do the first/subsequent/cumulative episodes follow from, or lead to, stunting? The seasonality analysis can then also be done in relation to these dynamics so that it goes beyond the state of knowledge: for example, are children who are wasted in the rainy season more at risk for the dry season or are these independent phenomena? Are the same children repeatedly wasted in rainy season or are different children affected?

The paper concludes that “Our results motivate a new focus on extending preventive interventions for wasting to pregnant and lactating mothers, ...” If the world were viewed only from the perspective of incidence (Figure 2B) this would make sense. But figure 4 seems to show that even WLZ itself largely converged between the two groups. More relevantly, to make such a statement there has to be a health argument based on a formal quantitative analysis: would an intervention (or in this case not being wasted) at that age change outcomes at older ages, conditional on wasting status at older ages? And what are those interventions – are they antenatal or neo/post-natal?

Partly related to the above comment, if in the statement below, “peak prevalence” refers to point

prevalence and not a cumulative measure, presumably the only interpretation of the statement "Here, we show through an analysis of 21 longitudinal cohorts that wasting is a highly dynamic process of onset and recovery, and incidence peaks between birth and 3 months — far earlier than peak prevalence at 12-15 months." would be that that episodes at 0-3 months are of shorter duration (assuming that figure 1 means that the frequency of measurement did not change over age for most cohorts so that the result is not affected by measurement). If so, the result can only be relevant together some measures of duration of the episode and how it matters for health including for the stunting results that are presented.

I am glad the seasonality analysis has been revised. On new results, the statement below in abstract does not seem to be supported by Figure 3B; rather, in the blue and green groups, the lowest mean was not in peak rainy season but in pre-peak season.

"In diverse populations with seasonal rainfall, population average weight-for-length varied substantially ($>0.5 z$ in some cohorts), with the lowest mean Z-scores during the rainiest months"

The analysis aside, the seasonality text is number heavy but should take advantage of larger sample size to make generalizations or contrasts. The statement that "the present study has not elucidated the mechanism that links seasonal rainfall with wasting" is a bit disappointing for a result that is made a big deal of. Although the authors may not have the relevant data to do so, there is a literature in seasonality of wasting (and nutrition more generally) with some studies in east Africa and south Asia going back decades which can be drawn upon in relation to the specific quantitative results with real implication for interventions – is this about water/waste? Is it about lingering effect of earlier dry periods in terms of available food? Or is it something else?

Referee #2 (Remarks to the Author):

Thanks to the authors for addressing specific comments provided. Overall, I have a minor comment along with one major remaining comment which are as follows:

Line 257 - are you really talking about rates here?

Broadly, when assessing and pinpointing the originality and novelty of this work, the latter primarily rests on the expansive sample size and geographical variation and original study design allowing for large population-based longitudinal assessment of growth faltering. The authors, across all 3 papers, highlight this sufficiently. What I find lacking, and in this paper particularly is any real discussion about how these findings compare, contrast and/or enhance other longitudinal, context-specific explorations on this same topic, i.e., the concurrence of wasting and studies. There are many beyond the usual Gambian study (Schoenbuchner et al) that is often cited. It would do the paper and its reader more justice to have a more in-depth discussion of these other papers that conduct very similar work in contrast to your own. Some references include:

Mutungu, M., Rutishauser-Perera, A., Lailou, A., Prak, S., Berger, J., Wieringa, F. T., & Bahwere, P. (2021). The relationship between wasting and stunting in Cambodian children: Secondary analysis of longitudinal data of children below 24 months of age followed up until the age of 59 months. *PLoS one*, 16(11), e0259765. <https://doi.org/10.1371/journal.pone.0259765>

Kohlmann, K., Sudfeld, C.R., Garba, S. et al. Exploring the relationships between wasting and stunting among a cohort of children under two years of age in Niger. *BMC Public Health* 21, 1713 (2021). <https://doi.org/10.1186/s12889-021-11689-6>

Per O. Iversen, Moses Ngari, Ane C. Westerberg, Grace Muhoozi, Prudence Atukunda, Child stunting concurrent with wasting or being overweight: A 6-y follow up of a randomized maternal education trial in Uganda, *Nutrition*, Volume 89, 2021,111281, ISSN 0899-9007, <https://doi.org/10.1016/j.nut.2021.111281>.

Odei Obeng-Amoako GA, Karamagi CAS, Nangendo J, et al. Factors associated with concurrent wasting and stunting among children 6-59 months in Karamoja, Uganda. *Matern Child Nutr.* 2021;17(1):e13074. doi:10.1111/mcn.13074

Referee #3 (Remarks to the Author):

Thank you for the hard work on the manuscript. It has been substantially strengthened, but a couple concerns remain that require further input from the authors.

Referee #4 (No remarks to the Author)

Referee #5 (Remarks to the Author):

This review focuses on the description of the analytical approach used for pooling the data presented in the Benjamin-Chung et al. article because that article was referenced in this article.

- I believe there is an error on p19, in lines 148-149 that describe the different parameters in equation 1. Further, τ^2 should be defined in case readers are unfamiliar with these equations. This would also make the difference between equations 1 and 2 clearer.
- Please define all parameters in equation 2 in lines 159-160.
- The following statement is included: If a model failed to converge, models were fit using a maximum likelihood estimator instead. In what analyses did this occur?
- In multiple places it is stated that the results using random effects models were comparable to those using fixed effects models. This would not be surprising if there is little heterogeneity in the results across studies. It would be helpful to provide a measure of the heterogeneity in the study-specific results either in the text or in the tables/figures when pooled results are provided. For example, there appears to be quite a bit of heterogeneity in the study-specific results in figure 3 for the South Asia region and in extended data figure 10.
- When conducting pooled analyses, you can examine potential sources of heterogeneity if results were heterogeneous across studies. Did you investigate potential sources of heterogeneity if present? This would provide additional information beyond just generating a pooled estimate.

Author Rebuttals to First Revision:

Referees' comments:

Referee #1 (Remarks to the Author):

1. Like the stunting and wasting papers, the paper has been systematically revised. My comments below focus on presentation, interpretation and contextualization in the light of metrics presented and especially the fact that the data used for causal inference (which is different from the stunting and wasting papers' descriptive scope) are observational.

Response: Thank you for your continued, constructive input to improve the paper. Your comments have led to additional refinements (detailed below), which we feel have substantially improved the presentation and interpretation of the results.

2. First, I think the paper should move away from the presentation of population intervention effect (which has now been made more explicit) and only present measures of risk that are independent of exposure. As the authors know, population intervention effect depends on both the level of exposure in the cohort and in the size of the causal association regardless of how simple or complex the function describing the latter is. The former is only relevant to the cohort studied and has no generalizability; the latter, presumably reflects some mechanism of risk that can be generalized. population intervention effect combines these two and hence can be interpreted as: in a group of children, at a specific place and time in the past, with some unspecified level of various exposures, population intervention effect was X%. For readers familiar with the metric, this is unhelpful; for those unfamiliar, it can be misleading as they may not realize the role of the exposure component. If they author want to show some sort of etiological fractions, they can do so at hypothetical exposures (e.g., in a population where X% of people have this level of exposure, like those in Country/Region X in 2020, etc).

Response: Thank you for sharing this perspective. We agree that presenting effects that are independent of the population's exposure level would provide valuable, additional results. We revised the central results presented in Figure 2 (now Fig 2 for LAZ, Fig 3 for WLZ) to display the average treatment effects for each exposure studied, along with the population intervention effect (PIE) estimated at a counterfactual level of exposure as in the previous version. We believe this higher resolution of information provides readers with a more complete view into the analysis and allows for a direct view into each exposure's independent association with child LAZ and WLZ. In discussion of the revised figure, the research consortium elected to remove mode of delivery (C-section, vaginal birth) as an exposure because the policy and public health implications of any

result would be unclear. Instead, we replaced it with small for gestational age (SGA), which integrates gestational age and birth weight — whilst providing a small bit of redundancy with gestational age and birth weight, SGA presents a key, biologically meaningful exposure that has clear public health interpretation and implications.

In the development of the analysis plan for this project, we discussed at length the relative merits of presenting different parameters of interest. We have presented Population Intervention Effects (PIE) because they are widely considered to be a more accurate depiction of the population-level importance of a risk factor. There are examples where observational analyses that estimate PIE more closely align with results from randomized trials because the effects are closer to what a real-world intervention could expect when delivered to the general population (one recent example from members of our team related to sanitation and child growth: Rogawski-McQuade et al. 2022 <https://pubmed.ncbi.nlm.nih.gov/34151953/>). Additionally, the PIE parameters are analogous to other large-scale syntheses, such as the Global Burden of Disease project at the Institute for Health Metrics and Evaluation, which uses population attributable fractions for binary outcomes (also see response to next comment), and are generally regarded as most informative in the translational step of moving from estimated effects (or associations) to expected impacts of intervening on exposures in actual populations (Westreich et al. 2016 <https://www.ncbi.nlm.nih.gov/pmc/articles/PMC4880276/> , Westreich 2017 <https://pubmed.ncbi.nlm.nih.gov/28282339/>).

As the referee notes, the PIE is a function of the prevalence of exposure in the population. An idea we discussed at the planning stage of the study was to use exposure distributions from outside data sources, such as the Demographic and Health Surveys (DHS) — however, such surveys do not include most of the exposures that were of interest in this study, so standardizing exposure distributions using DHS was impossible. Since this analysis draws from a large number of cohorts across diverse populations we felt that PIE estimates would be meaningful even if based on the empirical distribution of the exposures in the cohorts.

We added this caveat to the Discussion:

Lines 363-367

“Population intervention effects were based on exposure distributions in the 33 cohorts, which were not necessarily representative of the general population in each setting. Use of external exposure distributions from population-based surveys would be difficult because many key exposures we considered, such as at-birth characteristics or longitudinal diarrhea prevalence, are not measured in such surveys.”

- 3. As a secondary issue, I suggest replace population intervention effect with population attributable fraction because the former, even though understood by causal inference epidemiologists to imply specific inference procedures, could easily be interpreted by the great majority of readers as the effect a real (vs. analytically constructed) intervention.**

Response: Thank you for this suggestion. We have presented population attributable fractions for binary outcomes. The population intervention effect (PIE) is a generalization of the population attributable fraction and allows for any type of outcome. Since our primary inference focused on continuous measures of child growth (LAZ, WLZ) we have used “population intervention effect” to be consistent with the terminology widely used in epidemiology (Westreich 2017 <https://pubmed.ncbi.nlm.nih.gov/28282339/>). When estimating comparable effects for binary outcomes (stunting, wasting), the effects are labeled “population attributable fractions” (Extended Data Figs 6, 7).

- 4. Second, while the authors are using causal inference methodology, such methods depend entirely on what confounders were assumed to be relevant and which ones had data available – leading to the usual concerns about residual confounding. Therefore the authors should systematically and comprehensively put their results in the context of current knowledge (a model that medical journals have developed over the years so that epidemiological studies do not overlook all the other evidence). Currently this is done briefly, in my view far too briefly, in the last paragraph of P. 9. The right way to do this is to have a table that has a row for each risk factor and presents the association estimated in this study together with a summary for the following types of evidence (separately from one another): trials, other prospective studies (other data/other methods compared to what is used here) and retrospective cohort studies when exposure is unlikely to have changed in relation to outcome (e.g., in DHS, it is impossible or highly unlikely for parity, type of toilet or parental education to change in response to a child’s growth). I realize this will involve additional work but for the sake of good science and policy, this sort of contextualization is necessary and can overcome the concerns that we all have about extent of residual confounding and model dependence in observational causal inference. I also realize how studies report things (measures of risk, age groups, etc) are not always comparable but it should be the authors’ responsibility to synthesize across these, as well do when we report etiological effects.**

Response: We agree that contextualization is important. A comprehensive, systematic review of the literature is valuable for any scientific report. That level of review is feasible when a study focuses on a single exposure or intervention — the typical scenario for many medical journal articles, in our experience. However, this study has assessed 30 different exposures with respect to linear growth faltering and weight-for-length. Conducting a systematic review of the literature, synthesizing it, and reporting results for each of 30 exposures and multiple outcomes is, in our view, beyond the scope of this paper. In response to this suggestion, we have updated Extended Data Table 2 to summarize key evidence for the exposures studied in this analysis. The evidence we summarized is not the result of a systematic review on each question (though we have listed recent reviews where present). We conducted a literature search for each risk factor and, based on the available evidence, preferentially report findings from meta-

analyses of trials, individual trial results, meta-analyses of cohorts, or pooled demographic and health survey analyses. After reviewing the summary, we made comparisons of effect estimates from the present study to previous studies where possible. Most of the previous reports on this topic were not from randomized studies, included older children on average than our study, and were from cross-sectional surveys or smaller cohorts, so we believe prior findings cannot be used to assess the extent of residual confounding or model dependence in this study. Additionally, there was substantial variation in study design, populations, exposure definitions, analytic methods, and specific comparisons, so we note specific differences between prior research and our study that limit comparability for each risk factor. This heterogeneity in the scientific record was a key motivation for this study's novel synthesis of 33 cohorts.

- 5. Related to the above, and for transparent reporting, the papers needs an appendix table that shows for each exposure: the variables adjusted for and those that were not adjusted for on the premise that they were mediators. This would be a summary of the DAG for each risk factor (I am assuming from the text that there was a separate DAG for each risk factor; if this was not the case, the text needs to be made for explicit). I leave the review of multivariate application of targeted maximum likelihood estimation to reviewers with expertise in causal inference methods.**

Response: Thank you for this suggestion. This paper includes a lot of online supplemental information and might have not been clearly linked in the text. The table the referee has requested is available in the online supplement, and we have now more clearly linked to this in the Methods: <https://child-growth.github.io/causes/dags.html>

Regarding a separate DAG for each risk factor, the team of faculty and postdocs that underwent the exercise at Berkeley did so on paper and white board through in-person sessions, and the main record from the consensus discussion was the covariate lists summarized in the table referenced above. We included an example DAG to illustrate the process in the online supplement along with the table (link above), but recreating the other 29 DAGs electronically would delay the revision substantially and we felt would not meaningfully improve validity assessment or reproducibility of the analysis.

- 6. Finally, the statement on pre-natal vs. post-natal opportunities for growth (P. 9) is far stronger than supported by this and other studies, and should be toned down. If this were true, at the extreme there is little to be done for growth catch up once a child reaches post-natal period. But programs like the Dutch school milk programs have shown that catch up is possible throughout the entire childhood and adolescence. The findings, and their implications, should be systematically discussed in context of our broader knowledge (among others, see Prentice et al Am J Clin Nutr 2013, and Georgiadis and Penny Lancet Public Health 2017).**

Response: Thank you for this suggestion. We agree and have revised the discussion in two ways. First, we clarified that our focus has been on the first 24 months but that does not preclude the potential for consequences (and interventions) at older ages:

Lines 375-380

“A final caveat is that we studied consequences through age 24 months — the primary age range of contributed ki cohort studies — and thus have not considered effects on longer-term outcomes. Several studies have suggested that puberty could be another potential window for intervention to enhance catch-up growth. Improving girls’ stature at any point through puberty could help blunt the intergenerational transfer of growth faltering by improving maternal height, which in turn could improve outcomes among their children (Figs 2, 3, 4a, 4b).”

Second, we have distinct language in the revised discussion that speaks to this point:

Lines 358-362

“Maternal anthropometric status strongly influenced birth size, but the parallel drop in postnatal Z-scores among children born to different maternal phenotypes was much larger than differences at birth, indicating that growth trajectories were not fully “programmed” at birth (Fig 4a-b). This accords with the transition from a placental to oral nutrient supply at birth.”

Lines 385-388

“A stronger focus on prenatal interventions should not distract from renewed efforts for postnatal prevention. The observed pre- and postnatal growth faltering we observed reinforce the need for sustained support of mothers and children throughout the first 1000 days.”

- 7. Analytically, the way to make the evidence on how pre-natal and post-natal periods matter more robust and relevant, and to link this paper with the two descriptive papers, is to do separate/stratified analysis of risk factors by birth size (clearly identifying those risk factors that are expected to act pre- and post-birth): what are the risk factors for growth catch up/non-catch up in children with small birth size? And what are the risk factors for growth failure/non-failure for children with healthier birth size? (this is different from Figure 4 which treats early infancy failure as a risk factor; rather it would involve analyzing the effects of other risk factors, conditional on status in early infancy).**

Response: In response to the referee’s suggestion, we have provided estimates of the relationship between each exposure and child LAZ or WLZ at 24 months, stratified by birthweight. In doing so, we reduced exposure levels so that they were binary to make the comparison clearer and to help ensure there would be sufficient observations in each strata given the additional level of stratification by birthweight. For example, rather than

consider maternal education categories of high, medium, and low, we compared (medium or high) vs low. A caveat to the analysis is that only 7 cohorts measured birthweight, and so estimates below are based on measurements from between 3 and 7 cohorts, depending on which measured each exposure of interest. Not all exposures were measured in this subset of cohorts.

Our broad conclusion from this analysis is that there is no substantial difference in measures of association between each exposure and child z-scores at 24 months among children with different birth status. Note that unlike stratifying by a confounder, the pooled estimates do not necessarily fall between the two stratified estimates for prenatal exposures. This is because birthweight is a mediator for them, and the complex causal structure can create unintuitive — and potentially less reliable — results, which we will discuss in more detail below.

Figure. Adjusted difference in length-for-age z-score (LAZ) and weight-for-length z-scores (WLZ) at 24 months, overall (unstratified) and stratified by child birthweight.

When developing the analysis plan for this project, we had several discussions amongst our team about whether to pursue this specific analysis. In the end, we did not for two reasons.

First, the analysis is complex in that birthweight is a likely mediator for many of the prenatal exposures we considered. Stratified estimates above thus represent *direct effects* measured in a model allowing for exposure-mediator interaction. The focus of our overall analysis has been on *total effects*, which include the overall effect through both direct and indirect (mediated) pathways. Capturing the overall effect is, in our view, most relevant when rank-ordering exposures based on population intervention effects because it reflects the expected impact in the study populations from a shift in the exposure.

Second, there is a long history in perinatal epidemiology of difficulties related to birthweight-stratified analyses — so much so, that it is often called the “birth weight paradox,” the structural basis for which was first described in 2006 as a form of collider-stratification bias (<https://pubmed.ncbi.nlm.nih.gov/16931543/>). Collider stratification bias results if there are unmeasured confounders of the mediator (birth weight) and the outcome (child LAZ at 24 months) — often leading to paradoxical results. This bias has been observed so often when conditioning on birthweight in studies of prenatal exposures that a member of this paper’s consortium wrote in 2009 it “...has provided food for thought, and papers for publication, for at least two generations of perinatal epidemiologists” (<https://pubmed.ncbi.nlm.nih.gov/19689490/>). There is reason to suspect this type of bias is present in the above analyses. There are several examples where the stratified (direct) effects, are larger than the pooled (total) effect. Child sex is one example — since the pooled effect is likely unconfounded (no common causes of child sex and LAZ or WLZ), stratifying by birthweight likely induces some bias in the effect away from the null.

For these two conceptual reasons, and the third practical reason that birthweight was not measured in a large number of cohorts (leading to restricted inference), we have not included this stratified analysis in the revision.

- 8. As a specific methodological issue, why impute using median/mode, for example compared to a multiple imputation approach? I suggest that the statistical reviewer is consulted on the implications of this. Regardless of the appropriateness of the method, for transparency there should be a table that shows extent of missingness by cohort and variable. Also for transparency, a complete case analysis should be presented as a sensitivity analysis.**

Response: As the referee has suggested, for transparency we added a new figure showing percent missingness by covariate and study and the complete-case sensitivity analysis (<https://child-growth.github.io/causes/missing.html>)

We would like to clarify that imputation was only used for covariates in adjusted analyses — outcomes and exposures were not imputed and so the study reflects a complete case analysis with respect to outcomes and exposures. We clarified this in the methods:

Lines 1091-1092

“Analyses used a complete case approach that only included children with non-missing exposure and outcome measurements.”

We used imputation for adjustment covariates to maximally include them where possible. To our knowledge, there is no straight-forward way to use a multiple imputation approach for covariates in individual cohort estimates that are then combined into an individual participant data meta-analysis. Instead, for covariates in the adjustment set, we imputed them at the median (for quantitative variables) or mode (for categorical variables) and additionally included an indicator variable to flag imputed values (Stuart 2010). The approach is technically sound and, compared with multiple imputation, easily integrates into the TMLE and meta-analysis analysis framework — we provide additional technical details below.

Technical details. Imputing covariates is relevant to our estimates of the two factors of the likelihood: (1) the outcome regression model of Y versus the explanatory variable of interest, A , and covariates, W (estimates of $E(Y|A, W)$) and (2) the estimate of the so-called treatment mechanism as a regression of categorical exposure A versus W ($P(A=a|W)$). There are many ways missingness can be handled and no theory that can be used to definitely predict which will handle missingness in the most efficient and least biased fashion. The method we used with indicators of missingness for any covariate that has missing information and simple imputations of the covariate values from summary statistics has several virtues. First, it allows us to make an explicit identification assumption, that is an assumption under which our estimators will be unbiased, so that we can be transparent and explicit about such an assumption. Specifically, our transformation of the W leads to a new set of measurable covariates, or, for every variable with missing data, j , we create a new W_j^*, Δ_j , where W_j^* contains either the original value of W_j if observed, or the imputed value, and Δ_j is the indicator of being observed ($=1$ if W_j was observed, 0 if missing). Then, the new W^* are used in the SuperLearner (SL) algorithm, an ensemble of regression models and machine learning algorithms for each of the factors in the likelihood. The SL is fit to minimize the squared residuals, and can use the flexibility to glean as much predictive information that is available in the actual covariate data, which W^* represents. Because the goal is “explain” as much residual variance as possible, the SL uses the actual observed data to control as much confounding as possible (assuming the original W would be preferable for this goal). Thus, the SL will use the predictive information contained in the Δ_j to adjust for confounding as aggressively as possible (that is if the fact that the values are missing is predictive in itself). In this new data structure, we thus can assert no residual unmeasured confounding based upon an explicit assumption, that is the variable of interest is independent of the counterfactuals, Y_a , conditional not on the fully measured W but on our transformation W^* , or $A \perp Y_a | W^*$. Another advantage of this approach is higher efficiency compared to an analysis that would exclude children with missing adjustment covariates. In addition, if many children are missing a single or just a few

weak confounders, then limiting the analysis to children with non-missing values of the weak confounders can result in a significant selection bias or become untenable if a very large proportion of observations have at least one covariate missing.

Stuart EA. Matching methods for causal inference: a review and a look forward. *Stat Science*. 2010;25(1):1–21.

Referee #2 (Remarks to the Author):

1. The detail and depth of the analysis are exemplary. Thank you for addressing my questions. I don't have additional comments, except that I spotted some typos in the revision:

Line 361: highlights should be plural.

Line 897-899: The sentence "We also calculated the difference in linear and ponderal growth velocities over three-month periods." appears twice.

Line 1022: " is show bin"

Response: Thank you for identifying these typos. We have fixed them all in the revision.

Referee #3 (Remarks to the Author):

Overall, the authors have adequately addressed issues I, as well as other reviewers have raised. The only pending points from my read are:

1. Lines 294-295: Authors do not provide a clear direction/ magnitude in this hypothesis.

Response: Thank you for this suggestion. The original wording was intentionally agnostic about the direction and magnitude because it differs by exposure and there are so many different exposures. In the revision, we replaced the topic sentence to provide two examples to help ground the hypothesis in more specific terms.

Lines 285-287:

"We hypothesized that causes of growth faltering could differ by age of growth faltering onset — for example, we expected children born preterm would have higher risk of incident growth faltering immediately after birth, while food insecurity might increase risk at older ages, after weaning."

2. Lines 399-405: There is a heavy focus on the conclusion the conclusions of the other papers.

Response: We agree. We revised the discussion to remove extensive reference to the companion papers.

- 3. Sex, specifically boys being consistently associated with being stunted, wasted or both, both as pooled estimate and across most study and regional contexts, was a finding of consequence and one that finds itself in the summary but is not flushed out as a finding at all throughout the paper. The authors may chalk this up to the fact that it is not modifiable but discussing this finding meaningful seems essential given the findings prominence.**

Response: We included a new sentence in the results that described this in more detail, and cites a recent review that proposed mechanisms. The revised text is:

Lines 251-252

“Girls had consistently better LAZ and WLZ than boys, potentially from sex-specific differences in immunology, nutritional demands, care practices, and intrauterine growth (Thurstans et al. 2022).”

Thurstans, S. et al. Understanding Sex Differences in Childhood Undernutrition: A Narrative Review. *Nutrients* 14, 948 (2022). <https://pubmed.ncbi.nlm.nih.gov/35267923/>

Referee #4 No remarks to the Author

Referee #5 (Remarks to the Author):

- 1. I would like to congratulate the authors on a tour de force of data harmonization, data analysis, and interpretations. What makes this work novel is that the underlying methodological framework allows for (almost) direct conclusions and comparisons across diverse cohorts. This would not have been possible without a statistical analytical framework, which separates sampling and data modelling, from interpretation. The former will be cohort depended while the latter can be universal. To the best of my knowledge this is the first time – within the subfield – such cross-cutting data analysis has been completed.**

Response: Thank you for this summary. We are grateful for the constructive suggestions, below.

- 2. A single methodological concern is if the cohorts have been “collected for a reason” and therefore perhaps are less representative for the country/population. This could be further addressed. In the methods section is it also somewhat**

unclear – at least for this reader – which values covariates are set to when computing population effects. Are they from the literature or derived from the cohorts? It seemed to be a mix. In short: the basic scientific merit and methodology is top-notch.

Response: This is an important detail that we have also discussed in our response to Referee 1 comment #2 (above). We agree that in the last version of the paper the chosen reference level for each exposure was listed but not in the most transparent way. We revised the previous Fig 2 (now Figs 2, 3) to include adjusted mean differences for each exposure level and clearly labeled the value each exposure was set to when estimating the population intervention effects (PIE). We agree with the referee that since the cohorts are not necessarily representative of the general population, that PIE estimates might be different if we had used different counterfactual distributions of the exposure. Many key exposures in this analysis are not measured in representative, population-based surveys, so it is difficult to establish a reference distribution for LMICs or study regions. Since this analysis draws from 33 cohorts across diverse populations we felt that PIE estimates would be meaningful even if based on the empirical distribution of the exposures in the cohorts. By additionally including the adjusted mean differences, which are independent of the exposure distribution, we have provided another view through which readers can assess the relative importance of each exposure.

We added this caveat to the Discussion:

Lines 363-367

“The analyses have caveats. Population intervention effects were based on exposure distributions in the 33 cohorts, which were not necessarily representative of the general population in each setting. Use of external exposure distributions from population-based surveys would be difficult because many key exposures we considered, such as at-birth characteristics or longitudinal diarrhea prevalence, are not measured in such surveys.”

- 3. The presentation is fairly verbal. It is of course a matter of taste if this is preferable. I would have preferred a few of the key-findings had been highlighted through tables. Instead the table material is mixes both summaries and very detailed. Given the amount of sensitivity analyses etc (which is of course good) it will be very difficult for the average reader to really take in the quantitative results. This is regrettable as one of the strengths of the paper exactly is the quantifications. In short: I suggest to make it more clear what the key tables are and where is more for the dedicated reader.**

Response: Our overall strategy was to use figures to synthesize the results, since the data underlying the analyses are very dense, and then use the text to highlight key findings (with reference to relevant main text and extended data figures). The referee's suggestion to summarize key points in a table or box is a very good suggestion. However, given the space limitations in the article format in the revision we have

continued to present the quantitative results in figures and use the text to summarize key messages from each figure to help general readers interpret the results.

- 4. The paper finds that the most important aspects causing missed growth are biological – and very hard to change. This has the obvious effect that whilst important biologically, it does really help for prevention. This is only partly acknowledged. It is said that the insight can be used to find out targets for interventions, but since no intervention can ever deliver the kind of changes hypothesized in the paper, it could be that the importance of certain aspects are over-sold. In short: The population effect reported in the paper should in some sense be multiplied by a factor (smaller than 1) corresponding to “ease of intervention”.**

Response: We acknowledge that many of the key determinants of growth faltering identified in this analysis are very hard to change and realistic interventions could not modify many of the characteristics over a short time frame — such as household wealth or maternal education. For unmodifiable characteristics, knowing they are important determinants can inform interventions because it motivates strategies to mitigate them (e.g., seasonal wasting) and/or targeting to subgroups. Comparison between risk factors on a single scale is also valuable. For example, preventing low birthweight would be expected to have similar impact on child LAZ as moving all households into the richest wealth quartile — this type of comparison puts into perspective how valuable interventions could be on that modifiable characteristic (low birthweight) for which there are known, effective interventions, such as nutritional supplementation for pregnant mothers and IPTp-SP treatment for malaria-endemic populations. Finally, demonstrating that key causes of growth faltering are in many cases intergenerational (from mothers) helps calibrate the global community to a realistic timeframe required to make progress on preventing child growth faltering — since our results show that prevention of child growth faltering will require improvements in maternal status, this implies a slower path to population-level improvements (years, decades) compared with health conditions that can be treated or prevented over the course of weeks or months.

We revised the final paragraph of the Discussion so that it clearly emphasizes modifiable characteristics and a path forward, namely: maternal nutritional status and child status at birth.

Lines 381 – 387

“Countries that have reduced stunting most have undergone improvements in maternal education, nutrition, reductions in number of pregnancies, and maternal and newborn health care, reinforcing the importance of interventions during the window from conception to one year, when fetal and infant growth velocity is high and energy expenditure for growth and development is about 50% above adult values (adjusted for fat-free mass). A stronger focus on prenatal interventions should not distract from renewed efforts for postnatal prevention. The observed pre- and postnatal growth

faltering we observed reinforce the need for sustained support of mothers and children throughout the first 1000 days. Efficacy trials that delivered prenatal nutrition supplements to pregnant mothers, therapeutics to reduce infection and inflammation for pregnant mothers, and nutritional supplements to children 6-24 months have reduced child growth faltering but have fallen short of completely preventing it. Our results suggest that the next generation of preventive interventions should focus on the early period of a child's first 1000 days — from preconception through the first months of life —because maternal status and at-birth characteristics are key determinants of growth faltering through 24 months. Halting the cycle of growth faltering early should reduce the risk of its severe consequences, including mortality, during this formative window of child development. Long-term investments and patience may be required, as it will take decades to eliminate the intergenerational factors limiting mothers' size.”

Reviewer Reports on the Second Revision:

Referees' comments:

Referee #1 (Remarks to the Author):

Like the stunting paper, the additional analyses provide valuable information, in particular those related to the number and duration of wasting episodes as well as the additional seasonality analyses.

Could the results in Figure 3B please be shown in two forms: as distribution of number of episodes (as currently shown) AND as distribution of proportion of all episodes – since the latter does not depend on the frequency of measurement. Alternatively, can a panel be added to Extended Data Figure 4 that gives the distribution of the proportion of visits at which children have wasting; Panel A of this figure does so for >50% but it would be useful to see the full distribution rather than a somewhat arbitrary cut-off of 50%. This additional graph should be easily doable based on the data that were used to create these figures.

Finally, rather than dwelling on how episodes of viral cold, which can lead to repeated low-grade inflammation, compares with episodes of wasting in terms of its life-long effects, I suggest the sentence "By age 24 months 29.2% of 106 children had experienced at least one wasting episode ..." in the abstract is expanded to include the information on distribution – i.e., change "at least one" with the distributional information either as number (X% with 1, Y% with 2, Z% with 3 or more) or as % of visits.

I would also change "wasting affects far more children than can be inferred through cross-sectional surveys" to something like "far more children experience an episode of wasting at some point during their first X months of life than prevalence cases at a single point in time" which is a less loaded wording and allows readers to decide the relevant metric for their purpose, be it point prevalence or cumulative incidence.

Referee #3 (Remarks to the Author):

I am quite satisfied with the authors' revisions to this manuscript. This is a really nice piece of work. Congratulations.

Referee #5 (Remarks to the Author):

Overall, the responses to the comments about the analytical approach of pooling the data are clear. However, given the authors response below to comment #4 from referee 5,

In wasting analyses, the point estimates for fixed effects models sometimes differ greatly from the random effects models, and fixed-effect confidence intervals are much narrower, because of heterogeneity in the background rates of wasting across cohorts.

For example, the largest difference in estimates from random versus fixed effects models is an estimated 3x greater prevalence of wasting at birth in Africa when using fixed effects models (24% versus 8%), driven by the MRC Kenaba cohort which is both the largest African study and has the highest wasting prevalence at birth. However, overall the two approaches do not lead to different scientific inferences around age-specific patterns or regional differences.

I do think it's important to point out that the choice of model impacted the strength of the association but that the overall conclusions were similar. I also think it would be helpful to point

out in a summary way a measure of heterogeneity especially since some think pooled estimates should not be presented if there is significant heterogeneity while others feel that using fixed effects models are preferred over random effects model.

**Author Rebuttals to Second Revision:
Referees' comments:**

Referee #1 (Remarks to the Author):

Like the stunning paper, the additional analyses provide valuable information, in particular those related to the number and duration of wasting episodes as well as the additional seasonality analyses.

Could the results in Figure 3B please be shown in two forms: as distribution of number of episodes (as currently shown) AND as distribution of proportion of all episodes – since the latter does not depend on the frequency of measurement. Alternatively, can a panel be added to Extended Data Figure 4 that gives the distribution of the proportion of visits at which children have wasting; Panel A of this figure does so for >50% but it would be useful to see the full distribution rather than a somewhat arbitrary cut-off of 50%. This additional graph should be easily doable based on the data that were used to create these figures.

Response: We have added a set of histogram plots to the new panel B in the Extended Data Figure 5 (copied below) that shows the distribution of the proportion of visits at which children have wasting overall and by region. We reference this new figure in the manuscript on lines 310-312:

“Children in South Asia had the highest proportion of measurements with wasting (Extended Data Fig 5b) and the highest prevalence of persistent wasting over the first 24 months of life (7.2%, 95% CI: 5.1 to 10.4, Extended Data Fig 5c).”

Extended Data Figure 5B. Histograms of the proportions of visits at which children have wasted measurements, overall and stratified by region (20 bins at 5% bin-width). The inset plots show the percent of wasted measurements among children with any wasted (excluding children never wasted under two years of age).

Finally, rather than dwelling on how episodes of viral cold, which can lead to

repeated low-grade inflammation, compares with episodes of wasting in terms of its life-long effects, I suggest the sentence “By age 24 months 29.2% of children had experienced at least one wasting episode ...” in the abstract is expanded to include the information on distribution – i.e., change “at least one” with the distributional information either as number (X% with 1, Y% with 2, Z% with 3 or more) or as % of visits.

I would also change “wasting affects far more children than can be inferred through cross-sectional surveys” to something like “far more children experience an episode of wasting at some point during their first X months of life than prevalence cases at a single point in time” which is a less loaded wording and allows readers to decide the relevant metric for their purpose, be it point prevalence or cumulative incidence.

Response: We have revised the wording in the abstract following these suggestions and now report the prevalence of repeated episodes briefly to stay within the limited the word count:

“Far more children experience an episode of wasting at some point during their first 24 months than prevalent cases at a single point in time suggest: at age 24 months 5.6% of children were wasted, yet by age 24 months 29.2% of children had experienced at least one wasting episode and 10.0% had experienced two or more episodes.”

Referee #3 (Remarks to the Author):

I am quite satisfied with the authors' revisions to this manuscript. This is a really nice piece of work. Congratulations.

Referee #5 (Remarks to the Author):

Overall, the responses to the comments about the analytical approach of pooling the data are clear. However, given the authors response below to comment #4 from referee 5,

In wasting analyses, the point estimates for fixed effects models sometimes differ greatly from the random effects models, and fixed-effect confidence intervals are much narrower, because of heterogeneity in the background rates of wasting across cohorts.

For example, the largest difference in estimates from random versus fixed effects models is an estimated 3x greater prevalence of wasting at birth in Africa when using fixed effects models (24% versus 8%), driven by the MRC Kenaba cohort which is both the largest African study and has the highest wasting prevalence at

birth. However, overall the two approaches do not lead to different scientific inferences around age-specific patterns or regional differences.

I do think it's important to point out that the choice of model impacted the strength of the association but that the overall conclusions were similar. I also think it would be helpful to point out in a summary way a measure of heterogeneity especially since some think pooled estimates should not be presented if there is significant heterogeneity while others feel that using fixed effects models are preferred over random effects model.

Response: We have added the median and interquartile range for the I-squared statistic to the figure captions of all main text figures that present meta-analyses as a summary way of showing heterogeneity. All estimates are also re-estimated using fixed-effects models and presented in Supplementary Note 2.